

# Integrating uncertainty propagation in GNSS radio occultation retrieval: from excess phase to atmospheric bending angle profiles

Jakob Schwarz[1], Gottfried Kirchengast[1], and Marc Schwaerz[1]

[1]Wegener Center for Climate and Global Change (WEGC) and Institute for Geophysics, Astrophysics and Meteorology/Institute of Physics University of Graz, Graz, Austria.

*Correspondence to:* Jakob Schwarz (jakob.schwarz@uni-graz.at)

**Abstract.** Global Navigation Satellite System (GNSS) radio occultation (RO) observations are highly accurate, long-term stable data sets, and are globally available as a continuous record since 2001. Essential climate variables for the thermodynamic state of the free atmosphere, such as pressure, temperature and tropospheric water vapor profiles (involving background information), can be derived from these records, which therefore have the potential to serve as climate benchmark data. However, to exploit this potential, atmospheric profile retrievals need to be very accurate and the remaining uncertainties quantified and traced throughout the retrieval chain from raw observations to essential climate variables. The new Reference Occultation Processing System (rOPS) at the Wegener Center aims to deliver such an accurate RO retrieval chain with integrated uncertainty propagation. Here we introduce and demonstrate the algorithms implemented in the rOPS for uncertainty propagation from excess phase to atmospheric bending angle profiles, for estimated systematic and random uncertainties, including vertical error correlations and resolution estimates. We estimated systematic uncertainty profiles with the same operators as used for the basic state profiles retrieval. The random uncertainty is traced through covariance propagation and validated using Monte-Carlo ensemble methods. The algorithm performance is demonstrated using test-day ensembles of simulated data as well as real RO event data from the satellite missions CHAMP and COSMIC. The results of the Monte-Carlo validation show that our covariance propagation delivers correct uncertainty quantification from excess phase to bending angle profiles. The results from the real RO event ensembles demonstrate that the new uncertainty estimation chain performs robustly. Together with the other parts of the rOPS processing chain this part is thus ready to provide integrated uncertainty propagation through the whole RO retrieval chain for the benefit of climate monitoring and other applications.

## 1 Introduction

Observation systems of the free atmosphere, focusing on the range from the top of the atmospheric boundary layer upwards, were historically designed for weather research and forecasting purposes. They have considerable shortcomings when looking at them from a climate monitoring perspective (Karl et al., 1995) and so the related global climate monitoring infrastructure remains fragile and incomplete until today (Bojinski et al., 2014). The Global Climate Observing System (GCOS) aims to improve the observational foundation for the climate sciences (GCOS, 2015). For this purpose the establishment of *climate benchmark data records* is essential. To qualify as climate benchmark, records need to be 1. of global coverage, 2. of high



accuracy, 3. long-term stable, 4. tested for systematic errors on-orbit, 5. tied to irrefutable standards, and they need to 6. measure Essential Climate Variables (ECVs) (National Research Council, 2007; GCOS, 2015).

Based on the quality and abundance of Global Navigation Satellite System (GNSS) signal sources, in particular from the Global Positioning System (GPS) so far, the GNSS radio occultation (RO) observation record is globally available (contin-
uously since 2001), long-term stable (due to the so-called self-calibration and high signal stability during the event), and highly accurate (accuracy traceable to the SI second). Due to the self-calibrating property, the accuracy is also ensured on-orbit (Leroy et al., 2006). The basic RO excess phase data can therefore serve as a Fundamental Climate Data Record (FCDR) as defined by GCOS (2010). From this FCDR with its unique properties, ECVs—in particular the thermodynamic ECVs pressure, temperature and humidity in the free atmosphere—can be derived using an RO *retrieval* chain.
In order to reliably serve as climate benchmark data record however, the retrieved ECV profiles and their claimed accuracy— expressed by the uncertainties provided— need to be traceable back to the (small) uncertainties of the FCDR and in turn to the raw data. This requires that 1. the RO retrieval is highly accurate and avoids any undue amplification of uncertainties associated with the quantities in the FCDR and that 2. the uncertainties are propagated through the entire retrieval chain, from the raw data to the ECV profiles, duly accounting for relevant side influences such as from background information. Developed at the
Wegener Center of the University of Graz (WEGC), together with international partners, the Reference Occultation Processing System (rOPS) (Kirchengast et al., 2015) aims to establish such a fully traceable RO processing for the first time (Kirchengast et al., 2016a, b).

In Figure 1 the basic steps of the RO retrieval chain in the rOPS, i.e., the precise orbit determination (POD) and excess phase processing (labeled 'L1a' in Figure 1), the subsequent atmospheric bending angle retrieval ('L1b'), the refractivity and dry-air
retrieval ('L2a'), and the moist air retrieval ('L2b') are sketched.

Kursinski et al. (1997), and more recently Hajj et al. (2002), Anthes (2011) and Steiner et al. (2011) provided detailed introductions and reviews of the RO technique and its applications in meteorology and climate. Ho et al. (2012) and Steiner et al. (2013) included comparative current RO retrieval chain descriptions of leading international RO processing centers, which all do not yet include uncertainty propagation. Empirical error (uncertainty) estimates computed statistically from retrieved RO
atmospheric profiles and climatologies have been derived by Kuo et al. (2004), Steiner and Kirchengast (2005) and Scherllin-Pirscher et al. (2011a, b, 2017), the latter with a focus on climate uses also providing simple analytical error models. These studies and many others have described the RO retrieval chain in detail and have shown the high accuracy and quality of RO data, particularly in the upper troposphere and lower stratosphere region.

The aim of the integrated uncertainty propagation in the rOPS is to eventually propagate uncertainties along this *entire*
retrieval chain from the raw measurement data to the ECVs (Kirchengast et al., 2016a, b), whereby the implementation of the rOPS uncertainty propagation occurs in the sequential blocks illustrated in Figure 1. The L2a processing and uncertainty propagation from atmospheric bending angle to dry-air profiles has already been introduced by Schwarz et al. (2017) [SKS2017 hereafter].

This study is a direct complement to the work in SKS2017. Using the same propagation and validation methods as ap-
plied in SKS2017, it focuses on the uncertainty propagation from excess phase to atmospheric bending angle profiles, i.e.,





the L1b processing. As in SKS2017, *random uncertainties* are propagated using covariance propagation (CP) and validated using Monte-Carlo ensemble methods (MC). As in the L2a processor, we also propagate (conservative bound) estimates for *systematic uncertainties* along the retrieval chain of the L1b processor. Additionally, *correlation length* profiles and *resolution* profiles are provided.

Uncertainty propagation as covariance propagation from excess phase to bending angle profiles has been outlined and demonstrated in a basic form, by Syndergaard (1999) and Rieder and Kirchengast (2001), but not been implemented yet in processing center retrieval chains and applied to real RO data. As visible in Figure 1, the L1b processor consists of three major retrieval parts, which are expanded into detailed substructure in Figure 2. We propagate estimated random uncertainties from excess phase profiles to Doppler shift profiles (Section (1) in Figure 2), further to geometric-optics (GO) bending angle profiles,

merged with wave-optics (WO) bending angle profiles (2), and finally to atmospheric bending angle profiles (3), using a full CP approach. In combination with the definitions of the main operators and variables in Table 1, and of the vertical grid and coordinate variables in Table 2, Figure 2 provides a concise overview on the detailed workflow of the L1b processor.

Uncertainty propagation for the WO bending angle retrieval has been implemented and demonstrated for simulated events by Gorbunov and Kirchengast (2015), estimation of random and systematic uncertainties for real events including boundary

layer bias correction is introduced by Gorbunov and Kirchengast (2017).

Other on-going rOPS retrieval advancements relevant to this study are the inclusion of the high altitude initialization algorithm, introduced by Li et al. (2013, 2015), in the L2a processor and the reduction of remaining higher-order ionospheric effects in the retrieved bending angle profiles of the L1b processor (based on work by Syndergaard (2000), Liu et al. (2015), Healy and Culverwell (2015) and Danzer et al. (2013, 2015)). Furthermore, the precise orbit determination (POD) of the RO receiver

satellite and the excess phase processing, also including the associated uncertainty propagation, are part of on-going work and manuscript preparation (Pock et al. 2017, manuscript in preparation for Journal of Geodesy, and Innerkofler et al. 2017, manuscript in preparation for Atmospheric Measurement Techniques; see also Innerkofler et al. (2016)).

Finally, related work and manuscript preparation on a new moist air retrieval algorithm (L2b) and corresponding L2b uncertainty propagation is on-going (Li et al. (2017) and Schwarz et al. 2017, manuscript in preparation for Remote Sensing of the

Environment; see also Kirchengast et al. (2017a)).

The paper is structured as follows. In section 2 we introduce the uncertainty estimation, propagation and validation methods and the data sources and preparation. In Section 3, with the help of a representative real RO event, the uncertainty propagation sequence is introduced. In Section 4 we present the results from the MC valdiation of the CP uncertainty estimates. In Section 5 the performance of the algorithm is then evaluated using test-day ensembles with real data from the RO missions CHAllenging

Minisatellite Payload (CHAMP) (Wickert et al., 2001) and FORMOSAT-3 Constellation Observing System for Meteorology, Ionosphere, and Climate (COSMIC) (Anthes et al., 2008), and with simulated data approximating characteristics of the Meteorological Operational Satellite A (Luntama et al., 2008) [simMetOp data hereafter]. We close with conclusions and outlook in section 6. A detailed description of the implemented uncertainty propagation algorithms can be found in Appendix A.



## 2 Methods and Data

### 2.1 Methods

We follow the *Guide to the expression of Uncertainty in Measurement* (JCGM, 2008a, b, 2011) [GUM hereafter] and aim to follow terminology as provided by the International Vocabulary of Metrology (JCGM, 2012), a terminology also adopted by the GUM. SKS2017 provides a more thorough introduction, including the motivation for using the respective uncertainty estimation, propagation and validation methods; we refer the closely interested reader to this companion (open access) work and provide the essential methods needed more in summarized form below.

We categorize uncertainties into *estimated random uncertainties* and *estimated systematic uncertainties*. Effects of unpredictable or stochastic temporal and spatial variations in repeated observations can be (also) estimated based on individual RO events—due to oversampling in the RO raw data profiles—and are included in the estimated random uncertainties.

Systematic effects (biases), which can not be quantified using statistical data analysis based on just one individual RO profile, are estimated and corrected for when known, as recommended by the GUM. The remaining residual biases are assumed to stay within a (conservative) bound estimate, which we refer to as estimated systematic uncertainty and by which we aim to provide at least 90 % likelihood coverage (confidence) that residual biases stay within the plus/minus envelope range of this uncertainty.

Depending on their nature, components of the systematic uncertainty that we need to estimate can be fundamentally systematic across different RO events, a subtype we term *estimated basic systematic uncertainties*, or appear systematic just for individual RO events, a second subtype that we term *estimated apparent systematic uncertainties*. It is important to distinguish these two subtypes, since the apparent systematic uncertainties will essentially behave as random uncertainties in ensemble-averaging over many RO events, such as when generating climatologies, while the basic systematic ones will not average out and therefore fundamentally limit the (absolute) accuracy of ensemble averages such as climatologies.

As for the L2a processor (SKS2017), the operators of the L1b processor (i.e., the boldfaced Items 1.2, 1.4, 2.1, 2.7, 2.9, 3.1, 3.5 in Figure 2) qualify as *explicit*, *multivariate*, *linear* measurement models, as defined in the GUM, with *correlated input* quantities. They can therefore be formulated as

$$Y = \mathbf{A}^{\mathrm{XY}} \cdot X, \tag{1}$$

where the input quantity $X$ and output quantity $Y$ are rank-1 vectors (profiles) of random variables, which we call *state* profiles. According to the GUM, their *random uncertainties* can be propagated using

$$\mathbf{C}_Y = \mathrm{E}[YY^{\mathrm{T}}] = \mathrm{E}[(\mathbf{A}^{\mathrm{XY}}X)(\mathbf{A}^{\mathrm{XY}}X)^{\mathrm{T}}] = \mathbf{A}^{\mathrm{XY}}\mathrm{E}[XX^{\mathrm{T}}](\mathbf{A}^{\mathrm{XY}})^{\mathrm{T}} = \mathbf{A}^{\mathrm{XY}}\mathbf{C}_X(\mathbf{A}^{\mathrm{XY}})^{\mathrm{T}}, \tag{2}$$

when the uncertainties are normally distributed. This assumption is reasonably justified, since the receiving system noise, i.e., thermal noise and residual clock estimation noise, and the ionospheric noise are essentially normally distributed (Kursinski et al., 1997; Syndergaard, 1999; Gorbunov, 2002b; Sokolovskiy et al., 2009) and are the main contribution to the random uncertainties in the excess phase profiles feeding into the L1b processor.





$\mathbf{C}_X$ and $\mathbf{C}_Y$ are the covariance matrices of the input and output variables, respectively, and $\mathbf{A}^{\mathrm{XY}}$ is the linear (or linearized) operator connecting $X$ and $Y$. Equation 3 formulates how the covariance matrix $\mathbf{C}_X$ is calculated from random uncertainty estimates $u_X^r$ and the correlation matrix $\mathbf{R}_X$,

$$C_{X,ij} = u_{X,i}^r \cdot u_{X,j}^r \cdot R_{X,ij}. \tag{3}$$

As a key variable characterizing $\mathbf{R}_X$, *correlation length* profiles $l_X$ are estimated from the correlation functions assembled in $\mathbf{R}_X$. The used algorithm estimates $l_X$ by searching for the distances downward and upward of the correlation functions' main peak at which the correlation function has dropped to to a value of $1/e\ (\approx 0.378)$. The adopted correlation length estimate is the arithmetic mean of these two upward and downward estimates (as the peak may be somewhat asymmetric). Additionally the correlation length is constrained by the data domain, i.e., the correlation length can never be larger than the profiles' vertical

range.

When the operator is linear, as is the case for the applicable L1b operators, estimated *systematic uncertainties* can be propagated by application of the state retrieval operator on the estimated systematic input uncertainty

$$u_Y^s = \mathbf{A}^{\mathrm{XY}} \cdot (X + u_X^s) - Y = \mathbf{A}^{\mathrm{XY}} \cdot X + \mathbf{A}^{\mathrm{XY}} \cdot u_X^s - Y = Y + \mathbf{A}^{\mathrm{XY}} \cdot u_X^s - Y = \mathbf{A}^{\mathrm{XY}} \cdot u_X^s, \tag{4}$$

where $u_X^s$ and $u_Y^s$ are the rank-1 systematic uncertainty profiles of the input and output variables.

In addition to random uncertainties, systematic uncertainties and the correlation length, we also estimate *resolution* profiles $w_X$ as context information along with the provided random uncertainty profiles (necessary, e.g., because smoothing can decrease random uncertainties, while making resolution coarser). This is enabled by careful selection and formulation of lowpass filter operations, in particular explicit filter cutoff frequency specification as the main driver of the resolution remaining after lowpass filtering.

We note that the (half-)Fresnel-scale *physical* resolution often ascribed to RO bending angle profiles retrieved by geometric-optics methods (e.g., Kursinski et al., 1997; Gorbunov et al., 2004) will generally be somewhat coarser than the *filter limited* resolution estimated here. This is intentional to maximize available information in the bending angle profiles provided by the L1b processor. In the rOPS, on input to the L2a processor and before high altitude initialization by statistical optimization, the resolution is then aligned to a joint altitude-dependent resolution for all profiles which reflects the half-Fresnel-scale

(SKS2017).

## 2.2   Data Sources and Preparation

The input variables needed for the L1b uncertainty propagation, visible in Figure 2 and defined in Table 1, are the excess phase profiles $L_{\mathrm{r},k}(t)$ and the associated systematic uncertainty profiles $u_{Lr,k}^s(t)$, random uncertainty profiles $u_{Lr,k}^r(t)$ and correlation matrices $\mathbf{R}_{Lr,k}$, as well as the orbit positions and velocities of receiver and transmitter satellite, $\boldsymbol{r}_{\mathrm{R}}(t)$, $\boldsymbol{v}_{\mathrm{R}}(t)$,

$\boldsymbol{r}_{\mathrm{T}}(t)$, $\boldsymbol{v}_{\mathrm{T}}(t)$, and their (systematic) uncertainties, $u_{\boldsymbol{r}\mathrm{R}}^s(t)$, $u_{\boldsymbol{v}\mathrm{R}}^s(t)$, $u_{\boldsymbol{r}\mathrm{T}}^s(t)$, $u_{\boldsymbol{v}\mathrm{T}}^s(t)$. For due limitation of depth of workflow detail in Figure 2 we do not separately show the propagation of the basic and apparent systematic uncertainties as they are both identically propagated through the operator chain shown for $u_{Lr,k}^s(t)$. All variables are provided on the time grid $t$ with





elements $t_i$, at $f_s = 50\,\mathrm{Hz}$ sampling rate, and for the two GPS carrier frequencies $f_{\mathrm{T}k}$, with $k \in \{1, 2\}$, $f_{\mathrm{T}1} = 1.57542\,\mathrm{GHz}$ and $f_{\mathrm{T}2} = 1.22760\,\mathrm{GHz}$.

We used excess phase state profiles $L_{\mathrm{r},k}(t)$ and the orbit state profiles $\boldsymbol{r}_{\mathrm{R}}(t)$, $\boldsymbol{v}_{\mathrm{R}}(t)$, $\boldsymbol{r}_{\mathrm{T}}(t)$, $\boldsymbol{v}_{\mathrm{T}}(t)$ from 15$^{\mathrm{th}}$ July 2008 as test-day ensemble. For CHAMP and COSMIC, orbit state and excess phase profiles were provided by the COSMIC Data Analysis and Archiving Center (CDAAC) of the University Corporation for Atmospheric Research (UCAR), Boulder, Colorado. The End-to-End GNSS Occultation Performance Simulation and Processing System (EGOPS) (Fritzer et al., 2009) was used for generating the simulated MetOp orbit state and excess phase profiles with realistic receiver noise (simMetOp). Figure 3 shows $L_{\mathrm{r},k}(t)$ in (a), $u^s_{Lr,k}(t)$ in (b), $u^r_{Lr,k}(t)$ in (c) and $\mathbf{R}_{Lr,k}$ in terms of representative correlation functions in (d) and (e), for a representative COSMIC RO event of the test-day ensemble from 15$^{\mathrm{th}}$ July 2008 (example case).

Exploiting the linearity of the (linearized) retrieval operators, the so-called *baseband-approach* (Kirchengast et al., 2016a) is applied throughout the rOPS. Hereby a zero-order model profile is subtracted from the input state profile and only the remaining delta-profile is processed through the operator. After application of the operator, the zero-order model profile of the output state profile is added back to the resulting delta-profile. This approach effectively avoids biases from numerical operations on (near-)exponentially varying RO profiles.

The model profiles used as zero-order states in the retrieval, i.e., $L_{\mathrm{m}}$, $D_{\mathrm{m}}$ and $\alpha_{\mathrm{m}}$ (cf. Table 1), were created from European Centre for Medium-Range Weather Forecasts (ECMWF) short-range (24h) forecast refractivity fields, accurately forward modeled to bending angle ($\alpha_{\mathrm{m}}$), Doppler shift ($D_{\mathrm{m}}$) and excess phase ($L_{\mathrm{m}}$) profiles, co-located to the latitude, longitude and time of the respective RO event processed in the rOPS. The ECMWF fields used have a horizontal resolution of about $300\,\mathrm{km}$ (triangular truncation T42)—which corresponds to the approximate horizontal resolution of RO profiles (e.g., Kursinski et al., 1997)—and are available at 91 vertical levels (L91).

ECMWF fields were chosen for their proven leading quality (Untch et al., 2006; Bauer et al., 2015) and thus high suitability for serving as zero-order state profiles; any other reasonable model profiles could be used as well since the retrieval results negligibly depend on the exactly chosen zero-order model profiles. For comparison we plotted $L_{\mathrm{m}}(t)$ for the COSMIC example case into Figure 3a, which demonstrates that the ECMWF short-range forecast lies very close to $L_{\mathrm{r}1}(t)$ and $L_{\mathrm{r}2}(t)$ and thus suits well as model profile.

While in the near future the random and systematic uncertainty profiles will be provided by the rOPS L1a processor (Innerkofler et al., 2016), they still had to be estimated from the excess phase profiles of UCAR/CDAAC, and simMetOp, respectively, for this study. To this end, each *estimated random uncertainty* profile $u^r_{Lr,k}(t)$ was estimated based on the noise of the respective retrieved excess phase profile $L_{\mathrm{r},k}(t)$. The noise was determined following the estimation scheme for bending angle observation errors described by Li et al. (2015), Section 2.2 therein; so we just briefly summarize how we used it here.

First, for both, the retrieved profile $L_{\mathrm{r},k}$ and for the model profile $L_{\mathrm{m}}$, the mean over all grid points between 60 and $70\,\mathrm{km}$ was determined. Then $L_{\mathrm{m}}$ was offset-corrected towards $L_{\mathrm{r},k}$ by subtracting the difference of these two means from $L_{\mathrm{m}}$, giving the offset-corrected model profile $L_{\tilde{\mathrm{m}}}$. Next, the delta-profile $\delta L_{\mathrm{r}\tilde{\mathrm{m}},k} = L_{\mathrm{r},k} - L_{\tilde{\mathrm{m}}}$ was calculated. After smoothing $\delta L_{\mathrm{r}\tilde{\mathrm{m}},k}$ with a $10\,\mathrm{km}$ moving average boxcar filter, the smoothed profile was subtracted from $\delta L_{\mathrm{r}\tilde{\mathrm{m}},k}$ again, to get $\delta\delta L_{\mathrm{r}\tilde{\mathrm{m}},k}$, the random




noise profile component of $L_{\mathrm{r},k}$ isolated in this way. Finally, the estimated random uncertainty was determined as

$$u^r_{L\mathrm{r},ik} = \sqrt{\sum_{j=i-M/2}^{i+M/2} \delta\delta L^2_{\mathrm{r}\tilde{\mathrm{m}},jk}} \;, \tag{5}$$

where $M$ is the number of grid points equivalent to a window width of $10\,\mathrm{km}$. To avoid boundary effects of the filter, $u^r_{L\mathrm{r},k}$ was only determined up to $z_{a\mathrm{Top}} - 5\,\mathrm{km}$, and down to $z_{a\mathrm{Gradr}}$ at $30\,\mathrm{km}$. It was constantly extended at the upper end and extended by a linear gradient below $z_{a\mathrm{Gradr}}$, using (in units [m])

$$u^r_{L\mathrm{r},ik} = u^r_{L\mathrm{r},k}(z_{a\mathrm{Gradr}}) + \frac{z_{a\mathrm{Gradr}} - z_{a,i}}{3 \cdot 10^6}, \tag{6}$$

for all elements of $u^r_{L\mathrm{r},k}(t)$ below $z_{a\mathrm{Gradr}}$, roughly following estimates of ESA/EUMETSAT (1998).

Since the noise components responsible for the random uncertainty at excess phase level are essentially uncorrelated at a sampling rate of $50\,\mathrm{Hz}$ (Syndergaard, 1999; Hajj et al., 2002), the correlation matrix $\mathbf{R}_{L\mathrm{r}}$ is set to unity in the diagonal and to zero outside (i.e., a Kronecker-$\delta$ assignment) for both channels. The elements of the covariance matrix $\mathbf{C}_{L\mathrm{r}}$ are hence (Item 1.1 in Figure 2),

$$C_{L\mathrm{r},ijk} = u^r_{L\mathrm{r},ik} \cdot u^r_{L\mathrm{r},jk} \cdot R_{L\mathrm{r},ijk} = u^r_{L\mathrm{r},ik} \cdot u^r_{L\mathrm{r},jk} \cdot \delta_{ij}. \tag{7}$$

For the MC validation of the CP, error profile realizations $\epsilon^{\mathrm{r}}_{L\mathrm{r}}$ were superimposed onto simulated 'true' excess phase profiles $L^{\mathrm{T}}_{\mathrm{r},k}(t)$. As source for $L^{\mathrm{T}}_{\mathrm{r},k}$ we used an EGOPS-simulated 'error-free' CHAMP event from August 8th, 2008 (i.e., no receiver system errors superimposed).

To create the error profiles, a representative $u^{r,\mathrm{STD}}_{L\mathrm{r}}$ uncertainty profile was selected from a COSMIC ensemble of uncertainty profiles, created according to Equations 5 and 6. The error profile realizations are random draws from a distribution characterized by these uncertainties, again assuming that $R_{L\mathrm{r},ij} = \delta_{ij}$, i.e., that there are no correlations between the individual grid levels (Item (a) in Figure 2; Figure 3f). The same standard profile $u^{r,\mathrm{STD}}_{L\mathrm{r}}$ was used as input for the CP to which the MC results are then compared. This MC validation method applied to test the rOPS L1b uncertainty propagation steps is essentially the same as in SKS2017, and described therein in more detail.

The *estimated systematic uncertainty* $u^s_{L\mathrm{r},k}$ was determined based on a simple model roughly following error estimates from ESA/EUMETSAT (1998), with constant uncertainty from $80\,\mathrm{km}$ down to $z_{a\mathrm{Grads}}$ at $8\,\mathrm{km}$, and a linear uncertainty gradient in the troposphere. The constant $u^s_{L\mathrm{r},k}$ above $z_{a\mathrm{Grads}}$ is $0.1\,\mathrm{mm}$ for $k=1$ and $0.2\,\mathrm{mm}$ for $k=2$ for simMetOp. This uncertainty is interpreted as an estimated basic systematic uncertainty, i.e., as a lower-bound estimate of available accuracy. For CHAMP and COSMIC $u^s_{L\mathrm{r},1} = 0.2\,\mathrm{mm}$ and $u^s_{L\mathrm{r},2} = 0.4\,\mathrm{mm}$. From $z_{a\mathrm{Grads}}$ downwards, $u^s_{L\mathrm{r},k}$ (in units [m]) increases by

$$u^s_{L\mathrm{r},ik} = u^s_{L\mathrm{r},k}(z_{a\mathrm{Grads}}) + \frac{z_{a\mathrm{Grads}} - z_{a,i}}{3 \cdot 10^7}. \tag{8}$$

In order to avoid a sharp kink in the $u^r_{L\mathrm{r},k}$ profiles at $z_{a\mathrm{Gradr}}$, and in the $u^s_{L\mathrm{r},k}$ profiles at $z_{a\mathrm{Grads}}$, a $2\,\mathrm{km}$-width moving average boxcar filter was applied to smooth these simple uncertainty models around these transition altitudes.





The orbit position and velocity uncertainties of the transmitter and the receiver satellites show little variation within the short duration of an individual RO event of about 45 sec to 2 min (Innerkofler et al., 2016) and can be assumed to be constant biases. They are thus counted to the systematic uncertainties, more precisely the apparent systematic uncertainties, since the actual values of the orbit-borne biases will generally change in a pseudo-random manner from event to event.

We set the transmitter position and velocity uncertainties to $u_{\boldsymbol{r}\mathrm{T}}^s = 3\,\mathrm{cm}$ and $u_{\boldsymbol{v}\mathrm{T}}^s = 0.01\,\mathrm{mms}^{-1}$, consistent with accuracies for GPS orbits available from GNSS orbit providers like the International GNSS Service (IGS). The receiver position and velocity uncertainties, $u_{\boldsymbol{r}\mathrm{R}}^s = 5\,\mathrm{cm}$ and $u_{\boldsymbol{v}\mathrm{R}}^s = 0.05\,\mathrm{mms}^{-1}$ for CHAMP and MetOp, are adopted four times smaller than those for COSMIC with $u_{\boldsymbol{r}\mathrm{R}}^s = 20\,\mathrm{cm}$ and $u_{\boldsymbol{v}\mathrm{R}}^s = 0.2\,\mathrm{mms}^{-1}$, as found by ongoing rOPS-related POD studies (Innerkofler et al., 2016), consistent with previous literature (e.g., Montenbruck et al., 2009; Schreiner et al., 2009).

## 10 3 Algorithm Sequence and Example Results

In this Section the L1b uncertainty propagation algorithm sequence is introduced. We illustrate the effects of the algorithm on the main uncertainty variables by way of the COSMIC example case already used for Figure 3.

For each L1b retrieval step, i.e., segments (1), (2), and (3) in Figure 2, the results for the principal variables are shown in Figures 4 to 8. These variables are the state profiles $X_\mathrm{r}$ (with $X_\mathrm{r} \in \{L_{\mathrm{F},k}(t), D_{\mathrm{r},k}(t), \alpha_{\mathrm{G},k}(t), \alpha_{\mathrm{G},k}(z_a), \alpha_{\mathrm{M},k}(z_a), \alpha_{\mathrm{F},k}(z_a), \alpha_\mathrm{r}(z_a)\}$),
the estimated systematic uncertainty profiles $u_{X\mathrm{r}}^s$, the estimated random uncertainty profiles $u_{X\mathrm{r}}^r$, representative correlation functions $R_{X\mathrm{r},i}$ (with $i$ such that $z_{a,i} \in \{10\,\mathrm{km}, 30\,\mathrm{km}, 50\,\mathrm{km}, 70\,\mathrm{km}\}$), and the correlation length profiles $l_{X\mathrm{r}}$ and resolution profiles $w_{X\mathrm{r}}$. Along with the dual-frequency state profiles we also show the collocated forward modeled short-range forecast profiles, i.e., model profiles $X_\mathrm{m}$ with $X_\mathrm{m} \in \{L_\mathrm{m}, D_\mathrm{m}, \alpha_\mathrm{m}\}$ for comparison.

A concise definition of the variables involved is provided in Table 1, as introduced above. The summary description in
this section is complemented by a complete step-by-step description of the algorithm along the entire L1b retrieval chain in Appendix A, which is organized for convenience into the same sequence of subsections.

To simplify the notation in the description we suppress index $k$ whenever steps are applied in an identical way to the data of both GNSS L-band channels with frequencies $f_{\mathrm{T1}}$ and $f_{\mathrm{T2}}$. Only if the two channels are treated differently, such as in Section 3.3, the index is considered again. For conciseness we also do not illustrate both the estimated basic and estimated
apparent systematic uncertainty but rather the total estimated systematic uncertainty as the overall result.

### 3.1 Doppler Shift Retrieval

### 3.1.1 Basic Lowpass Filtering

A Blackman-Windowed-Sinc (BWS) lowpass filter with a filter cutoff frequency $f_\mathrm{c} = 2.5\,\mathrm{Hz}$ (boxcar-equivalent filter width of $0.2\,\mathrm{s}$) (Item 1.2 in Figure 2) is applied onto the excess phase profile $L_\mathrm{r}(t)$, before the Doppler differentiation (Item 1.4
in Figure 2), to avoid an amplification of high-frequency noise in the phase profile by the derivative operation. This filter suppresses the noise and consequentially the filtered excess phase profile $L_\mathrm{F}(t)$, shown in Figure 4a, is expected to have



random uncertainties $u^r_{LF}$ of smaller magnitude, but correlated over the length of the filter-window. The uncertainties obtained through the implemented algorithm confirm these expectations, i.e., random uncertainty profiles in Figure 4c are less than a third in magnitude of those in Figure 3c, and Figures 4d-e show how the correlation functions widened and the correlation length/vertical resolution reaches $\sim 0.5\,\mathrm{km}/\sim 0.6\,\mathrm{km}$ above about 30 km impact altitude (Figure 4f).

The random uncertainty propagation algorithm, i.e., the covariance propagation from $\mathbf{C}_{Lr}$ to $\mathbf{C}_{LF}$ is described by Equation A6 and Item 1.3 in Figure 2, and justified by Equation 2. To obtain $u^r_{LF}$ and $\mathbf{R}_{LF}$, we use Equations A7 and A8.

To propagate the estimated systematic uncertainty $u^s_{Lr}$, which characterizes long-range-correlated offsets or biases, we use the same BWS filter as for the state profile, i.e., making use of Equation 4. Because the input uncertainty profile $u^s_{Lr}$ is chosen to be constant down to $z_{a\mathrm{Grads}}$, the filter has little effect, and $u^s_{LF}$, shown in Figure 4b, is essentially equal to $u^s_{Lr}$, shown in Figure 3b.

The resolution profile $w_{Lr}$ is determined by the filter-width according to Equations A11 and A13. After the BWS filtering, the resolution is roughly equal to the correlation length $l_{LF}$, amounting to $\sim 0.6\,\mathrm{km}$ above about 30 km impact altitude and becoming finer downwards due to the increasing refraction (Figure 4f).

### 3.1.2 Doppler Shift Derivation

The next step is a five-point differentiation operation (Item 1.4 in Figure 2) used to calculate the Doppler shift profile $D_r(t)$ from the filtered excess phase profile $L_F(t)$. The resulting dual-frequency Doppler shift profiles are plotted along with the model profile $D_m(t)$ in Figure 5a for the example case.

As for the filtered excess phase, we apply CP (Equation A18, Item 1.5 in Figure 2) to first calculate the covariance matrix $\mathbf{C}_{Dr}$ and then extract $u^r_{Dr}$ (shown in Figure 5c) and $\mathbf{R}_{Dr}$ (Figures 5d and 5e) from it. The choice of the x-axis range shows the random uncertianties increased, but the differentiation actually does increase relative random uncertainties (relative to the state profile). It also causes anti-correlation with neighbouring elements, as visualized by the negative side-peaks of the correlation functions in Figures 5d and 5e. The correlation length $l_{Dr}$ (of the main correlation function peak) decreases accordingly (now smaller than 0.3 km throughout), because the correlation functions fall off steeper on both sides of the main peak (Figure 5f).

For calculating the estimated systematic uncertainty we use the state operator, i.e., we just differentiate $u^s_{LF}$ and get $u^s_{Dr}$ (shown in Figure 5b). With the current illustrative choice of input uncertianties the systematic uncertainty of the Doppler shift profile is zero above the transition to the troposphere, where the estimated systematic uncertainty of the excess phase is assumed constant; in the troposphere a Doppler shift offset of $\sim 0.02\,\mathrm{mm\,s^{-1}}$ occurs.

The resolution profile $w_{Dr}$ shows that the vertical resolution stays unaffected by this operator (cf. Figures 5f and 4f), because the BWS filter width of the preceding lowpass filtering (intentionally) stretched beyond the five neighboring points involved in the differentiation.





## 3.2 Bending Angle Retrieval

### 3.2.1 GO Bending Angle Retrieval

The next operator is the geometric-optics (GO) bending angle retrieval in which retrieved GO bending angle profiles $\alpha_{\mathrm{G}}(t)$ are calculated from Doppler shift profiles $D_{\mathrm{r}}(t)$ and the orbit position and velocity vectors $\boldsymbol{r}_{\mathrm{T}}(t)$, $\boldsymbol{r}_{\mathrm{R}}(t)$, $\boldsymbol{v}_{\mathrm{T}}(t)$, $\boldsymbol{v}_{\mathrm{R}}(t)$ (Item 2.1

in Figure 2) and then interpolated to the (common monotonic) impact altitude grid $z_a$ (Item 2.6 in Figure 2).

Figure 6a shows retrieved $\alpha_{\mathrm{G}}$ profiles together with the model profile $\alpha_{\mathrm{m}}$. The mildly non-linear, implicit-type bending angle retrieval operator needs to be solved iteratively, and requires linearization for both random and systematic uncertainty propagation, as described in detail in Appendix A (Section A2). Because this retrieval step is performed level by level, keeping levels independent, the GO bending angle retrieval leaves correlation functions and resolution unchanged (cf. Figures 6d-f and

5d-f).

The estimated random uncertainties $u^r_{\alpha\mathrm{G}}$, as shown in Figure 6c, now increase more strongly in the lower stratosphere and troposphere (to about $40$ to $50\,\mu\mathrm{rad}$ near $10\,\mathrm{km}$), because they are depending on the vertical gradient of the impact parameter $a_t$, which is increasingly larger towards lower altitudes from the increasing refraction.

The main contributions to the estimated systematic uncertainty $u^s_{\alpha\mathrm{G}}$ are induced by systematic uncertainties in orbit velocity

and position of the transmitter and the receiver satellite (details in Section A2), which in total amount to about $0.05\,\mu\mathrm{rad}$ (Figure 6b).

### 3.2.2 WO Bending Angle Retrieval

Due to strong refractivity gradients and multipath effects, the GO bending angle retrieval can be inadequate in the troposphere, and therefore wave-optics (WO) algorithms are applied to reconstruct the geometric optical ray structure of the wave field (e.g.,

Gorbunov, 2002a; Gorbunov and Lauritsen, 2004).

In the rOPS, along with the WO bending angle profile $\alpha_{\mathrm{W}}(z_a)$, the systematic uncertainty profile $u^s_{\alpha\mathrm{W}}$, the random uncertainty profile $u^r_{\alpha\mathrm{W}}$, the correlation matrix $R_{\alpha\mathrm{W}}$, and the resolution profile $w_{\alpha\mathrm{W}}$ are retrieved (Item 2.7 in Figure 2).

The WO bending angle retrieval algorithm used is a canonical transform (CT2) algorithm (Gorbunov et al., 2004) and the associated uncertainty propagation algorithm is not described here, but separately by Gorbunov and Kirchengast (2015, 2017).

The WO retrieval and uncertainty propagation results are supplied up to $20\,\mathrm{km}$ impact altitude by the WO algorithms.

### 3.2.3 Merging of GO and WO Bending Angle Profiles

In the rOPS bending angle retrieval the results from the WO retrieval, $\alpha_{\mathrm{W}}$, are merged with GO retrieval results, $\alpha_{\mathrm{G}}$, at around a transition altitude $z_a^{\mathrm{GW}}$ in a transition range $z_a^{\mathrm{GW}} \pm \Delta z_a^{\mathrm{GW}}$, to get merged profiles $\alpha_{\mathrm{M}}$ (Item 2.9 in Figure 2). The determination of the transition altitude and the merging algorithm are described in Appendix A (Section A2.3). We use a

specialized covariance propagation to propagate the GO and WO uncertainties, expressed by the covariance matrices $\mathbf{C}_{\alpha\mathrm{G}}$





and $\mathbf{C}_{\alpha W}$, to properly obtain the covariance matrix of the merged bending angle $\mathbf{C}_{\alpha M}$ (Equations A37 and A38, Item 2.10 in Figure 2).

Because the rOPS implementation of the WO uncertainty propagation is towards completion currently but not yet finished (Gorbunov and Kirchengast, 2017), all examples in this study are GO-only, i.e., only the GO retrieval is performed. Results for
$\alpha_M$ are thus unchanged from those shown in Figure 6 and not separately illustrated.

In order to nevertheless test and validate the uncertainty propagation of the merging algorithm, WO retrieval results were artificially substituted by GO retrieval results (and consequently random uncertainties were assumed to be correlated rather than uncorrelated) for the MC validation (Section 4).

### 3.3 Atmospheric Bending Angle Derivation

**3.3.1 Adaptive Lowpass Filtering and Minor Channel Extrapolation**

To prepare the merged bending angle profiles $\alpha_{M,k}$ for the ionospheric correction they are first filtered by another BWS filter operation (Item 3.1 in Figure 2) in order to ensure adequately smoothed bending angle profiles $\alpha_{F,k}$, with $k \in \{1,2\}$. The chosen filter cutoff-frequency for $k=1$ is $f_{c1} = 2.5\,\mathrm{Hz}$, same as the basic filtering (Section 3.1.1) just to ensure clearness of any residual operator noise since excess phase filtering, while $f_{c2}$ is set noise-dependent, between $2.5$ and $0.5\,\mathrm{Hz}$ (boxcar
equivalent width of $0.2$ to $1.0\,\mathrm{s}$). In events in which the $\alpha_{F2}$ profile does not reach down as far as $\alpha_{F1}$, it is extrapolated down to the bottom of $\alpha_{F1}$, $z_{aBot}$. The results for the filtered bending angle state profiles $\alpha_{F,k}$ are displayed in Figure 7a, together with the associated model bending angle profile $\alpha_m$. The filter has considerably reduced the noise of the profile, particularly for $\alpha_{F2}$, where a cutoff frequency $f_{c2} = 10/7\,\mathrm{Hz}$ appears to have been selected in this example case.

The relevant covariance-propagated random uncertainties $u^r_{\alpha F,k}$ are shown in Figure 7c (blue and red), illustrating the
reduced noise, especially for $\alpha_{F2}$. In return, the peaks of the correlation functions broaden (cf. Figure 7d-e and 6d-e), with correlation lengths $l_{\alpha F,k}$ at near $0.4\,\mathrm{km}$ for $\alpha_{F1}$ and above $0.5\,\mathrm{km}$ for $\alpha_{F2}$ (Figure 7f).

The estimated systematic uncertainty remains largely unchanged (Figure 7b) due to its smooth character.

The resolution of the filtered bending angle profiles (according to Equations A11 and A13) is determined by the cutoff-frequencies $f_{c,k}$ of the BWS filters. In the example case it is therefore essentially unchanged for $\alpha_{F1}$, while significantly
decreased for $\alpha_{F2}$ (cf. Figure 7f and 6f) since $f_{c2} = 10/7\,\mathrm{Hz}$. That is, the resolution $w_{\alpha F2}$ in the upper stratosphere for example, where the vertical scanning velocity of this RO event is about $3.2\,\mathrm{km\,s^{-1}}$, is near $1.1\,\mathrm{km}$ (Figure 7f).

**3.3.2 Ionospheric Correction**

The final step of the L1b processor is the ionospheric correction (Item 3.5 in Figure 2). The atmospheric bending angle $\alpha_r$ is obtained by applying a linear dual-frequency combination of $\alpha_{F1}$ and $\alpha_{F2}$, such that ionospheric effects are largely removed
(details are described in Section A3). The final retrieved atmospheric bending angle $\alpha_r$ of the example case is shown in Figure 8a. The propagation results for the estimated random uncertainty are shown in Figure 8c. The linear combination of the ionospheric correction amplifies noise and $u^r_{\alpha r}$ is therefore considerably larger than $u^r_{\alpha F1}$ and $u^r_{\alpha F2}$ (cf. Figure 8c and 7c).





Figure 8d shows how the correlation functions—as obtained through covariance propagation—are combining the characteristics of the correlation functions from the two matrices $R_{\alpha F1}$ and $R_{\alpha F2}$, with essentially inheriting the $\alpha_{F1}$ behavior, since the $\alpha_{F2}$ influence into the ionospheric correction is comparatively minor (see Section A3).

The residual higher order ionospheric effects are accounted for by a 'conservative best-guess' value ($0.05\,\mu$rad, reflecting results of Liu et al. (2015) and Danzer et al. (2013, 2015)) and added (in root-mean-square form) to the systematic uncertainty profile $u_{\alpha r}^s$, leading to a total estimated systematic uncertainty in this example case of $\sim 0.07\,\mu$rad (Figure 8b). Within this uncertainty, the one dominating component from orbit uncertainties ($\sim 0.05\,\mu$rad, cf. Figure 6b) can be considered an apparent systematic uncertainty that will essentially average out in ensemble-averaging (e.g., climatologies) while the other dominating component from residual higher-order ionospheric biases (also estimated $\sim 0.05\,\mu$rad as noted above) can be considered a basic systematic uncertainty. For the latter it is therefore useful and prepared for in the rOPS—in line with GUM recommendations and as discussed in the introductory Section 1—to correct for the quantifiable part of it in the future so that the total basic systematic uncertainty may be mitigated down to the $\sim 0.01\,\mu$rad level.

The resolution profile $w_{\alpha r}$ of the retrieved bending angle (Figure 8f) is dominated by the contribution of $\alpha_{F1}$ that strongly dominates (intentionally by construction) the ionospheric correction results in terms of the small-scale bending angle variability. Similar as for the correlation length profile $l_{\alpha r}$ it is therefore very close to $w_{\alpha F1}$ and only slightly larger.

## 4 Algorithm Validation

The GUM advises to use a Monte-Carlo (MC) method for uncertainty propagation if the retrieval operators do not fulfill the criteria for a GUM-type CP. In our case the MC method is put to another beneficial use, to validate the results of the CP, as recommended by JCGM (2011).

For the validation of the covariance propagation by the MC method, we sampled the input excess phase profile random error distribution, statistically described by

$$\mathbf{C}_{Lr}^{MC} = u_{Lr,i}^{r,STD} \cdot u_{Lr,j}^{r,STD} \cdot \delta_{ij}, \tag{9}$$

by a large number $M$ of draws $L_r^T + \epsilon_{Lr,j}^r$ (with $j \in \{1,...,M\}$ and $M = 1000$). For each of these $M$ profile realizations, the state retrieval is run through the L1b retrieval chain, to give $M$ realizations of the output variable $X_j$ (with $X_j \in \{L_{F,kj}(t), D_{r,kj}(t), \alpha_{G,kj}(z_a), \alpha_{F,kj}(z_a), \alpha_{r,j}(z_a)\}$ and $k \in \{1,2\}$). From these individual realizations the mean profiles $X^{MC}$, and the covariance matrices $\mathbf{C}_X^{MC}$,

$$\mathbf{C}_X^{MC} = \frac{1}{M-1}[(X_1 - X^{MC})(X_1 - X^{MC})^T + ... + (X_M - X^{MC})(X_M - X^{MC})^T)], \tag{10}$$

are calculated (Items b-g in Figure 2) and compared to the CP-propagated covariance matrices. In order to be able to attribute potential changes between CP and MC covariance matrices better, we decompose $\mathbf{C}_X$ into $u_X^r$ and $\mathbf{R}_X$ (Equations A7 and A8), and compare them separately.

Figure 9 shows the different steps along the retrieval chain from $L_{F,k}(t)$ to $D_{r,k}(t)$, $\alpha_{G,k}(z_a)$, $\alpha_{F,k}(z_a)$, and $\alpha_r(z_a)$ in the rows, for $k = 1$ (GPS $f_{T1}$ frequency) in the left column and for $k = 2$ (GPS $f_{T2}$ frequency) in the middle column. The right





column shows multiple representative correlation functions, from near $10\,\mathrm{km}$ to near $70\,\mathrm{km}$. Due to the limited number of MC draws, the MC results (black lines) show some jitter both in the estimated random uncertainty and in the correlation functions. Since the purpose of the MC results is only to demonstrate the correctness of the CP result, we can disregard this behavior.

Figures 9a (light blue) and 9b (orange) show the random uncertainties $u^r_{Lr,1}$ and $u^r_{Lr,2}$ respectively, which characterize the input distribution and from which the random error profiles $\epsilon^r_{Lr,j}$ are drawn. They also show the CP results for the random uncertainty $u^r_{LF1}$ (dark blue in Figure 9a) and $u^r_{LF2}$ (red in Figure 9b), compared to the MC propagated random uncertainties (black).

The CP and MC lines match very well and show that the implemented CP algorithm delivers correct results for the basic filtering step. For $f_{T2}$, the MC uncertainties do not reach down as far as the CP uncertainties, because the shortest of all draws of the large ensemble of size $M$ determines how far down the recombined MC covariance matrix reaches. Figure 9c compares CP correlation functions $R_{LF,i1}$ (blue) and $R_{LF,i2}$ (red) to the corresponding MC correlation functions (black dashed).

Also the CP and MC correlation functions agree well. Both capture the narrow peak, broadened by the BWS filter. Again the MC correlation functions fluctuate around zero left and right of the peak, from the finite ensemble size, but it is obvious that the CP delivers the correct off-peak results (i.e., zero). The MC validation (black) of $u^r_{Dr1}$ (Figure 9d), $u^r_{Dr2}$ (Figure 9e) and $R_{Dr,i}$ (Figure 9f, blue and red) demonstrates that the CP through the Doppler shift derivation performs correctly as well.

The next row, Figures 9g to 9i show the results for the GO bending angle $\alpha_G(z_a)$, i.e., after the interpolation of all quantities to the (common monotonic) impact altitude grid $z_a$. For comparison, in Figures 9a to 9f, all quantities were provided on the common time grid ('setting time' relative to time zero at $80\,\mathrm{km}$ altitude) with $50\,\mathrm{Hz}$ sampling rate (the corresponding impact altitude of the 'true' profile $L^T_r$ is provided for orientation on the RHS axis). In Figures 9g to 9o all profiles have already been interpolated to the $z_a$ grid (and the corresponding setting time for the 'true' profile is provided for orientation on the RHS in these cases).

The results for the filtered bending angle $\alpha_F$ follow in Figures 9j to 9l. Also here the MC results match the CP result well. Due to the lower BWS cutoff-frequency for $\alpha_{F2}$, now $u^r_{\alpha F2}$ is smaller than $u^r_{\alpha F1}$, even though $u^r_{\alpha G2}$ was larger than $u^r_{\alpha G1}$. Correspondingly the peak of the correlation functions $R_{\alpha F,i2}$ widened more than those of $R_{\alpha F,i1}$ (cf. Figure 9l and 9i).

Finally, Figures 9m to 9o show the CP results for retrieved atmospheric bending angle $\alpha_r$, where Figure 9n is included as a special cross-comparision in case only variance propagation would be used instead of CP. Figures 9m and 9o confirm that CP results are also correct for this final L1b variable, both in terms of random uncertainty and correlation functions.

In order to demonstrate that a full CP is necessary to propagate random uncertainties correctly, we also calculated random uncertainties $u^r_{\alpha r}$ based on mere variance propagation (VP) from $\alpha_G$ to $\alpha_r$ for comparison. A description of this VP algorithm (i.e., only diagonal elements of the covariance matrices are considered) is provided in Appendix B. Figure 9n clearly shows that VP would overestimate random uncertainties in $\alpha_r$ considerably, pointing to the importance of the complete CP implementation in the L1b retrieval chain, even though the correlation lengths involved in the processing steps are rather small.





## 5 Performance Demonstration

To statistically evaluate the performance of the new L1b uncertainty propagation algorithm, we also processed a complete test-day of real (CHAMP, COSMIC) and simulated (simMetOp) data from GNSS RO satellite missions. Figure 10 shows the results for estimated systematic and random uncertainty profiles, as well as correlation length and resolution profiles for filtered

excess phase profiles $L_{\mathrm{F},1}$. Figure 11 subsequently illustrates the ensemble mean of the same variables for $L_{\mathrm{F}1}$, $D_{\mathrm{r}1}$, $\alpha_{\mathrm{G}1}$ and $\alpha_{\mathrm{r}}$ for the test-day ensemble. In Figure 10 we also co-illustrate the number of events processed for each of the RO missions (middle column).

About $5\%$ of the total number of processed profiles for each mission have been discarded, because they were detected as outliers (these are not included in the number of profiles shown). All profiles are shown as function of impact altitude, because

each of the profiles in the ensembles needed to be interpolated to the same (standard) impact altitude grid, to orderly calculate their mean profiles.

Figure 10a shows $u^r_{LF1}$ and $u^s_{LF1}$ for all $\sim 100$ CHAMP events. It is visible (also in Figures 10d and 10g) that the random uncertainty is estimated based on excess phase noise between 30 and $75\,\mathrm{km}$ and synthetically extended above and below, as described in Section 2.2. For the large majority of events, $u^r_{LF1}$ lies between about 0.5 and $3\,\mathrm{mm}$ in the range between $30\,\mathrm{km}$

and $75\,\mathrm{km}$. Note that these results show the random uncertainties after the application of the basic BWS filter (Section 3.1.1), but the input uncertainties $u^r_{Lr1}$ are of similar shape (though larger in magnitude).

Figure 10b shows that the correlation length profiles of the CHAMP ensemble (gray) and its ensemble mean (yellow) are of relatively constant magnitude from 35 to $80\,\mathrm{km}$, but then get smaller downward, because the RO event's scan velocity decreases (see Equation A13). Since the BWS filter determines the vertical resolution and the correlation length at the same time, the

resolution profiles $w_{LF1}$ (Figure 10c) are quite similar to the correlation length profiles $l_{LF1}$ (Figure 10b).

The number-of-events profile shows that most CHAMP events end between 5 and $12\,\mathrm{km}$ (Figure 10b, black). This is because the GO profiles illustrated here are cut off right at the lower end of the GO-WO transition range at $z^{\mathrm{GW}}_a - \Delta z^{\mathrm{GW}}_a$ (cf. Table 2).

Compared to CHAMP, the mean random uncertainty $u^r_{LF1}$ (Figure 10d) for the $\sim 1500$ events of the COSMIC ensemble is smaller, particularly above $30\,\mathrm{km}$, indicating the improved data quality of this later mission. The mean of the correlation length

profiles $l_{LF1}$ (Figure 10e) is higher than for CHAMP (Figure 10b) and correspondingly the resolution of the COSMIC profiles also somewhat coarser (Figure 10f and 10c). The cutoff-frequency and sampling rate—and thus the resolution in time—is set to be the same in the rOPS, irrespective of the missions; these differences hence are due to the different vertical scan velocities of the missions induced by the differences in orbit altitudes (CHAMP $\sim 400\,\mathrm{km}$, COSMIC $\sim 700\,\mathrm{km}$).

For simMetOp, $u^r_{LF1}$ is even smaller than for COSMIC and from 35 to $80\,\mathrm{km}$ the mean random uncertainty profile stays

below $1\,\mathrm{mm}$ (Figure 10g). On average the correlation length/resolution profile of the $\sim 700$ ensemble members is larger/coarser than for CHAMP and COSMIC, due to an even somewhat higher scan velocity of the MetOp satellite ($\sim 820\,\mathrm{km}$ orbit altitude).

The systematic uncertainty $u^s_{LF1}$, just co-illustrated for completeness in Figures 10a, d, g, is almost left unchanged by the BWS filter and is essentially equal to the preset input uncertainty for all three missions (Section 2.2).





Figure 11 shows how the $u_{LF1}^s$, $u_{LF1}^r$, $l_{LF1}$ and $w_{LF1}$ profiles are on average affected by the uncertainty propagation. The color code for the different satellite missions is the same as in Figure 10. The propagation effects visible are similar to those already seen in Figures 3 to 8. The Doppler shift derivation increases the relative uncertainties and reduces correlation length (of the main peak), while the resolution stays the same (Figure 11d-f). The GO bending angle retrieval leaves correlation length

and resolution unchanged, while random uncertainties increase strongly in the lower stratosphere and troposphere due to the increasing refractive effects (Figure 11g-i).

Finally, the BWS filtering before the ionospheric correction decreases random uncertainties and increases correlation length, and resolution somewhat. However, the linear combination of the two bending angle profiles $\alpha_{F1}$ and $\alpha_{F2}$ then increases the random uncertainty again (cf. Figures 11j and 11g). The adaptive minor channel cutoff-frequency $f_{c2}$ for the relatively noisy

CHAMP profiles is generally lower than for the other two missions, and the filter effect is therefore stronger for CHAMP (indicated by the larger $l_{\alpha r}$ in Figure 11k)

The estimated systematic uncertainty of the atmospheric bending angle $u_{\alpha r}^s$, indicated for completeness in Figure 11 (left column, enflated by a factor of 10 in (a) and 100 in (d, g, j) for somewhat better visibility), stays below $0.1\,\mu$rad for all three missions.

**6   Conclusions**

In order to deliver climate benchmark datasets it is essential to integrate uncertainty propagation in RO retrievals. In this study we presented the uncertainty propagation algorithm chain from excess phase profiles to atmospheric bending angle profiles (L1b processing), as newly implemented in the rOPS at the WEGC. Along with the basic profiles retrieval, we provide estimates for systematic and random uncertainties, error correlation matrices and vertical resolution profiles, which is unique

amongst all existing RO processing systems so far (Ho et al., 2012; Steiner et al., 2013).

We validated the implemented algorithm via comparison to Monte-Carlo sample propagation results and demonstrated the performance of the algorithm using real data ensembles. We find close agreement between the implemented covariance propagation of random uncertainties and the Monte-Carlo validation runs, verifying the correctness of the implemented algorithm. The test-day ensembles for three different missions (CHAMP, COSMIC, simMetOp) show reliable, robust and consistent

results that provide valuable insight and understanding of retrieval chain details.

Together with the integration of the uncertainty propagation algorithm from atmospheric bending angle profiles to dry-air profiles (L2a processing) presented by Schwarz et al. (2017), the rOPS can now provide estimates of systematic and random uncertainty profiles, of error correlation matrices and resolution, and of observation-to-background weighting ratio profiles from excess phase to dry-air profiles.

The next step towards the final atmospheric profiles, currently ongoing, is the introduction of integrated uncertainty propagation for the moist-air retrieval (L2b processing). Implementation of uncertainty propagation for the wave-optics bending angle retrieval and for the orbit determination and excess phase processing (L1a processing) is on-going as well.





Once completed, the full rOPS retrieval chain will run with integrated uncertainty estimation, a major step towards climate benchmark data provision, and beneficial for the wide diversity of uses in atmospheric and climate science and applications.

## Appendix A: Algorithm Description

In this Appendix the rOPS L1b uncertainty propagation algorithm is introduced, following the L1b retrieval chain (Figure 2; Section 3) step by step, starting with excess phase profile $L_r$ as input and proceeding to $L_F$, $D_r$, $\alpha_G$, $\alpha_M$, $\alpha_F$ and finally $\alpha_r$. The relevant variable definitions and symbol explanations are summarized in Tables 1 and 2. A fully detailed algorithmic description is provided by Kirchengast et al. (2017b).

If not stated otherwise, elements of the vector-type vertical profiles are addressed using subscript $i$ (with $i \in \{1, 2, ..., N\}$), and optionally $j$ (with $j \in \{1, 2, ..., N\}$), running from top downward towards the bottom of the profile, where $N$ is the number of vertical grid levels. Until the interpolation of all quantities to the common monotonic impact altitude grid $z_a$, all quantities are provided on an equidistant $50\,\mathrm{Hz}$ time grid $t$ with grid-points $t_i$.

All steps in Sections A1 and A2 are applied to each of the GNSS transmitter channels' carrier frequencies $f_{Tk}$, as also indicated by the index $k$ in Figure 2. In the notation of these sections we therefore suppress the index $k$ for the convenience of simplified readability. Also for conciseness we write the estimated systematic uncertainty equations only for the total systematic uncertainties $u^s$ and briefly address the type of the relevant components (whether basic systematic uncertainty $u^b$ or apparent systematic uncertainty $u^a$) in the surrounding text.

### A1 Doppler Shift Retrieval

#### A1.1 Basic Lowpass Filtering

The Doppler differentiation (Item 1.4 in Figure 2) would potentially amplify high-frequency noise in the excess phase profiles. To avoid this amplification, a Blackman-Windowed-Sinc (BWS) lowpass filter (e.g., Smith, 1999) is applied onto the excess phase profiles first (Item 1.2 in Figure 2).

For this basic filtering the relative cutoff-frequency $f_c/f_s$ is set to $0.05$, equivalent to $f_c = 2.5\,\mathrm{Hz}$, 21 grid points, or a cutoff-period $\tau_c = 1/f_c = 0.4\,\mathrm{s}$, for the standard sampling rate $f_s$ of $50\,\mathrm{Hz}$ used for all RO profiles in the L1b processor of the rOPS. The corresponding sample width of the Blackman window $\tilde{M}$ (with samples $m \in \{0, ..., M\}$) is set to $\tilde{M} = 2 \cdot f_s/f_c$, yielding 41 grid points. This ensures a reliable filter performance, also allowing to robustly quantify the vertical resolution of the filtered data.

With such a design, the BWS lowpass filter combines efficient removal of high frequency noise with a narrow smoothing window. The BWS filter thus achieves a better smoothing effect, while keeping a higher resolution $w_{LF}$ than a simple moving average Boxcar (BC) filter. Based on a time segment of a few seconds of the excess phase delta-profile of the COSMIC example event (also used for Figures 3 to 8), Figure A1 illustrates how the BWS filter compares to Boxcar filters of 11 and 21 grid points. The corresponding filter functions are displayed in Figure A1a, while Figure A1b compares the filter results.




It is clearly seen that the smoothing window width of the BWS filter best corresponds to an 11 point boxcar filter (confirmed nummerically by minimization of the sum of squared differences between Boxcar and BWS filter result) while giving considerably better filtering results (as for example visible between $31.5\,\mathrm{s}$ and $32.0\,\mathrm{s}$, where the 11 points Boxcar filter zigzags around the BWS result). The effective filter width of the BWS filter, which we also term 'boxcar-equivalent width', is therefore its full

width at half maximum (see Figure A1a), corresponding to $\tilde{M}/4+1$ samples with our design.

The *actually used* sample width $M$ of the BWS filter is equal to $\tilde{M}$, except that it decreases at the top and bottom of the profile such that it does not reach beyond the first/last element of the vector to be filtered. At the $i^{\text{th}}$ grid point (with $i \in \{1,2,...,N\}$, and $N$ being the profile length in grid points), the filterwidth $M$ is thus

$$
M = \begin{cases} \tilde{M} & \text{for } \tilde{M}/2 < i < N - \tilde{M}/2 \\ 2i - 1 & \text{for } 1 \le i \le \tilde{M}/2 \\ 2(N-i)+1 & \text{for } N - \tilde{M}/2 < i < N \end{cases} . \tag{A1}
$$

The *state* profile of the filtered phase $L_\mathrm{F}$ is obtained using the 'baseband approach' (Kirchengast et al., 2016a), i.e., by first subtracting a zero-order model profile $L_\mathrm{m}$ and applying the filter only to the delta-profile $\delta L_\mathrm{rm} = L_\mathrm{r} - L_\mathrm{m}$. This approach efficiently mitigates residual numerical biases. After the application of the BWS filter, the model profile is added back again. We express the BWS filter as a linear matrix operator $\mathbf{A}^{\mathrm{BWS}}$ and get (Item 1.2 in Figure 2)

$$
L_{\mathrm{F}i} = L_{\mathrm{m}i} + \sum_{j=0}^{N} A_{ij}^{\mathrm{BWS}} \cdot \delta L_{\mathrm{rm}j}, \tag{A2}
$$

for the filtered excess phase, where $j \in \{1,2,...,N\}$. The band matrix operator $\mathbf{A}^{\mathrm{BWS}}$ has elements

$$
A_{ij}^{\mathrm{BWS}} = \begin{cases} 0 & \text{for } j < i - M/2 \text{ and for } j > i + M/2 \\ w_{j-i+M/2} & \text{for } i - M/2 < j < i + M/2 \end{cases} . \tag{A3}
$$

The central filter weight $w_{0+M/2}$ at $j=i$ is the $(M/2)^{\text{th}}$ filter-element (according to the definition of the BWS weights below), therefore its index is $M/2$. With $m = j - i + M/2$ (and therefore $0 \le m \le M$), each single BWS weight is calculated using

$$
w_m = \frac{w_{\mathrm{raw},m}}{\sum_{m=0}^{M} w_{\mathrm{raw},m}} \tag{A4}
$$

and

$$
w_{\mathrm{raw},m} = \begin{cases} \frac{\sin(2\pi f_\mathrm{c}/f_\mathrm{s}(m-M/2))}{m-M/2}\left[0.42 - 0.5\cos\left(2\pi\frac{m}{M}\right) + 0.08\cos\left(4\pi\frac{m}{M}\right)\right] & \text{for } m \neq M/2 \\ 2\pi f_\mathrm{c}/f_\mathrm{s} & \text{for } m = M/2 \end{cases} . \tag{A5}
$$

The *estimated random uncertainty* is then propagated by covariance propagation (Item 1.3 in Figure 2),

$$
\mathbf{C}_{LF} = \mathbf{A}^{\mathrm{BWS}} \cdot \mathbf{C}_{Lr} \cdot (\mathbf{A}^{\mathrm{BWS}})^{\mathrm{T}}. \tag{A6}
$$





The random uncertainty profile $u^r_{LF}$ and the error correlation matrix $R_{LF}$ are not needed for the subsequent random uncertainty propagation, but are calculated from $\mathbf{C}_{LF}$ for being available for the L1b output, using

$$u^r_{\mathrm{LF},i} = \sqrt{C_{\mathrm{LF},ii}} \tag{A7}$$

and

$$R_{\mathrm{LF},ij} = \frac{C_{\mathrm{LF},ij}}{u^r_{\mathrm{LF},i} u^r_{\mathrm{LF},j}}. \tag{A8}$$

The *correlation length* profile $l_{Lr}$ has elements

$$l_{\mathrm{Lr},i} = \left.\frac{\mathrm{d}z}{\mathrm{d}t}\right|_i \cdot |t_i - t(R_{\mathrm{LF},ij} = 1/e)| \tag{A9}$$

computed upward and downward from the main peak of the correlation function and then averaged. Here $\mathrm{d}z/\mathrm{d}t$ is the scan velocity profile, obtained from using the msl altitude grid $z_t$ calculated as part of the forward modeling towards $L_{\mathrm{m}}$ at the corresponding time grid $t$ (cf. Table 2).

We note that after the L2a refractivity retrieval also the msl altitude grid consistent with the retrieved refractivity profile could be used (as described by SKS2017, Appendix A therein), from a repeated forward modeling. The difference for the scan velocity estimate is found very small, however, since the forward-modeled $z_t$ based on co-located refractivity profiles from ECMWF short-range forecast fields is already sufficiently reliable and this also keeps the L1b processor as a decoupled predecessor of the L2a processor.

For the *estimated systematic uncertainty*, interpreted as a basic systematic uncertainty (Section 2.2), we apply the same lowpass filter as used for the *state* profile (Item 1.2 in Figure 2), but with no zero-order profile subtracted, i.e.,

$$u^s_{\mathrm{LF}i} = \sum_{j=0}^{N} A^{\mathrm{BWS}}_{ij} \cdot u^s_{\mathrm{Lr}j}. \tag{A10}$$

The *resolution* in time of $L_F$ and its uncertainties, $\tau_{\mathrm{BW}}$, is the Boxcar-equivalent width (cf. Figure A1a) determined by the cutoff-frequency $f_{\mathrm{c}}$ of the BWS Filter,

$$\tau_{\mathrm{LF}} \approx \frac{1}{f_{\mathrm{c}} + \Delta f_{\mathrm{c}}/2} \approx \frac{1}{2f_{\mathrm{c}}}, \tag{A11}$$

with our design choice $\tilde{M} = 2(f_{\mathrm{s}}/f_{\mathrm{c}})$ and using that the BWS filter stopband-to-passband transition width is (Smith, 1999)

$$\Delta f_{\mathrm{c}} \approx \frac{4f_s}{\tilde{M}}. \tag{A12}$$

Given $f_{\mathrm{c}} = 2.5\,\mathrm{Hz}$, this results in an effective resolution $\tau_{\mathrm{LF}} = 0.2\,\mathrm{s}$ and corresponds to the resolution obtained when applying a 11 pts boxcar filter as explained at the beginning of this section above. The filter window inter-comparison in Figure A1a also illustrates this, because the full width at half maximum of the $2.5\,\mathrm{Hz}$ - 41 pts BWS filter is 11 pts.

This resolution in time can finally be converted to the vertical (msl altitude) resolution in space

$$w_{\mathrm{LF},i} = \left.\frac{\mathrm{d}z}{\mathrm{d}t}\right|_i \cdot \tau_{\mathrm{LF}}, \tag{A13}$$

where, as for the correlation length estimation (Equation A9), the scan velocity profile is employed to convert from the time domain to msl altitude domain.



### A1.2 Doppler Shift Derivation

After the application of the BWS filter to the excess phase profiles $L_\mathrm{r}$ (for both carrier frequencies of the given GNSS system), the *state* profile of the Doppler is derived from the filtered phase profiles $L_\mathrm{F}$ (Item 1.4 in Figure 2). To minimize systematic errors from the numerical differentiation to negligible magnitude, the model profile $L_\mathrm{m}$ is again subtracted from the filtered phase profile,

$$\delta L_\mathrm{Fm} = L_\mathrm{F} - L_\mathrm{m}, \tag{A14}$$

and the resulting delta-profile $\delta L_\mathrm{Fm}$ is then differentiated. After the derivative, the zero-order model profile $D_\mathrm{m}$ (also available from the L1b forward modeling) is added.

Based on careful tests of different formulations, we use a five-point derviative scheme. The discretization of this five-point derivative $\delta D_{\mathrm{rm},i}$ is given by

$$\delta D_{\mathrm{rm},i} = \frac{\mathrm{d}\delta L_\mathrm{Fm}(t)}{\mathrm{d}t}\bigg|_i = \frac{-\delta L_{\mathrm{Fm},i-2} + 8\delta L_{\mathrm{Fm},i-1} - 8\delta L_{\mathrm{Fm},i+1} + \delta L_{\mathrm{Fm},i+2}}{-t_{i-2} + 8t_{i-1} - 8t_{i+1} + t_{i+2}}, \tag{A15}$$

for each of the frequencies (e.g., Syndergaard, 1999). This can be expressed in matrix form as

$$D_{\mathrm{r},i} = D_{\mathrm{m},i} + \delta D_{\mathrm{rm},i} = D_{\mathrm{m},i} + \sum_{j=1}^{N} A_{ij}^{\mathrm{L2D}} \cdot \delta L_{\mathrm{Fm},j}, \tag{A16}$$

using matrix operator $\mathbf{A}^{\mathrm{L2D}}$ with

$$\mathbf{A}^{\mathrm{L2D}} = \frac{1}{12\Delta t}\begin{bmatrix} -18 & 24 & -6 & 0 & 0 & 0 & & 0 & 0 \\ -6 & 0 & -6 & 0 & 0 & 0 & \ddots & 0 & 0 \\ -1 & 8 & 0 & -8 & 1 & 0 & & 0 & 0 \\ 0 & -1 & 8 & 0 & -8 & 1 & \ddots & 0 & 0 \\ & \ddots & & \ddots & & \ddots & & \ddots & \\ 0 & 0 & 0 & 0 & 0 & 0 & \ddots & -8 & 1 \\ 0 & 0 & 0 & 0 & 0 & 0 & & 0 & -6 \\ 0 & 0 & 0 & 0 & 0 & 0 & & 24 & -18 \end{bmatrix}, \tag{A17}$$

where $\Delta t = t_{i+1} - t_i$, being $0.02$s in our case of $f_\mathrm{s} = 50\,\mathrm{Hz}$.

The *estimated random uncertainty* can then be propagated (Item 1.5 in Figure 2) using

$$\mathbf{C}_D = \mathbf{A}^{\mathrm{L2D}} \cdot \mathbf{C}_{LF} \cdot (\mathbf{A}^{\mathrm{L2D}})^\mathrm{T}. \tag{A18}$$

The covariance matrix is again (cf. Equations A7 and A8) decomposed into estimated random uncertainties and error correlation matrix (Item 2.2 in Figure 2) using

$$u^r_{Dr,i} = \sqrt{C_{Dr,ii}} \tag{A19}$$





and

$$R_{D\mathrm{r},ij} = \frac{C_{D\mathrm{r},ij}}{u^r_{D\mathrm{r},i} u^r_{D\mathrm{r},j}}. \tag{A20}$$

For the *estimated systematic uncertainty*, further on interpreted as basic systematic uncertainty (cf. Equation A10), we apply the derivative operator (Item 1.4 in Figure 2) to the systematic uncertainties, with no zero-order profile subtracted, i.e.,

$$u^s_{D\mathrm{r},i} = \sum_{j=1}^{N} A^{\mathrm{L2D}}_{ij} \cdot u^s_{\mathrm{LF},j}. \tag{A21}$$

The *resolution* remains unaffected by the Doppler shift derivation, since the five-point sample width of the derivative operator is fully within the eleven-point effective filter width (stopband) of the BWS filter applied before, so that $\tau_{D\mathrm{r}} = \tau_{\mathrm{LF}}$ and $w_{D\mathrm{r}} = w_{\mathrm{LF}}$.

## A2 Bending Angle Retrieval

### A2.1 GO Bending Angle Retrieval

From the Doppler shift *state* profile $D_\mathrm{r}$ (again for both frequencies of the given GNSS system) we can derive the impact parameter profile $a_t$ and geometric-optics (GO) bending angle profile $\alpha_\mathrm{G}$ (Item 2.1 in Figure 2) using first the geometric relation

$$D_{\mathrm{r},i} = [v_{\mathrm{R},i} \cos(\phi_{\mathrm{R},i}) - v_{\mathrm{T},i} \cos(\phi_{\mathrm{T},i})] - \dot{r}_{\mathrm{RT},i}, \tag{A22}$$

where

$$\phi_{\mathrm{R},i} = \eta_{\mathrm{R},i} - \arcsin\left(\frac{a_{t,i}}{r_{\mathrm{R},i}}\right), \tag{A23}$$

and

$$\phi_{\mathrm{T},i} = (\pi - \eta_{\mathrm{T},i}) - \arcsin\left(\frac{a_{t,i}}{r_{\mathrm{T},i}}\right) \tag{A24}$$

for each individual level of the time grid $t_i$, in order to determine $a_t$ from sequential application to all levels (Kursinski et al., 1997; Syndergaard, 1999). Here $v_{\mathrm{R},i} := |\boldsymbol{v}_{\mathrm{R},i}|$ is the receiver velocity, $r_{\mathrm{R},i} := |\boldsymbol{r}_{\mathrm{R},i}|$ the receiver radial position, $\eta_{\mathrm{R},i}$ the angle between the receiver velocity and position vectors, $\phi_{\mathrm{R},i}$ then the angle between the receiver velocity and raypath vectors (and all these equivalently for the transmitter), and $\dot{r}_{\mathrm{RT},i} := \left|\frac{d(\boldsymbol{r}_\mathrm{T} - \boldsymbol{r}_\mathrm{R})}{dt}\right|_i$ is the time-derivative of the distance between the transmitter and the receiver at time $t_i$, i.e., the 'kinematic straight-line Doppler shift' to be subtracted in Equation A22 to match the (excess) Doppler shift $D_{\mathrm{r},i}$ induced by the atmosphere (and ionosphere).

Based on $a_t$, the elements of the GO bending angle profile $\alpha_\mathrm{G}$ are subsequently calculated using another geometrical relation,

$$\alpha_{\mathrm{G},i} = \theta_{\mathrm{RT},i} - \arccos\left(\frac{a_{t,i}}{r_{\mathrm{R},i}}\right) - \arccos\left(\frac{a_{t,i}}{r_{\mathrm{T},i}}\right), \tag{A25}$$





where $\theta_{\mathrm{RT},i}$ is the opening angle between the transmitter and receiver position vectors. Syndergaard (1999), Figure 1.5 therein, provides an illustration of the relevant geometry.

All the variables in Equations A22–A25 are defined in the occultation plane spanned by the receiver and transmitter position vectors after oblateness correction (Syndergaard, 1998), i.e., after they have been transformed to originate in the Earth ellipsoid's center of local curvature in the occultation plane at the mean tangent point location of the RO event. Using this center of local curvature rather than in the Earth's center of mass as the origin is essential to ensure that the assumption of spherical symmetry, implicit in Equations A22 to A25, is accurately valid geometrically.

The impact parameter retrieval is solved iteratively, because it is impossible to rearrange Equations A22 to A24 into an explicit expression for the retrieval of the impact parameter; but it is mildly non-linear and converges fast, in particular if the initial guess for $a_{t,i}$ is estimated from the previous level (starting at the top level with the straight-line impact parameter).

After the GO bending angle retrieval, the bending angles of all GNSS frequencies are interpolated to a common monotonic impact altitude grid $z_a$ (Item 2.6 in Figure 2), based on the monotonically sorted impact parameter grid of the leading channel, $a_{t1}$ (i.e., $k = 1$).

For each element of $z_a$ we get (Item 2.3 in Figure 2)

$$z_{a,i} = a_{t,j1} - h_{\mathrm{G}} - R_{\mathrm{C}}, \tag{A26}$$

where $j$ is the index of the elements of the sorted impact parameter grid $a_{t1}$. $h_{\mathrm{G}}$ is the geoid undulation (see Scherllin-Pirscher et al. (2017) for a detailed discussion of its use in RO analysis), and $R_{\mathrm{C}}$ is the local radius of curvature of the RO event.

Because the impact parameter is only implicitly expressed in Equations A22–A24, but GUM-type uncertainty propagation along Equations 2 and 4 requires an explicit measurement model, we make use of a linearization of the bending angle retrieval. We use the approach described by Melbourne et al. (1994), and applied to uncertainty propagation by Syndergaard (1999), for the propagation of the *estimated random uncertainty* from Doppler shift $D_{\mathrm{r}}$ to GO bending angle $\alpha_{\mathrm{G}}$ (Item 2.5 in Figure 2).

This linearization establishes a direct relation between random uncertainties of the Doppler shift $u^r_{D\mathrm{r}}$, and the uncertainties of the bending angle $u^r_{\alpha\mathrm{G}}$, using

$$u^r_{\alpha\mathrm{G}(t),i} \approx - \left( \frac{\mathrm{d}a_{\mathrm{SL}}}{\mathrm{d}t} \right)^{-1} \Bigg|_i u^r_{D\mathrm{r}(t),i}, \tag{A27}$$

where $a_{\mathrm{SL}}$ is the straight-line impact parameter. These bending angle uncertainties $u^r_{\alpha\mathrm{G}}$ are relative to the time grid as independent coordinate. To get the desired uncertainties with respect to the impact altitude grid $z_a$ (introduced in Equation A26), the uncertainties of the impact altitude $z_a$ need to be transferred to the bending angle, so that the $z_a$ grid can subsequently be considered free of error. Syndergaard (1999) showed that this can be done by replacing Equation A27 with

$$u^r_{\alpha\mathrm{G}(z_a),i} \approx - \left( \frac{\mathrm{d}a^{\mathrm{T}}_t}{\mathrm{d}t} \right)^{-1} \Bigg|_i u^r_{D\mathrm{r}(t),i}, \tag{A28}$$

where $a^{\mathrm{T}}_t$ is the '*true*' impact parameter. We use the forward modeled impact parameter $a_{t\mathrm{m}}$ instead (i.e., adopt $a_{t\mathrm{m}} = a^{\mathrm{T}}_t$) and assume that the relative error made thereby is smaller than the $2\%$ relative error estimated by Melbourne et al. (1994) for the





linearization applied here. This is a reasonable assumption given the high quality of our forward modeled profiles derived from ECMWF short-range forecast refractivity fields. We do account for the linearization error by adding a factor $f_{u\alpha lin} = 0.02$ to the uncertainty of the retrieved GO bending angle

$$u^r_{\alpha G,i} = u^r_{\alpha G(z_a),i} \cdot (1 + f_{u\alpha lin}). \tag{A29}$$

We note that although the calculation of the *state* of the bending angle does not make use of the linearization, and therefore the linearization does not increase the uncertainty of the *state* profile, it may increase the error in the uncertainty estimate itself.

Finally, the $u^r_{\alpha G}$ profile is also interpolated to the common monotonic impact altitude grid $z_a$.

The bending angle values at each grid point only depend on the Doppler shift values of the same grid points, i.e., the existing correlations between the errors at different levels are left unchanged, i.e., $R_{\alpha G} = R_{Dr}$. The covariance matrix can hence be

calculated by recombining the Doppler shift correlation matrix with the propagated uncertainties (Item 2.8 in Figure 2),

$$C_{\alpha G,ij} = u^r_{\alpha G,i} \cdot u^r_{\alpha G,j} \cdot R_{Dr,ij}. \tag{A30}$$

For the propagation of the *estimated systematic uncertainty* (Item 2.4 in Figure 2) three types of potential systematic errors adding to the impact parameter uncertainty $u^s_{at}$, and consequentially the bending angle uncertainty $u^s_{\alpha G}$, are taken into account. Systematic errors in the Doppler shift, i.e., $u^s_{Dr}$, systematic errors in the velocities of the satellites, i.e., $u^s_{\boldsymbol{v}R}$ and $u^s_{\boldsymbol{v}T}$, and

systematic errors in the positions of the satellites, i.e., $u^s_{\boldsymbol{r}R}$ and $u^s_{\boldsymbol{r}T}$. The latter two orbit-borne types are interpreted as apparent systematic uncertainties (Section 2.2) while the excess phase-borne uncertainty $u^s_{Dr}$ is a basic systematic uncertainty.

For the propagation of these estimated systematic uncertainties to $u^s_{\alpha G}$, Equations A22–A24 are linearized around the retrieved state quantities as zero-order contributions and no terms higher than first-order are kept. Then $\phi_R$ and $\phi_T$ in Equation A22 are substituted by the linearized versions of Equations A23 and A24 and the resulting equation is solved (level by

level) for the impact parameter $a_{t,i} = f(D_{r,i}, r_{R,i}, r_{T,i}, v_{R,i}, v_{T,i})$, with $i = 1, 2, ..., N$. Adopting the first-order deviations to represent the estimated systematic uncertainties we obtain

$$u^s_{at,i} = \frac{1}{k_{at,i}} \sqrt{(u^s_{Dr,i})^2 + (k_{vR,i} \cdot u^s_{vR,i})^2 + (k_{rR,i} \cdot u^s_{rR,i})^2 + (k_{vT,i} \cdot u^s_{vT,i})^2 + (k_{rT,i} \cdot u^s_{rT,i})^2}, \tag{A31}$$

where




$$k_{at,i} := \frac{\partial D_r}{\partial \phi_R}\Big|_i \cdot \frac{\partial \phi_R}{\partial a_t}\Big|_i + \frac{\partial D_r}{\partial \phi_T}\Big|_i \cdot \frac{\partial \phi_T}{\partial a_t}\Big|_i = -v_{R,i} \cdot \sin\phi_{Ri} \cdot \frac{1}{\sqrt{r_{R,i}^2 - a_{t,i}^2}} - v_{T,i} \cdot \sin\phi_{Ti} \frac{1}{\sqrt{r_{T,i}^2 - a_{t,i}^2}}, \tag{A32}$$

$$k_{vR,i} := -\frac{\partial D_r}{\partial v_R}\Big|_i = -\cos\phi_{Ri},$$

$$k_{vT,i} := -\frac{\partial D_r}{\partial v_T}\Big|_i = -\cos\phi_{Ti},$$

$$k_{rR,i} := \frac{\partial D_r}{\partial \phi_R}\Big|_i \cdot \frac{\partial \phi_R}{\partial r_R}\Big|_i = \frac{v_{R,i} \cdot \sin\phi_{Ri} \cdot a_{t,i}}{r_{R,i}\sqrt{r_{R,i}^2 - a_{t,i}^2}},$$

$$k_{rT,i} := \frac{\partial D_r}{\partial \phi_T}\Big|_i \cdot \frac{\partial \phi_T}{\partial r_T}\Big|_i = \frac{v_{T,i} \cdot \sin\phi_{Ti} \cdot a_{t,i}}{r_{T,i}\sqrt{r_{T,i}^2 - a_{t,i}^2}}.$$

A number of simplifications have been made to arrive at this result. First, the last term in Equation A22 is disregarded since errors in the positions are assumed to be constant with respect to the short time duration of an RO event; remaining errors $\Delta\dot{r}_{RT}$ after taking the derivative are therefore of higher order. Next, orbit position and velocity uncertainties are both assumed to be constant within the short duration of an event and the velocity uncertainties obtained are interpreted as uncertainties along the direction of the velocity vector. Consequentially, the uncertainty is also projected along with the vector into the raypath direction. A more conservative estimation (that we consider overly conservative in context) would interpret the uncertainties as ellipsoids at the velocity vectors' heads, and would hence take the full magnitude of the uncertainties along the raypath direction (without projection).

Furthermore, since all error sources (the processing of the occultation tracking data and the POD for transmitter and receiver) are essentially independent from each other, the different input uncertainties are assumed to be uncorrelated. Finally, we reasonably assumed the errors of the angle between the position and velocity vectors ($\eta$) to be negligible ($u_\eta^s \approx 0$) for the purpose here, for both the transmitter and receiver.

In order to finally derive the systematic uncertainty of the bending angle from the impact parameter's uncertainty, we continue with a linearization of Equation A25 and arrive at

$$u_{\alpha G,i}^s = \sqrt{(k_{at,i} \cdot u_{at,i}^s)^2 + (k_{rR,i} \cdot u_{rR,i}^s)^2 + (k_{rT,i} \cdot u_{rT,i}^s)^2}, \tag{A33}$$

where

$$k_{at,i} := \frac{\partial \alpha}{\partial a_t}\Big|_i = \frac{1}{\sqrt{r_{R,i}^2 + a_{t,i}^2}} + \frac{1}{\sqrt{r_{T,i}^2 + a_{t,i}^2}}, \tag{A34}$$

$$k_{rR,i} := \frac{\partial \alpha}{\partial r_R}\Big|_i = -\frac{a_{t,i}}{r_{R,i}\sqrt{r_{R,i}^2 - a_{t,i}^2}},$$

$$k_{rT,i} := \frac{\partial \alpha}{\partial r_T}\Big|_i = -\frac{a_{t,i}}{r_{T,i}\sqrt{r_{T,i}^2 - a_{t,i}^2}}.$$





In practice we separately calculate the basic and apparent systematic uncertainty estimates ($u_{\alpha G}^b$ from the first RHS terms in Equations A31 and A33, $u_{\alpha G}^a$ from the orbit-borne terms) and afterwards obtain $u_{\alpha G}^s$ as a combined result, in order to enable separate propagation in subsequent processing steps.

The *resolution* profile remains unaffected by the bending angle retrieval, since the level-by-level approach of the algorithm
does not create extra correlation and further vertical smoothing, so that $\tau_{\alpha G} = \tau_{Dr}$ and $w_{\alpha G} = w_{Dr}$.

### A2.2   WO Bending Angle Retrieval

After the GO bending angle, the wave-optics (WO) bending angle *state* profile $\alpha_W(z_a)$ is retrieved (Item 2.7 in Figure 2) from excess phase profile $L_r(t)$ (and its uncertainties) and the amplitude profile $A_r(t)$ (and uncertainties) in a WO retrieval following Gorbunov and Kirchengast (2015, 2017). Along with the state profile, the *systematic uncertainty* profile $u_{\alpha W}^s$, the covariance
matrix $\mathbf{C}_{\alpha W}$, and the *resolution* profile $w_{\alpha W}$ are derived.

The covariance matrix $\mathbf{C}_{\alpha W}$ is then decomposed to *random uncertainty* profile $u_{\alpha W}^r$ and correlation matrix $R_{\alpha W}$ in the same form as done above for $\mathbf{C}_{Dr}$ (Equations A19 and A20) and $\mathbf{C}_{LF}$ (Equations A7 and A8). The estimated systematic uncertainty $u_{\alpha W}^s$ is composed of a basic systematic uncertainty $u_{\alpha W}^b$, propagated through the wave-optical retrieval from the excess phase uncertainty $u_{Lr}^s$, and an apparent systematic uncertainty $u_{\alpha W}^a$, estimated in the lower troposphere as residual bias uncertainty
of a regression-based boundary layer bias correction (Gorbunov and Kirchengast, 2017).

The WO bending angle retrieval algorithm and the associated uncertainty propagation algorithm are not explicitly described here; the reader is referred to Gorbunov and Kirchengast (2015) and Gorbunov and Kirchengast (2017). However, we have prepared the merging with the WO bending angle variables (they will be included when their ongoing integration and testing in the rOPS is complete), which is described next.

### A2.3   Merging of GO and WO Bending Angle Profiles

The $\alpha_W$ profile, prepared on the common grid $z_a$, and the $\alpha_G$ profile are merged over an upper tropospheric transition range (Item 2.9 in Figure 2). The gradual transition, weighted by a symmetric half-sine function, has a defined impact altitude transition of half-width $\Delta z_a^{GW} = 2\,\text{km}$ around transition altitude $z_a^{GW}$, allowed within $9\,\text{km}$ to $14\,\text{km}$, estimated from $\alpha_G$ data quality. The resulting merged bending angle profile $\alpha_M$ is

$$\alpha_{M,i} = \gamma_i \cdot \alpha_{G,i} + (1 - \gamma_i) \cdot \alpha_{W,i}, \tag{A35}$$

where the weighting profile $\gamma$ is formulated as

$$\gamma_i = \begin{cases} 1 & \text{for } z_{a,i} \geq z_a^{GW} + \Delta z_a^{GW} \\ 0.5 \cdot \left[ \sin\left( \frac{\pi}{2} \cdot \frac{z_{a,i} - z_a^{GW}}{\Delta z_a^{GW}} \right) + 1 \right] & \text{for } |z_{a,i} - z_a^{GW}| < \Delta z_a^{GW} \\ 0 & \text{for } z_{a,i} \leq z_a^{GW} - \Delta z_a^{GW}. \end{cases} \tag{A36}$$

To determine the random uncertainties for the merged GO-WO input bending angle, we need to merge the covariance matrices of both bending angles.



We can assume both incoming covariance matrices $\mathbf{C}_{\alpha\text{G}}$ and $\mathbf{C}_{\alpha\text{W}}$ are provided on the common monotonic target grid $z_a$ (i.e., also the WO uncertainties and correlations are interpolated to this common grid before the merger). We further can reasonably assume that there are no cross-correlations between GO and WO errors, given the very different retrieval schemes. Based on this we can compose the covariance matrix of the merged bending angle profile, $\mathbf{C}_{\alpha\text{M}}$ (Item 2.10 in Figure 2) as

5 follows. Outside the merging zone (i.e., outside of $z_a^{\text{GW}} \pm \Delta z_a^{\text{GW}}$) we can assign

$$
\mathrm{C}_{\alpha\text{M},ij} =
\begin{cases}
\mathrm{C}_{\alpha\text{G},ij} & \text{for } z_{a\text{Top}} > z_{a,i} > z_a^{\text{GW}} + \Delta z_a^{\text{GW}} \text{ and } z_{a\text{Top}} > z_{a,j} > z_a^{\text{GW}} + \Delta z_a^{\text{GW}} \\
\mathrm{C}_{\alpha\text{W},ij} & \text{for } z_a^{\text{GW}} - \Delta z_a^{\text{GW}} > z_{a,i} > z_{a\text{Bot}} \text{ and } z_a^{\text{GW}} - \Delta z_a^{\text{GW}} > z_{a,j} > z_{a\text{Bot}} \\
0 & \text{for } z_{a\text{Top}} > z_{a,i} > z_a^{\text{GW}} + \Delta z_a^{\text{GW}} \text{ and } z_a^{\text{GW}} - \Delta z_a^{\text{GW}} > z_{a,j} > z_{a\text{Bot}} \\
& \text{for } z_a^{\text{GW}} - \Delta z_a^{\text{GW}} > z_{a,i} > z_{a\text{Bot}} \text{ and } z_{a\text{Top}} > z_{a,j} > z_a^{\text{GW}} + \Delta z_a^{\text{GW}}
\end{cases},
\tag{A37}
$$

while within the merging zone we can assign

$$
\mathrm{C}_{\alpha\text{M},ij} = \gamma_i \gamma_j \mathrm{C}_{\alpha\text{G},ij} + (1 - \gamma_i)(1 - \gamma_j)\mathrm{C}_{\alpha\text{W},ij},
\tag{A38}
$$

wherein $i$ is understood such that $z_a^{\text{GW}} + \Delta z_a^{\text{GW}} > z_{a,i} > z_a^{\text{GW}} - \Delta z_a^{\text{GW}}$ and $j$ such that $z_{a\text{Top}} > z_{a,j} > z_{a\text{Bot}}$.

Because of the symmetry of the covariance matrix, the covariance elements in the merging zone orthogonal to the one above, i.e., for $z_a^{\text{GW}} + \Delta z_a^{\text{GW}} > z_{a,j} > z_a^{\text{GW}} - \Delta z_a^{\text{GW}}$ and $z_{a\text{Top}} > z_{a,i} > z_{a\text{Bot}}$, are calculated according to the same formula.

Due to the linear relation between $\alpha_\text{M}$, $\alpha_\text{G}$ and $\alpha_\text{W}$, expressed by Equation A35, a bias $u_{\alpha\text{G}}^s$ in the GO bending angle and a bias $u_{\alpha\text{W}}^s$ in the WO bending angle can be as well linearly combined and we can compute the *estimated systematic uncertainty* of the merged bending angle $u_{\alpha\text{M}}^s$ according to (Item 2.9 in Figure 2)

$$
u_{\alpha\text{M},i}^s = \gamma_i \cdot u_{\alpha\text{G},i}^s + (1 - \gamma_i) \cdot u_{\alpha\text{W},i}^s.
\tag{A39}
$$

In practice this formulation is again applied separately for the basic and apparent systematic uncertainty estimates, afterwards obtaining the $u_{\alpha\text{M}}^s$ profile as a combined result, in order to allow separate propagation in subsequent processing steps.

The *resolution* profile of the bending angle, $w_{\alpha\text{M}}$, is equal to the GO resolution $w_{\alpha\text{G}}$ above $z_a^{\text{GW}} + \Delta z_a^{\text{GW}}$, equal to the the WO resolution $w_{\alpha\text{W}}$ below $z_a^{\text{GW}} - \Delta z_a^{\text{GW}}$ and has a transition with transition weight $\gamma_i$ in between, again following the linear

formulation such as in Equations A35 and A39.

Because the integration and testing of the uncertainty propagation through the rOPS WO bending angle retrieval is currently still ongoing, as noted in Section A2.2 above, the examples shown in this study are all GO-only, i.e., only the GO retrieval is performed. The merging algorithm as described is ready to include the WO bending angles, however.

## A3  Atmospheric Bending Angle Derivation

In order to retrieve the atmospheric bending angle profile $\alpha_\text{r}$, ionospheric effects need to be corrected for, using the retrieved bending angles from each transmitter frequency channel. Since the only GNSS constellation currently used for RO is the GPS—except for recent initial data from the Chinese GNOS instrument using BeiDou signals (Liao et al., 2016; Bai et al.,





2017)—the data characteristics of the GPS case (with $k \in \{1,2\}$, $f_{T1} = 1.57542\,\mathrm{GHz}$, and $f_{T2} = 1.22760\,\mathrm{GHz}$) are in the prime focus of this section.

This concerns in particular special provisions for the minor (L2) channel noise filtering and its tropospheric extrapolation. In general the algorithms are applicable for any of the available GNSS systems, however; if the minor channel ($f_{T2}$) delivers
similar data quality as the major one ($f_{T1}$), the special provisions for the former will practically take no effect.

**A3.1   Adaptive Lowpass Filtering and Minor Channel Extrapolation**

Before applying the dual-frequency ionospheric correction, the merged bending angle *state* profiles $\alpha_{M,k}(z_a)$ at the common $z_a$ grid are filtered with further BWS lowpass filter operations and the minor channel is extrapolated.

For $\alpha_{M1}$ the filter is set to the same cutoff-frequency as the basic BWS filter preceeding the Doppler derivation (i.e., $f_{c1} =$
$2.5\,\mathrm{Hz}$), ensuring a reliable reference resolution and basic smoothness of the whole merged profile. For filtering of $\alpha_{M2}$ a (GPS L2) noise-minimization algorithm is used, following the approach of Sokolovskiy et al. (2009) for optimal filtering for ionospheric correction. We search for minimized noise employing a flexible cutoff-frequency $f_{c2} \in \{2.5\,\mathrm{Hz}, 2\,\mathrm{Hz}, 10/7\,\mathrm{Hz}, 1\,\mathrm{Hz}, 5/7\,\mathrm{Hz}, 0.5\,\mathrm{Hz}\}$, corresponding to using cutoff-periods $\tau_{c2}$ from $0.4\,\mathrm{s}$ to $2\,\mathrm{s}$ and sample widths of $M = 40$ to $M = 200$ (on BWS filter design details see Section A1.1).

We adopt that cutoff-frequency $f_{c2}$ for $\alpha_{M2}$ filtering that minimizes the noise fluctuations of the ionosphere-corrected atmospheric bending angle delta-profile $\delta\alpha_{\mathrm{rm}}^{fc2}(z_a) = \alpha_{\mathrm{r}}^{fc2}(z_a) - \alpha_{\mathrm{m}}(z_a)$ when evaluated over the mesospheric altitude range between $50\,\mathrm{km}$ and $70\,\mathrm{km}$ (similar to the functional minimization of Sokolovskiy et al. (2009); Eq. 4 therein). At these high altitudes the residual atmospheric mean signal after subtraction of the forward-modeled signal $\alpha_m(z_a)$ is very small ($< 0.03$–$0.3\,\mu\mathrm{rad}$) and therefore the noise level representative for the given RO event is well quantifiable.

The weight-matrix of the BWS filter, $\mathbf{A}_k^{\mathrm{BWS}}$, is determined for both frequencies analogously to Equations A3 to A5. Using the baseband approach with model profile $\alpha_{\mathrm{m}}$ to create the delta-profile $\delta\alpha_{\mathrm{Mm}}$ with elements

$$\delta\alpha_{\mathrm{Mm}i,k} = \alpha_{\mathrm{M}i,k} - \alpha_{\mathrm{m}i}, \tag{A40}$$

the filtered bending angle is then (Item 3.1 in Figure 2)

$$\alpha_{\mathrm{F}i,k} = \alpha_{\mathrm{m}i} + \sum_{j=0}^{N} A_{ij,k}^{\mathrm{BWS}} \cdot \delta\alpha_{\mathrm{Mm}j,k}, \tag{A41}$$

where $i,j \in \{1,2,...,N\}$ and $k \in \{1,2\}$.

Due to the stronger power of the L1 signal for (most of) the GPS satellites, the GPS signals of both frequencies are not of the same quality and the L2 data (for those satellites where encrypted and hence power-degraded L2 signals are transmitted) do not reach down as far as the L1 data (i.e., $z_{a\mathrm{Bot}2} > z_{a\mathrm{Bot}1}$). If due to this reason $\alpha_{F2}$ does not reach down as far as $\alpha_{F1}$ and $z_{a\mathrm{Bot}2} \leq z_{a\mathrm{Bot}2\mathrm{Max}}$ (with $z_{a\mathrm{Bot}2\mathrm{Max}}$ currently set to $15\,\mathrm{km}$), a *tropospheric bending angle extrapolation* (TBAE) is applied in
order to artificially extend $\alpha_{M2}$ to also reach down to $z_{a\mathrm{Bot}1}$ (Item 3.3 in Figure 2).

Briefly summarized, this TBAE is currrently implemented as follows. A linear gradient profile for the difference profile between the two bending angles, $\alpha_{F12} = (\alpha_{F1} - \alpha_{F2})$, is estimated by a least squares fit over a sufficiently wide impact altitude





range from $z_{a\text{Bot}2}$ upward (as wide as the extrapolation range, at least 10 km). This gradient profile is then linearly extended down to $z_{a\text{Bot}1}$ and subtracted from $\alpha_{\text{F}1}$, to obtain the extrapolated part of $\alpha_{\text{F}2}$ from $z_{a\text{Bot}2}$ to $z_{a\text{Bot}1}$. If $z_{a\text{Bot}2} > z_{a\text{Bot}2\text{Max}}$ then no TBAE is performed since the extrapolation range is considered too large. Details are provided by Kirchengast et al. (2017b), where the most recent version of the atmospheric bending angle derivation is described that includes this $\alpha_{\text{F}12}$ ex-
trapolation in a further advanced form.

For the propagation of the *estimated random uncertainty* we get (Item 3.2 in Figure 2),

$$\mathbf{C}_{\alpha\text{F},k} = \mathbf{A}_k^{\text{BWS}} \cdot \mathbf{C}_{\alpha\text{M},k} \cdot (\mathbf{A}_k^{\text{BWS}})^{\text{T}}, \tag{A42}$$

for the bending angle error covariance matrices of the leading ($k = 1$) and minor ($k = 2$) channel.

In case a TBAE is applied to $\alpha_{\text{F}2}$, the random uncertainty of $\alpha_{\text{F}2}$ below $z_{a\text{Bot}2}$ is equal to the one of $\alpha_{\text{F}1}$, because the noise
is "copied" from $\alpha_{\text{F}1}$ since the linear gradient profile from fitting $\alpha_{\text{F}12}$ is noise-free. As a consequence, in these cases, we set the matrix elements of $\mathbf{C}_{\alpha\text{F}2}$ to (Item 3.4 in Figure 2)

$$C_{\alpha\text{F}2,ij} = \begin{cases} C_{\alpha\text{F}2,ij} & \text{for } z_{a\text{Top}} > z_{a,i} > z_{a\text{Bot}2} \text{ and } z_{a\text{Top}} > z_{a,j} > z_{a\text{Bot}2} \\ C_{\alpha\text{F}1,ij} & \text{for } z_{a\text{Bot}2} > z_{a,i} > z_{a\text{Bot}1} \text{ and } z_{a\text{Bot}2} > z_{a,j} > z_{a\text{Bot}1} \\ 0 & \text{for } z_{a\text{Bot}2} > z_{a,i} > z_{a\text{Bot}1} \text{ and } z_{a\text{Top}} > z_{a,j} > z_{a\text{Bot}2} \\ 0 & \text{and } z_{a\text{Top}} > z_{a,i} > z_{a\text{Bot}2} \text{ for } z_{a\text{Bot}2} > z_{a,j} > z_{a\text{Bot}1} \end{cases}. \tag{A43}$$

$\mathbf{C}_{\alpha\text{F}1}$ and $\mathbf{C}_{\alpha\text{F}2}$ can then be decomposed as needed into $u_{\alpha\text{F}1}^r$, $\mathbf{R}_{\alpha\text{F}1}$, and $u_{\alpha\text{F}2}^r$, $\mathbf{R}_{\alpha\text{F}2}$, respectively. Kirchengast et al. (2017b) describe the most recent version consistent with a further advanced form of the TBAE, where the separate assignments accord-
ing to Equation A43 are no longer needed.

The *estimated systematic uncertainties* $u_{\alpha\text{M},k}^s$ (in practice the basic and apparent systematic uncertainty estimates separately) are filtered with the same filter settings as for the state profiles (Item 3.1 in Figure 2) and are thus obtained in the form

$$u_{\alpha\text{F}i,k}^s = \sum_{j=0}^{N} A_{ij,k}^{\text{BWS}} \cdot u_{\alpha\text{M}j,k}^s. \tag{A44}$$

Since these are smooth profiles they are marginally changed by this lowpass filtering. The systematic uncertainty component
contributed by the TBAE to the estimated systematic uncertainty is added after the ionospheric correction (see next subsection).

As for the basic lowpass filtering of excess phases (Section A1.1), the *resolution* profiles of the filtered bending angles $w_{\alpha\text{F}1}$ and $w_{\alpha\text{F}2}$ are determined by the cutoff-frequencies $f_{\text{c}1}$ and $f_{\text{c}2}$ of the BWS filters, following Equations A11 and A13.

**A3.2 Ionospheric Correction**

Based on the filtered and sometimes extrapolated *state* profiles $\alpha_{\text{F}1}$ and $\alpha_{\text{F}2}$, the ionospheric refractive effects are corrected
for by the standard dual-frequency correction of bending angles (Vorob'ev and Krasil'nikova, 1994) used in the $f_{\text{T}1}-f_{\text{T}2}$ difference-profile form (Sokolovskiy et al., 2009) (Item 3.5 in Figure 2). For the elements of the retrieved atmospheric bending





angle profile $\alpha_r$ we thus get

$$\alpha_{r,i} = \alpha_{F1,i} + \gamma_{fT12} \cdot \delta\alpha_{F12,i}, \tag{A45}$$

where

$$\delta\alpha_{F12,i} = \alpha_{F1,i} - \alpha_{F2,i}, \tag{A46}$$

and

$$\gamma_{fT12} = \frac{f_{T2}^2}{f_{T1}^2 - f_{T2}^2}. \tag{A47}$$

Propagated through the operator of the ionospheric correction (Equation A45, currently used here in the classical form with $f_{T1}$ and $f_{T2}$ terms) the *estimated random uncertainty* of the resulting atmospheric bending angle, expressed by the error covariance matrix $\mathbf{C}_{\alpha r}$ (Item 3.6 in Figure 2), is obtained as

$$\mathbf{C}_{\alpha r} = (1 + \gamma_{fT12})^2 \mathbf{C}_{\alpha F1} + \gamma_{fT12}^2 \mathbf{C}_{\alpha F2}. \tag{A48}$$

$\mathbf{C}_{\alpha r}$ can then also be decomposed into $u_{\alpha r}^r$ and $\mathbf{R}_{\alpha r}$ with the usual equations (cf., e.g., Equations A19 and A20).

Equation A45 is as well applied to propagate the *estimated systematic uncertainty* (in practice the basic and apparent systematic uncertainty estimates separately) through the ionospheric correction using (Item 3.5 in Figure 2)

$$u_{\alpha r,i}^s = u_{\alpha F1,i}^s + \gamma_{fT12} \cdot (u_{\alpha F1,i}^s - u_{\alpha F2,i}^s), \tag{A49}$$

where the systematic errors in $\alpha_{F1}$ and $\alpha_{F2}$ are assumed to be positively correlated (same sign), because both frequency channels share their sources for non-ionospheric systematic uncertainties (Doppler shift, orbit velocity, and orbit position uncertainties).

In case of TBAE, Equation A49 needs to be supplemented below $z_{aBot2}$, since additional uncertainties $u_{\alpha 2TE}^s$ arise from the errors made in the fitting parameters and in the extrapolation model (linear extrapolation) of the TBAE. Hence, for the range

$z_{aBot2} > z_{a,i} \geq z_{aBot1}$,

$$u_{\alpha r,i}^s = u_{\alpha r}^s(z_{aBot2}) + u_{\alpha 2TE,i}^s, \tag{A50}$$

with $u_{\alpha 2TE}^s$ being the conservative estimate for additional (apparent) systematic uncertainty within the extrapolated impact altitude range. We set $u_{\alpha 2TE}^s$ to zero at $z_{aBot2}$ and linearly increase it from there downwards with a gradient of $1\,\mu\mathrm{rad}$ per $10\,\mathrm{km}$ (an experience based best guess; cf. Scherllin-Pirscher et al. (2011b, a), who also address aspects of such tropospheric

extrapolation in their discussions of error sources). It is interpreted as an apparent systematic uncertainty estimate, since due to the linear fit-based TBAE construction its event-to-event bias character will be essentially random (Scherllin-Pirscher et al., 2011b).

Also, the ionospheric correction currently applied in the rOPS is just a first-order correction, which will leave higher-order residual ionospheric errors in $\alpha_r$ (e.g., Syndergaard, 2000; Danzer et al., 2013; Liu et al., 2013, 2015; Healy and Culverwell,



2015). The uncertainty from higher-order *residual ionospheric biases* (RIB), $u_{\mathrm{RIB}}^s$, is therefore added to the propagated (basic) systematic uncertainty. $u_{\mathrm{RIB}}^s$ is interpreted as basic systematic uncertainty, since the higher-order ionospheric residuals may not vanish in ensemble-of-events averaging. The other non-ionospheric sources of systematic errors and the RIBs can be reasonably assumed to be uncorrelated. The total estimated systematic uncertainty of the retrieved atmospheric bending angle $\alpha_{\mathrm{r}}$ hence is

$$u_{\alpha\mathrm{r},i}^s := \sqrt{(u_{\alpha\mathrm{r},i}^s)^2 + (u_{\mathrm{RIB}}^s)^2}. \tag{A51}$$

Based on previous studies (e.g., Liu et al., 2013, 2015; Danzer et al., 2013, 2015), $u_{\mathrm{RIB}}^s$ is taken to be constant along the entire profile, and is estimated to amount to $0.05\,\mu\mathrm{rad}$. These last two components, $u_{\alpha2\mathrm{TE}}^s$ and $u_{\mathrm{RIB}}^s$, are indicated as Item 3.7 in Figure 2. It is clear that this initial systematic uncertainty estimation can be significantly improved by future dedicated work on better quantifying and (if suitable) correcting for the systematic uncertainty components.

The *resolution* of the retrieved bending angle, $w_{\alpha\mathrm{r}}$, essentially corresponds to the higher resolution of the two bending angle profiles $\alpha_{\mathrm{F1}}$ and $\alpha_{\mathrm{F2}}$, and thus generally closely matches $w_{\alpha\mathrm{F1}}$ in most cases. As a simple but robust and suitable estimate, assuming that the resolutions of $\alpha_{\mathrm{r}}$ and $\alpha_{\mathrm{F1}}$ scale in the same way as the correlation lengths $l_{\alpha\mathrm{r}}$ and $l_{\alpha\mathrm{F1}}$ (derived from $\mathbf{R}_{\alpha\mathrm{r}}$ and $\mathbf{R}_{\alpha\mathrm{F1}}$ as described for $\mathbf{R}_{LF}$ in Equation A9), we compute $w_{\alpha\mathrm{r}}$ as

$$w_{\alpha\mathrm{r},i} = \frac{l_{\alpha\mathrm{r},i}}{l_{\alpha\mathrm{F1},i}} \cdot w_{\alpha\mathrm{F1},i}. \tag{A52}$$

In concluding we note that the atmospheric bending angle derivation algorithms used in this study, i.e., the adaptive filtering, TBAE, and ionospheric correction parts as described in this section, have recently received further advancement towards a form fully based on the combination of $\alpha_{\mathrm{F1}}$ and the difference-profile $\alpha_{\mathrm{F12}}$ (rather than of $\alpha_{\mathrm{F1}}$ and $\alpha_{\mathrm{F2}}$), more aligned with the concept of Sokolovskiy et al. (2009). A detailed description of this most recent version is found in Kirchengast et al. (2017b).

## Appendix B: Variance Propagation for Comparison

The full covariance propagation applied to propagate random uncertainties requires numerically 'expensive' matrix operations and therefore considerable efforts were made to seize opportunities for reducing the number of numerical operations (e.g., by only calculating with those elements of the band-matrix $\mathbf{A}^{\mathrm{BWS}}$ which lie within the width of the filter window).

However, as demonstrated in Section 4, simplification to a mere variance propagation (i.e., only considering the diagonal elements of the covariance matrices) is not reasonably possible because it leads to an unacceptable overestimation of random uncertainties. This overestimation occurs since the influence of the covariance elements—and thus for example the partially compensating impact of the negative side-peaks in the correlation functions—is disregarded.

Here we state the two equations used to obtain the variances-only propagation results shown for comparison purposes in Figure 9: the estimated random uncertainty was propagated through the BWS filter using

$$(u_{\alpha\mathrm{F}i,k}^r)^2 = \sum_{j=0}^{N} (A_{ij,k}^{\mathrm{BWS}})^2 \cdot (u_{\alpha\mathrm{M}j,k}^r)^2, \tag{B1}$$




and subsequently through the ionospheric correction using

$$(u_{\alpha r,i}^{r})^2 = (1 + \gamma_{f\text{T}12})^2 \cdot (u_{\alpha\text{F}1,i}^{r})^2 + \gamma_{f\text{T}12}^2 \cdot (u_{\alpha\text{F}2,i}^{r})^2. \tag{B2}$$

*Author contributions.* Jakob Schwarz and Gottfried Kirchengast designed the study, including comments by Marc Schwaerz, based on the concept and framework of uncertainty propagation into the reference occultation processing system (rOPS) formulated by Gottfried
5    Kirchengast. Jakob Schwarz elaborated the detailed algorithms, performed the computational implementation and the analysis, prepared the figures, and wrote the first draft of the paper, advised and supported in this work by Gottfried Kirchengast and Marc Schwaerz. As part of this support Gottfried Kirchengast provided detailed design input and feedback and substantially contributed to the writing of the paper and Marc Schwaerz substantially contributed to the computational work. Based on a primary role of the first two authors, all authors contributed to consolidating the paper for submission and towards publication.

10    *Competing interests.* The authors declare that they have no conflicts of interest.

*Acknowledgements.* We thank UCAR/CDAAC Boulder for access to their RO excess phase and orbit data (available at http://cdaac-www. cosmic.ucar.edu/) as well as ECMWF Reading for access to their analysis and forecast data (available at http://www.ecmwf.int/en/forecasts/ datasets). To access the relevant result files of the uncertainty propagation, please contact the corresponding author. The developed algorithm is provided in the appendix. The work was funded by the Austrian Aeronautics and Space Agency of the Austrian Research Promotion Agency
15    (FFG-ALR) under projects OPSCLIMPROP and OPSCLIMTRACE and by the European Space Agency (ESA) under project MMValRO-E.





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





**Table 1.** Principal variables for the rOPS L1b uncertainty propagation.

| Variable | Unit | Description |
|---|---|---|
| $X_{\mathrm{r}}$ | U | state profile of retrieved excess phase/filtered excess phase/retrieved Doppler shift/retrieved geometric optics bending angle/merged GO WO bending angle/filtered bending angle/retrieved bending angle, with $X_{\mathrm{r}} \in \{L_{\mathrm{r},k}(t), L_{\mathrm{F},k}(t), D_{\mathrm{r},k}(t), \alpha_{\mathrm{G},k}(t), \alpha_{\mathrm{G},k}(z_a), \alpha_{\mathrm{M},k}(z_a), \alpha_{\mathrm{F},k}(z_a), \alpha_{\mathrm{r}}(z_a)\}$, $k \in \{1,2\}$ (frequencies $f_{\mathrm{T1}}$, $f_{\mathrm{T2}}$) and unit $\mathrm{U} \in \{\mathrm{m}, \mathrm{m}, \mathrm{ms}^{-1}, \mathrm{rad}, \mathrm{rad}, \mathrm{rad}, \mathrm{rad}, \mathrm{rad}\}$, comprising elements $X_{\mathrm{r},i}$. |
| $u_X^s$ | U | estimated systematic uncertainty profile of $X$ (with $X$ and U as defined above), comprising elements $u_{X,i}^s$ (including estimated basic and estimated apparent systematic uncertainties, $u_{X,i}^b$ and $u_{X,i}^a$). |
| $u_X^r$ | U | estimated random uncertainty profile of $X$ (with $X$ and U as defined above), comprising elements $u_{X,i}^r$, . |
| $\mathbf{R}_X$ | 1 | error correlation matrix of $X$ (with $X$ as defined above), comprising elements $R_{X,ij}$. |
| $\mathbf{C}_X$ | $\mathrm{U}^2$ | error covariance matrix of $X$ (with $X$ and U as defined above), comprising elements $C_{X,ij} = u_{X,i}^r \cdot u_{X,j}^r \cdot R_{X,ij}$. |
| $l_X$ | m | correlation length profile of $X$ (with $X$ as defined above), comprising elements $l_{X,i}$. |
| $\tau_X$ | s | resolution profile of $X$ (with $X$ as defined above) in time domain, comprising elements $\tau_{X,i}$. |
| $w_X$ | m | resolution profile of $X$ (with $X$ as defined above) in altitude domain (along impact altitude), comprising elements $w_{X,i}$. |
| $X_{\mathrm{m}}$ | U | model excess phase/Doppler shift/bending angle profiles based on forward modeling of co-located refractivity profiles from ECMWF short-range forecast fields, with $X_{\mathrm{m}} \in \{L_{\mathrm{m}}(t), D_{\mathrm{m}}(t), \alpha_{\mathrm{m}}(z_a)\}$ and $\mathrm{U} \in \{\mathrm{m}, \mathrm{ms}^{-1}, \mathrm{rad}\}$, comprising elements $X_{\mathrm{m},i}$. |
| $\boldsymbol{x}_{\mathrm{S}}$ | U | profiles of cartesian position/velocity vectors of the receiving/transmitting satellite relative to the center of refraction, with $\boldsymbol{x}_{\mathrm{S}} \in \{\boldsymbol{r}_{\mathrm{T}}(t), \boldsymbol{r}_{\mathrm{R}}(t), \boldsymbol{v}_{\mathrm{T}}(t), \boldsymbol{v}_{\mathrm{R}}(t)\}$ and unit $\mathrm{U} \in \{\mathrm{m}, \mathrm{m}, \mathrm{ms}^{-1}, \mathrm{ms}^{-1}\}$, comprising elements $\boldsymbol{x}_{\mathrm{S},i}$. |
| $u_{\boldsymbol{x}_{\mathrm{S}}}^s$ | U | estimated (systematic) uncertainty profiles of $\boldsymbol{x}_{\mathrm{S}}$ (with $\boldsymbol{x}_{\mathrm{S}}$ and U as defined above), comprising elements $u_{\boldsymbol{x}_{\mathrm{S}},i}^s$. |
| $\mathbf{A}^{\mathrm{BWS}}$ | 1 | BWS filter matrix operator, comprising the blackman windowed-sinc (BWS) lowpass filter weights (normalized filter functions) in form of a band matrix. |
| $\mathbf{A}^{\mathrm{L2D}}$ | $\mathrm{s}^{-1}$ | Doppler differentiation matrix operator, transforming the filtered excess phase profile to the Doppler shift profile. |





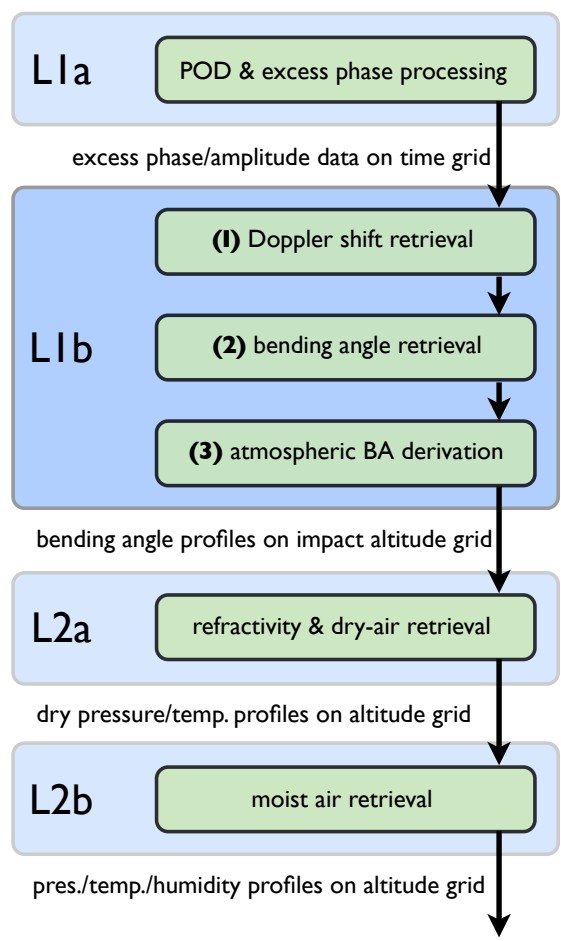

**Figure 1.** Schematic view of the main processors of the retrieval chain in the rOPS (L1a, L1b highlighted, L2a, L2b) and the main operators of the L1b processor (1, 2, 3), which are in the focus of this study.



**Figure 2.** Detailed workflow for state retrieval and uncertainty propagation of the main L1b operators from excess phase to atmospheric bending angle profiles (1)-(3) and of the subroutines used in the MC testing framework (a)-(g). The mathematical notation, including all symbols, is introduced in Tables 1 and 2.





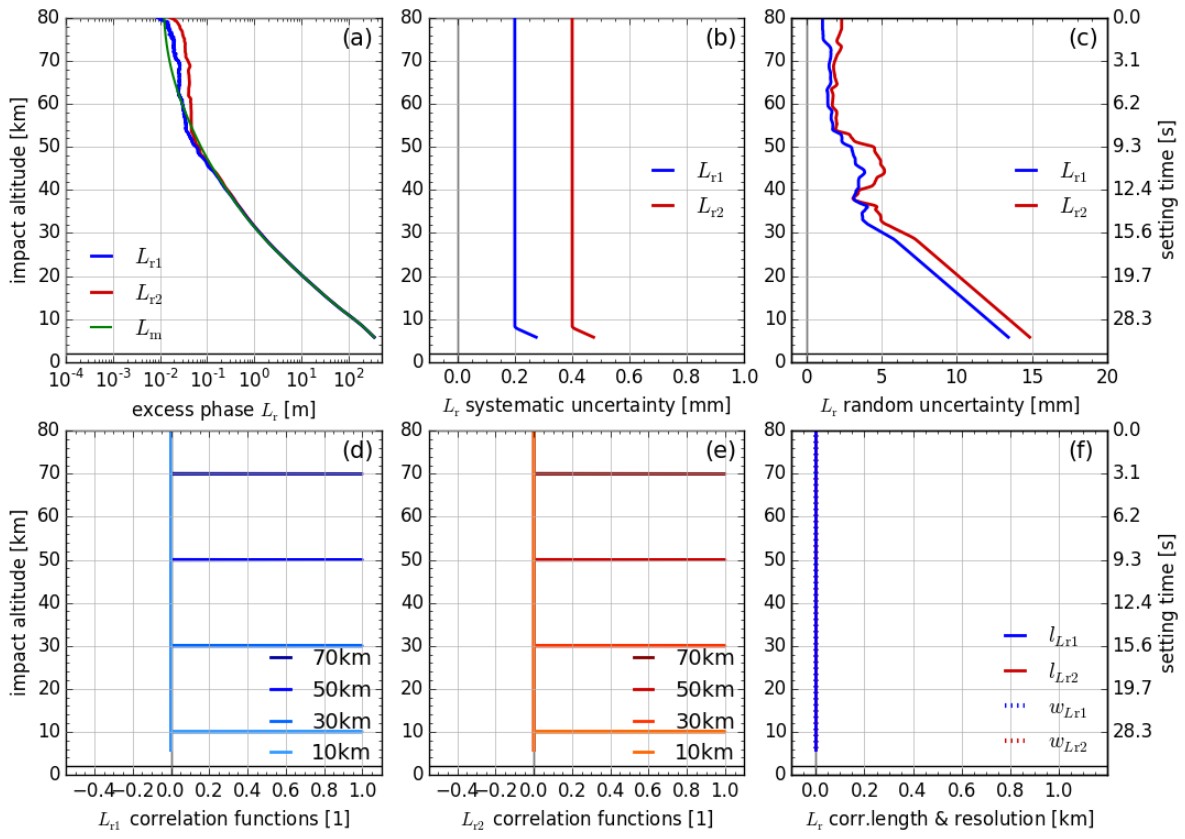

**Figure 3.** Input profiles of retrieved excess phase $L_r$ (with model profile $L_m$ for comparison) in (a), estimated systematic uncertainty profiles $u_{Lr}^s$ in (b), estimated random uncertainty profiles $u_{Lr}^r$ in (c), representative correlation functions $R_{Lr,i}$ (at 10, 30, 50 and 70 km) in (d) and (e), and correlation length $l_{Lr}$ (solid) and resolution profiles $w_{Lr}$ (dotted) in (f), which are set zero for these initial essentially uncorrelated input data. All profiles are shown for both GPS carrier frequencies $f_{T1}$ (blue) and $f_{T2}$ (red).



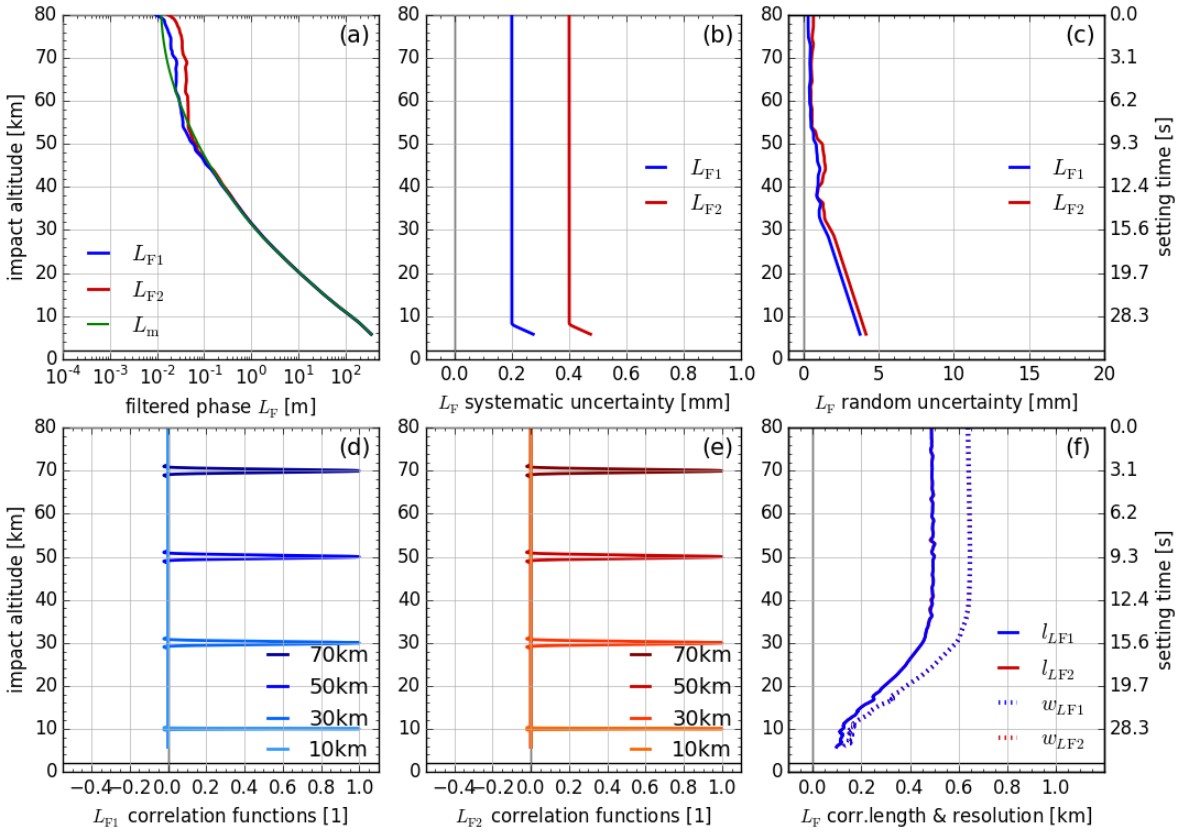

**Figure 4.** Results for filtered excess phase profiles $L_F$ (with model profile $L_m$ for comparison) in (a), estimated systematic uncertainty profiles $u^s_{LF}$ in (b), estimated random uncertainty profiles $u^r_{LF}$ in (c), representative correlation functions $R_{LF,i}$ (at 10, 30, 50 and 70 km) in (d) and (e), and correlation length $l_{LF}$ (solid) and resolution profiles $w_{LF}$ (dotted) in (f). All profiles are shown for both GPS carrier frequencies $f_{T1}$ (blue) and $f_{T2}$ (red).





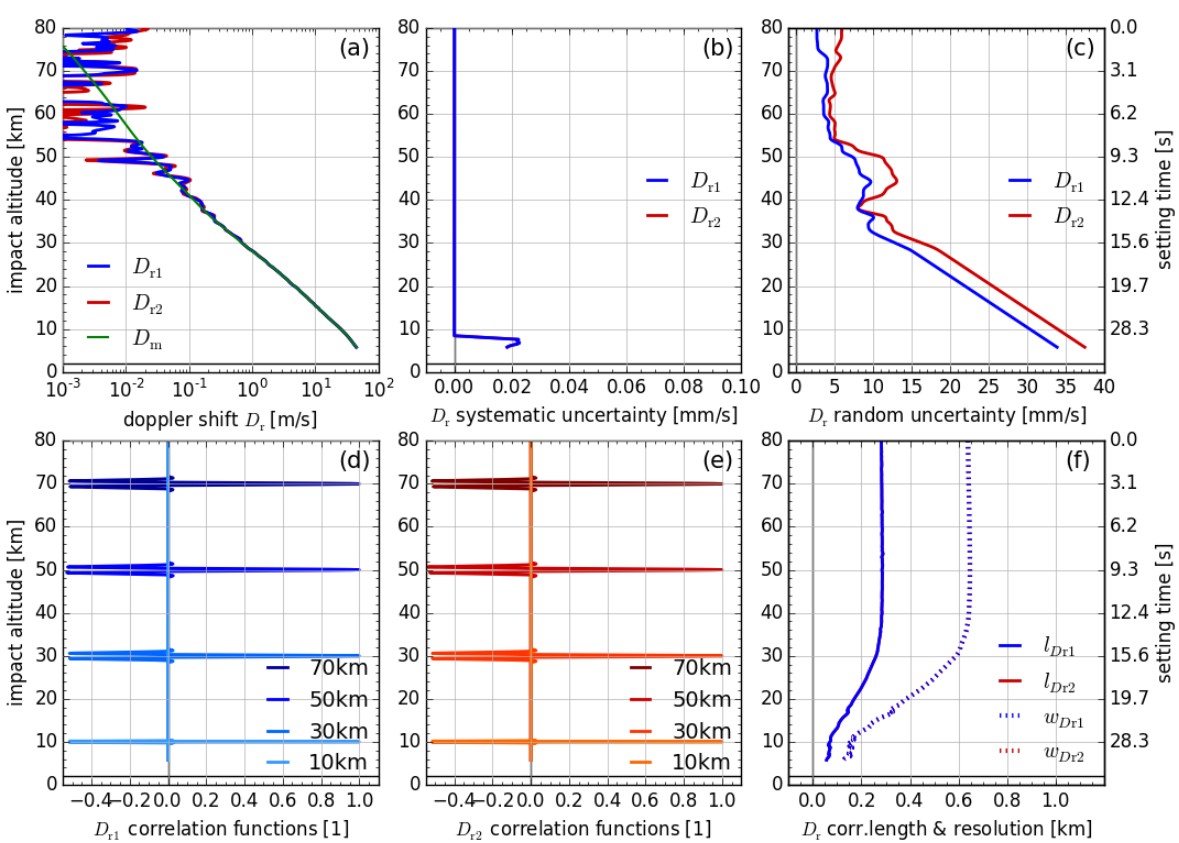

**Figure 5.** Results for retrieved Doppler shift profiles $D_r$ (with model profile $D_m$ for comparison) in (a), estimated systematic uncertainty profiles $u_{Dr}^s$ in (b), estimated random uncertainty profiles $u_{Dr}^r$ in (c), representative correlation functions $R_{Dr,i}$ (at 10, 30, 50 and 70 km) in (d) and (e), and correlation length $l_{Dr}$ (solid) and resolution profiles $w_{Dr}$ (dotted, estimated for main peak) in (f). All profiles are shown for both GPS carrier frequencies $f_{T1}$ (blue) and $f_{T2}$ (red).





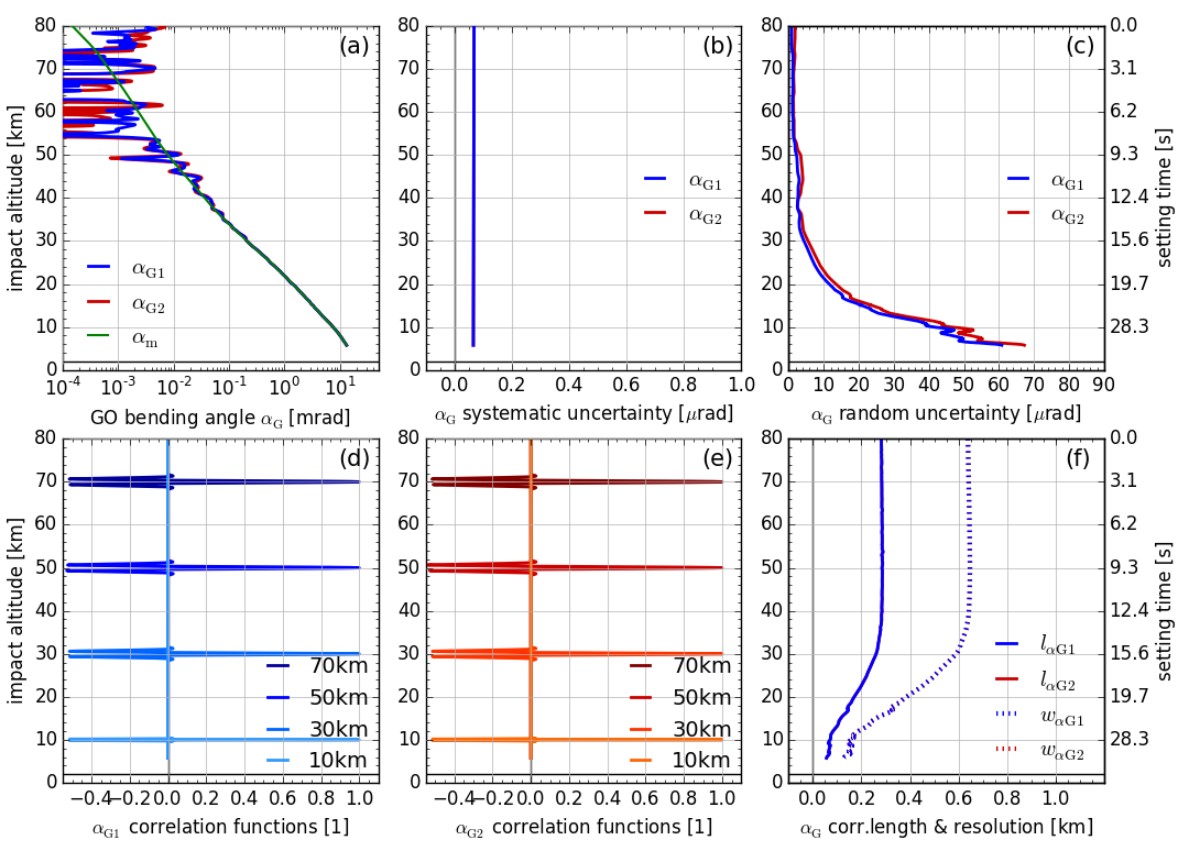

**Figure 6.** Results for geometric optics bending angle profiles $\alpha_G$ (with model profile $\alpha_m$ for comparison) in (a), estimated systematic uncertainty profiles $u^s_{\alpha G}$ in (b), estimated random uncertainty profiles $u^r_{\alpha G}$ in (c), representative correlation functions $R_{\alpha G,i}$ (at 10, 30, 50 and 70 km) in (d) and (e), and correlation length $l_{\alpha G}$ (solid) and resolution profile $w_{\alpha G}$ (dotted, estimated for main peak) in (f). All profiles are shown for both GPS carrier frequencies $f_{T1}$ (blue) and $f_{T2}$ (red); in panels (b) and (f) both profiles are essentially identical (so that blue shadows the red color).




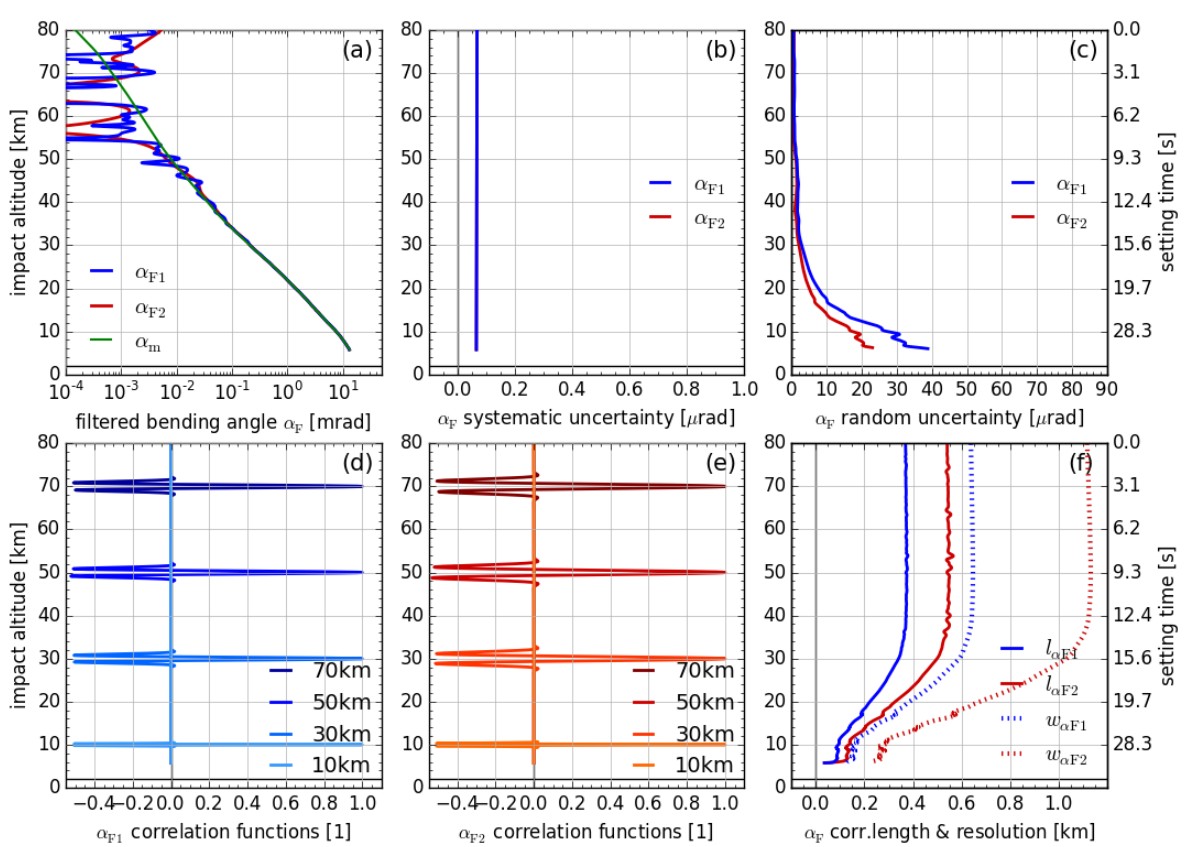

**Figure 7.** Results for filtered bending angle profiles $\alpha_F$ (with model profile $\alpha_m$ for comparison) in (a), systematic uncertainty profiles $u^s_{\alpha F}$ in (b), random uncertainty profiles $u^r_{\alpha F}$ in (c), representative correlation functions $R_{\alpha F,i}$ (at 10, 30, 50 and 70 km) in (d) and (e), and correlation length $l_{\alpha F}$ (solid) and resolution profiles $w_{\alpha F}$ (dotted, estimated for main peak) in (f). All profiles are shown for both GPS carrier frequencies $f_{T1}$ (blue) and $f_{T2}$ (red).





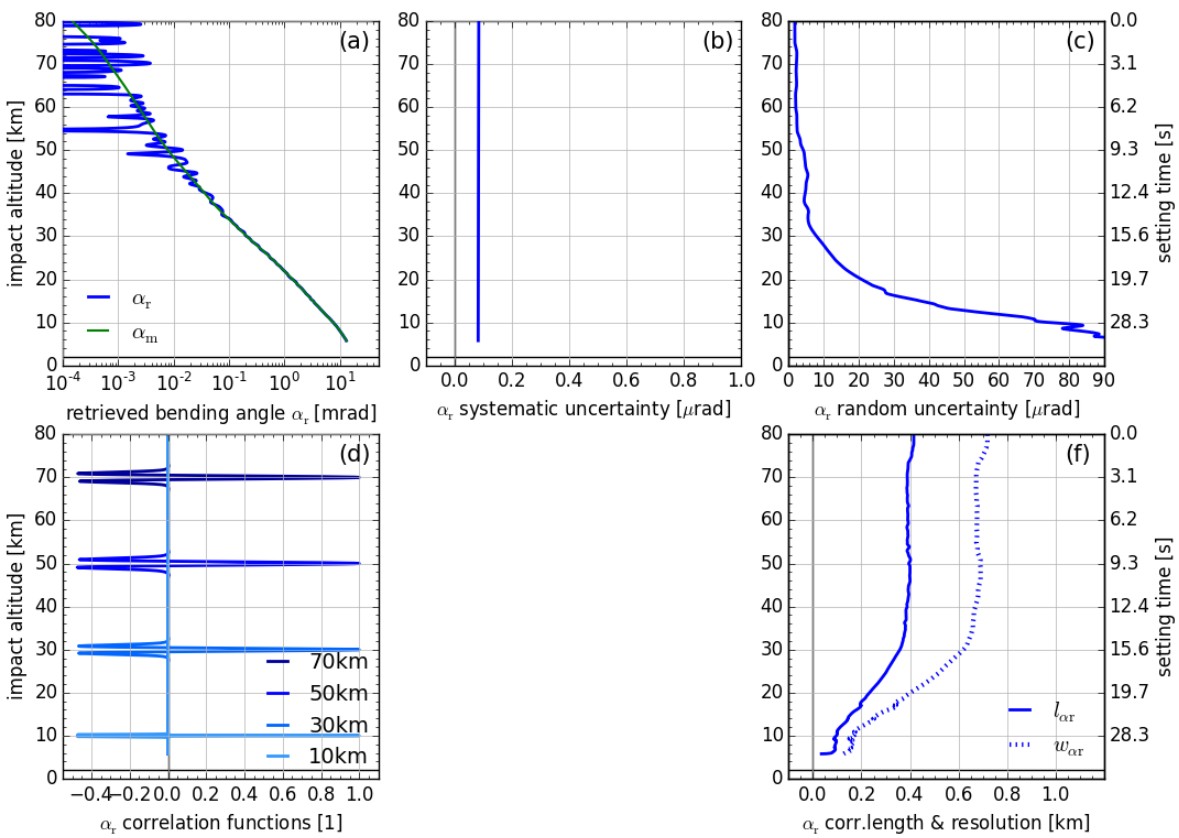

**Figure 8.** Results for atmospheric bending angle profile $\alpha_r$ (with model profile $\alpha_m$ for comparison) in (a), systematic uncertainty profile $u_{\alpha r}^s$ in (b), random uncertainty profile $u_{\alpha r}^r$ in (c), representative correlation functions $R_{\alpha r,i}$ (at 10, 30, 50 and 70 km) in (d), and correlation length $l_{\alpha r}$ (solid) and resolution profile $w_{\alpha r}$ (dotted, estimated for main peak) in (f).







**Figure 9.** Results from the validation of CP covariance matrices $\mathbf{C}_{Lr}^{CP}$ ('CP') by MC covariance matrices $\mathbf{C}_{Lr}^{MC}$ ('MC') (first four rows): The consecutive retrieval steps are shown for $L_F$ (a-c) and $D_r$ (d-f) relative to setting time $t$, and for $\alpha_G$ (g-i) and $\alpha_F$ (j-l) relative to impact altitude $z_a$. The left column shows estimated random uncertainties for $f_{T1}$ (CP in blue, MC in black), the middle column for $f_{T2}$ (CP in red, MC in black), the right column representative correlation functions at 60, 36, 12 km for $f_{T1}$ (CP in blue, MC in black), and 72, 48, 24 km for $f_{T2}$ (CP in red, MC in black). The last row (m-o) shows CP (blue) and MC (black) results for estimated $\alpha_r$ random uncertainties (m) and representative correlation functions at 72, 60, 48, 36, 24 and 13 km (o), as well as variance propagation ('VP') results (blue) for $\alpha_r$ in (n).



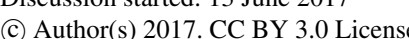

**Figure 10.** Uncertainty propagation results for real data ensembles from July $15^{\text{th}}$ 2008, for the filtered excess phase profile $L_{\text{F1}}$ of the leading channel ($f_{\text{T1}}$, GPS L1 frequency). Left column: Estimated random $u^r_{LF1}$ (heavy) and systematic $u^s_{LF1}$ (light) uncertainty profiles of each ensemble member (gray), and the ensemble mean (color) for CHAMP (a), COSMIC (d) and simMetOp (g). Middle column: Correlation length profiles $l_{LF1}$ of each ensemble member (gray), the ensemble mean (color) and the ensemble size profile (black, scale at upper axis) for CHAMP (b), COSMIC (e) and simMetOp (h). Right column: Estimated resolution profile $w_{LF1}$ of each ensemble member (gray) and the ensemble mean (color) for CHAMP (c), COSMIC (f) and simMetOp (i).





**Figure 11.** Uncertainty propagation results for real data ensembles from July $15^{\text{th}}$ 2008 for output profiles of the leading channel ($f_{\text{T1}}$, GPS L1 frequency). The first row shows results for $L_{\text{F1}}$ (a-c), the second for $D_{\text{r1}}$ (d-f), the third for $\alpha_{\text{G1}}$ (g-i), and the fourth for $\alpha_{\text{r}}$ (j-l). The different ensemble mean profiles are shown in colors (CHAMP (yellow), COSMIC (orange) and simMetOp (red)). Left column: Mean random uncertainty $u_{X\text{r}}^{r}$ (heavy) and mean systematic uncertainty $u_{X\text{r}}^{s}$ (light) profiles (panels a, d, g, j); the latter shown as $10 \times u_{L\text{F1}}^{s}$ (in a) and $100 \times u_{X\text{r}}^{s}$ (in d, g, j) for enabling visibility of these small quantities. Middle column: Correlation length profiles $l_{X\text{r}}$ (panels b, e, h, k). Right column: Vertical resolution profiles $w_{X\text{r}}$ (panels c, f, i, l).





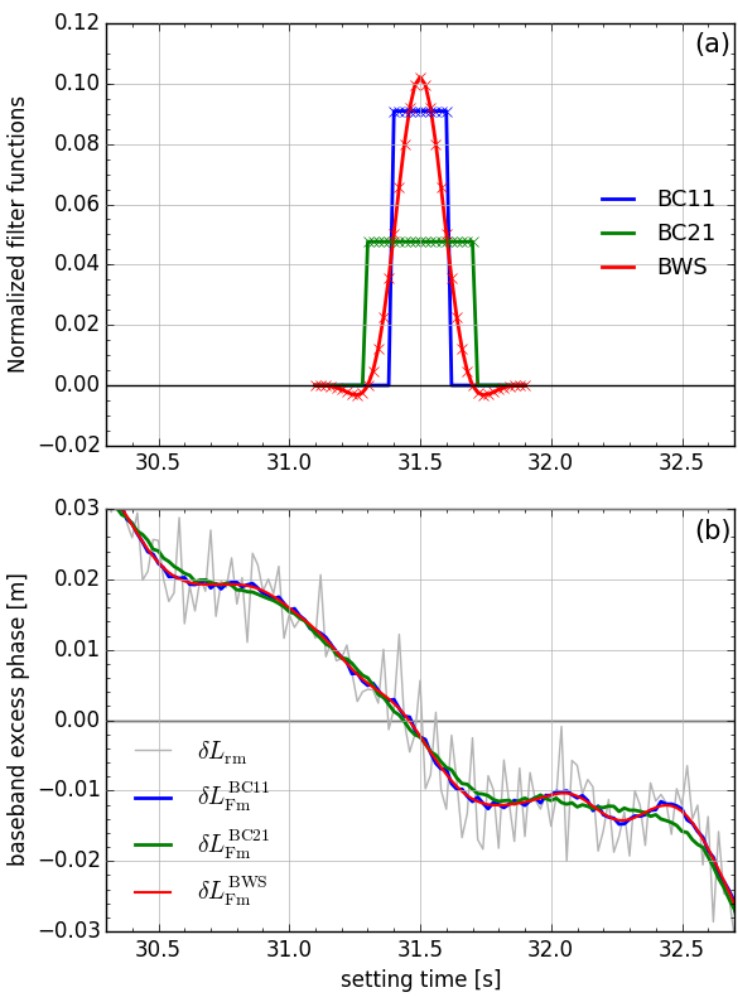

**Figure A1.** Comparison of the Blackman windowed-sinc (BWS) lowpass filter and boxcar (BC) filters based on a representative segment (between 30.3 and 32.7 s) of the excess phase profile $L_{r1}$ of the COSMIC example event. Panel (a): Filter functions for the BWS filter with $f_c = 2.5\,\mathrm{Hz}$ and $M = 41\,\mathrm{pts}$ ('BWS', red) and boxcar filters with $M = 21\,\mathrm{pts}$ ('BC21', green) and with $M = 11\,\mathrm{pts}$ ('BC11', blue), around the central value of the segment (31.5 s). Panel (b): Filter effects on the excess phase profile $L_{r1}$ from running the filters over the segment. Shown are the unfiltered excess phase delta profile ('$\delta L_{rm}$', light gray), the BWS filtered profile with $f_c = 2.5\,\mathrm{Hz}$ and $M = 41\,\mathrm{pts}$ ('$\delta L_{Fm}^{BWS}$', red), and the Boxcar filtered profiles with $M = 21\,\mathrm{pts}$ ('$\delta L_{Fm}^{BC21}$', green) and $M = 11\,\mathrm{pts}$ ('$\delta L_{Fm}^{BC11}$', blue), respectively.





**Table 2.** Vertical grids, coordinate variables, and specific settings for the rOPS L1b processing system

| Variable | Unit | Description |
|---|---|---|
| $f_{\mathrm{T}}$ | Hz | transmitter signal carrier frequency, with elements $f_{\mathrm{T}k}$ (for GPS transmitters $k \in \{1,2\}$ denoting the L-band frequencies $f_{\mathrm{T}1} = 1.57542\,\mathrm{GHz}$ and $f_{\mathrm{T}2} = 1.22760\,\mathrm{GHz}$). |
| $f_s$ | Hz | measurement sampling frequency (also called sampling rate); $50\,\mathrm{Hz}$ is generally used for the input excess phase profiles. |
| $f_{\mathrm{c}}$ | Hz | Blackman Windowed-Sinc (BWS) lowpass filter cutoff-frequency; set to $2.5\,\mathrm{Hz}$ (but noise-dependent for the $f_{\mathrm{T}(1)2}$ filtering for ionospheric correction, with $f_{\mathrm{c}(1)2} \in \{2.5\,\mathrm{Hz}, 2\,\mathrm{Hz}, 10/7\,\mathrm{Hz}, 1\,\mathrm{Hz}, 5/7\,\mathrm{Hz}, 0.5\,\mathrm{Hz}\}$). |
| $t$ | s | time grid of the measurements at sampling rate $f_s$, with elements $t_i$, $i \in \{1, 2, ...N\}$, where $N$ is the number of grid points of the RO profile. |
| $a_t$ | m | impact parameter grid corresponding to time grid $t$. |
| $z_a$ | m | common monotonic impact altitude grid, calculated from sorted impact parameters $a_{t,i}$ of the leading channel ($f_{\mathrm{T}1}$) bending angle, via $z_{a,i} = a_{t,i} - h_{\mathrm{G}} - R_{\mathrm{C}}$. Used as standard vertical grid after interpolation of all dependent quantities to $z_a$. |
| $z_t$ | m | msl altitude grid corresponding to time grid $t$, obtained as part of the forward modeling towards $\alpha_m$, $D_m$, and $L_m$ (cf. Table 1). |
| $z_{a\,\mathrm{Top}}$ | m | impact altitude of the top of the RO profile, can lie between $70\,\mathrm{km}$ and $80\,\mathrm{km}$. |
| $z_{a\,\mathrm{Bot}}$ | m | impact altitude of the bottom of the RO profile, can lie between $25\,\mathrm{km}$ and the Earth's surface. It can be different for the different GNSS frequencies (i.e., $z_{a\,\mathrm{Bot},k}$, for $k \in \{1,2\}$). |
| $z_a^{\mathrm{GW}}$ | m | impact altitude at the center of the sinusoidal transition range of half-width $\Delta z_a^{\mathrm{GW}}$ between the GO and WO bending angle profiles; $z_a^{\mathrm{GW}}$ can lie within $9\,\mathrm{km}$ and $14\,\mathrm{km}$, depending on GO bending angle data quality. |
| $\Delta z_a^{\mathrm{GW}}$ | m | impact altitude transition half-width of the half-sine-weighted transition between the GO and WO bending angle profile. Set to $2\,\mathrm{km}$. |
| $z_{a\,\mathrm{Gradr}}$ | m | impact altitude at the lower end of the excess phase uncertainty estimation range used in this study, below which the estimated random uncertainties are extended by a linear gradient. Set to $30\,\mathrm{km}$. |
| $z_{a\,\mathrm{Grads}}$ | m | impact altitude at the lower end of the range with constant excess phase systematic uncertainty used in this study, below which the estimated systematic uncertainties continue with a linear gradient. Set to $8\,\mathrm{km}$. |