# Peer review of "Integrating uncertainty propagation in GNSS radio occultation retrieval: from excess phase to atmospheric bending angle profiles"

_Atmospheric Measurement Techniques, 2017_

## Referee Comment (RC1) · Anonymous Referee #1 · 11 Jul 2017

General comments

The paper details the algorithms, the underlying assumptions and the results of the uncertainties estimation (random and systematic) in the GNSS radio occultation (RO) retrieval, from the excess-phases to bending angle profiles. The paper, together with other similar papers which address the same topic once applied to the other portion of the entire RO processing chain (thus describing error propagation through the POD solution and through the retrieval steps of atmospheric profiles starting from bending angles), provided for the first time a rigorous mathematical overview of how errors are propagated through the entire RO processing chain. From the scientific point of view

the paper is almost complete; involved data, algorithms, results, discussion and conclusions are presented in a well structured and clear way, even if the paper is excessively long.

In what follows some general comments are provided.

1. One of the most critical step of the RO processing chain is the retrieval of bending angles in the lower troposphere, which involves more "advanced" methods (the so called Wave Optic retrieval). Unfortunately error propagation through this step is not detailed (even if a paper is already available since 2015, it is written a couple of times in the manuscript that the implementation of WO uncertainty propagation in the rOPS is not yet finished... so it is not fully clear whether the results are already available or not). In order to have a complete and more coherent overview, this part is somehow necessary. My suggestion is, if results are not available yet, to remove from this paper all the discussions provided on the WO retrieval and error propagation through it (Sect 3.2.2, Sect 3.2.3, Annex A2.2, Annex A2.3), changing a bit also the title reflecting that only GO retrieval is applied.

2. Another general issue I want to highlight is that the entire discussion and the main results shown from Fig 3 to Fig 8 are based considering an excess-phases "time series" and associated random/systematic uncertainties observed during one real occultation event from the COSMIC mission, taken as representative for the entire day. Not sure that the heuristic associated to the choice of that particular profile is correct, since the ensemble mean profiles of the uncertainties associated to the various variables involved in the retrieval (Fig 10 and Fig 11) show quite different behavior (see for example the increasing of uncertainties with height). It would be nice to have different examples (maybe the best and the wrong together with this most representative one).

3. Regarding the same input profile I found a bit inconsistent the choice of extrapolating downward 30 km the representative excess-phase random uncertainty profile (defined on the above mentioned COSMIC observation) with a linear behavior (I guess linear in

order to provide a "worst" boundary, could you confirm this in the paper?), whose gradient follows some estimate defined for the GRAS instrument on board METOP satellites (and I don't think that in the provided reference [ESA/EUMETSAT] such information can be easily found). The systematic uncertainty on the same input variable is also defined following error estimates characterizing the GRAS receiver. The big question is why you mixed the random uncertainty "rigorously" derived considering a COSMIC observations with some heuristic based on the GRAS instrument? Was not better to derive the random error profile from GRAS real observations? And why for the GRAS case you demonstrated the uncertainty propagation using simulated data only? Please provide a clear motivation on this or repeat the entire analysis using some representative examples taken by GRAS excess-phases.

4. And, always on the definition of the input random uncertainty example profile, does it takes into account the merging of open loop and closed loop data somewhere in the lower part of the profile? I guess that the uncertainties related to the open loop and closed loop tracking should be different (because of different noise levels added by the different tracking behavior, because of different sampling rates between OL and CL, which is different from COSMIC and GRAS for example). Maybe the "boundary" set with the linear extrapolation downward the 30 km includes already these different uncertainty levels. But I expect that a discussion on these aspect will be added in the paper.

5. All the analysis is based on the covariance matrix propagation through linear (or linearized) operators. This is fine. But what are the residuals effects due to the linearization? I guess that the most "critical" linearization is the one applied to impact parameter retrieval (in GO). In that case I guess you can quantify the second/higher order effects by considering more terms in the Taylor's expansion of the operator.

6. Finally, the most important references related the main topic of the paper (the description of rOPS and of the algorithms involved) are presentations to conferences or not accessible technical reports. This is a pity. But at least, for the presentations it

would be nice to have the full web-link specified.

The paper is absolutely worth to be published, but some review/clarifications should be provided. That's why I suggest for a major review.

Specific and technical comments are provided in the next two sections.

Specific comments

**6: what does it mean that "the accuracy is also ensured on-orbit"? please specify**

**21-22: remove references to unpublished papers or to papers in preparation. It is already mentioned that work is on-going.**

**31: first mention of the use of simulated data for GRAS.**

2.1 Methods: all the analysis is based on evaluating random/systematic uncertainties as complete independent processes. Could you please explain why the two are completely decoupled? This is probably trivial under the assumption that noise is normally distributed. But is this really the case starting from excess-phases?

**9-10: not clear why the effects of temporal/spatial variations (which kind of variations have you in mind?) in repeated observations can be estimated based on an individual RO profile. This is one of the main issue. Describe/motivate this better. And, by the way, why this should be true based to oversampling in the RO raw data profile?**

**29: what do you mean with ionospheric noise? Here you are describing the noise in excess-phase measurements, so on both L1 and L2. Are you referring to scintillations? I don't think these can be treated as "normally distributed noise". Define this better.**

**29: Why random errors on positions/velocity are not considered here (later on, page**

**30 you introduce such effects within the systematic uncertainties)?**

**24-25: not clear what "is aligned to a joint resolution..." mean. Please rephrase/clarify.**

**7: It would be nice to motivate clearly here why you are using simulated data for propagating the uncertainties for the GRAS case.**

**13-14: clarify why the "baseband" approach allows to avoid biases from numerical operations on near exponentially varying profiles (also repeated at page 17, #11-12).**

**4-7: here is where the downward extrapolation based on a GRAS "model" is described. See point 3 of General Comments.**

**4-7: why linearly extrapolating downward and constantly extending upward a "random" uncertainty profile does not introduce a systematic effect?**

**8: Please clarify further why all the noise components responsible for random uncertainty at excess-phase level are uncorrelated (also referring to the ionospheric noise). I believe that at carrier phase level noise is uncorrelated (we have basically phase noise). But when geometry is removed as well as clock biases, and when the result is interpolated to match 50 Hz sampling rate (maybe between open and closed loop observations), I'm not sure that the various components are still uncorrelated.**

**14: Being he entire analysis based on a COSMIC profile, why for the MC validation you started from a simulated CHAMP excess-phase profile and not from a realistic COSMIC profile? Motivate it.**

**24: in defining the input random uncertainty profile, you are computing the excess**

phase mean value between 60-70 km. Is the constant part of the systematic uncertainty consistent with such a mean?

**Eq 6 and 8: Explain why the "gradient" (1/3exp(xx)) for the random and the systematic uncertainties below zGrad is different.**

**20-21: not clear the sentence. By the way, it would be nice to show the relative error (to the state profile) instead the absolute one.**

**25: What is written there is true, but this is not clear whether these results (and the merging with bending angles uncertainties derived by GO processing) are available or not (#3-5 p 11 it is clearly stated that their integration in rOPS is not yet finished). I suggest to remove any reference to uncertainty propagation through WO/merging unless the results can be shown. See general comment 1).**

**6-8: this is definitely not clear. How you can "artificially" substitute WO with GO uncertainty propagation, being the vertical domain of their applicability different?**

**17 and 20: Not clear why the low-pass filtering of alphaM1 provides so different random uncertainty levels and broaden the correlation functions if the cut-off frequency is the same of the one already used to filter the corresponding excess-phases. I expected negligible or a smaller effect.**

**23: I found inconsistent defining the input data for the MC simulations by using a "background" error-free excess-phase profile taken by a simulated CHAMP event (mentioned not here but at page 7, #14) plus error realizations taken by a COSMIC dataset. The description of this input dataset should be better motivated and clarified.**

Moreover the input for the covariance matrix (for the MC simulation) seems to be some standard profile uSTD,Lr not well identified.

**28: Here you are saying that the comparisons between MC simulations and covariance matrix propagation (CP) are carried out in terms of their decomposition (error profiles and correlation functions/lengths). This is fine but it is not clear where the CP-propagated covariance matrix comes from. What is the input? The same COSMIC profile taken as example in the previous discussion? Another excess-phase profile (more likely yes, since the results shown in Fig 9 [red, blue plots] are bit different from the same results shown in the previous figures. In #19, p7, you mentioned that "the same standard profile was used as input for the CP"). Rephrase and clarify the entire introduction to Set. 4.**

**7: the information provided in Fig 9 are not so clear. First of all are the random error profiles taken to lye between the boundaries provided by blue/orange lines used to plot Lr (thus varying up to 20 mm)? Could you use other styles to plot such information (dotted lines or different colors). Why the area within the LF random uncertainty is colored?**

**13-14: why it is "obvious" that the CP delivers the correct off-peak results? Provide a clarification.**

**17-21: rephrase a bit the sentence, since it is not clear.**

**9: Are outliers defined in term of individual random uncertainty profiles? Or input Excess-phases profiles?**

**7: this reference is a technical report, not accessible. Since it is quite important for**

the main topic of the paper, please provide an open reference if available (or avoid to cite e technical report which is not public).

Eq. A2, #14: The baseband approach allows the application of a certain model (in this case the excess-phase filter) to a delta variable, which is the original variable minus a zero-order model of that variable. In this particular case the excess-phase model is obtained by forward propagating ECMWF atmospheric field. Totally fine. My issue is that the filter is then applied to the delta profile only, and the filtered "delta" excess-phase is then added back to the excess-phase model. High frequency components inherently contained in the model are not filtered out. Do I mislead something? Could you better clarify, since the reference points to a presentation given to a congress?

Eq A16, #13: the same as before.

Eq A17, #15: The different coefficients identified in the first two and last two lines of AL2D matrix takes care of boundary effects in applying the 5 points derivative scheme. Could you please provide a further clarification on these values or a proper reference? Also, the denominator (12 Delta t) is not so straightforward.

**5: about the "mean" tangent point location. How it is computed? Is it the lat/long of the intersection between the GPS-LEO line with WGS84 when the line is tangent to the ellipsoid (Straight Line Tangent Height = 0). Is there any effect on the uncertainty propagation in the bending angle profile due to an error in locating the center of symmetry? Could you provide a clarification on this?**

**7: what does it mean "is accurately valid geometrically"?**

**9: the impact parameter retrieval is "mildly non-linear". Fine. But what are the effects**

of the non linear part in the uncertainty propagation? (see also point 5) in general comments).

**31 - 3 p.22: This sentence is crucial but not clear. Is 2% the residual error on the bending angle or impact parameter left to the non-linearity (see above comment)? You are saying that the assumption is reasonable given the high quality of the forward modeled profiles? Why the high quality related to this error?**

A29, #4: G(za) also in the first member? I'd say that is the linearization error of 2% applied to all the levels?

**7: in Eq A29, the random uncertainty profile ualphaG is already interpolated to common monotonic impact height grid. Why it is repeated here?**

**18: "zero-order contributions and no terms higher than...". Is this the Taylor expansion used to linearize A22-A24 (zero-order is already used to identify the model applied to the baseband approach for the state vector retrieval)? Referring to point 5) in the general comment, you can estimate the effects of non-linearity on uncertainty propagation evaluating ualphaG also including higher order terms.**

A33, #20: why the derivative of alpha wrt theta is not considered?

Page 24 and Page 25

I'd remove both A2.2 and A2.3. See issue 1) in the general comment section And, by the way, the effect of the apparent systematic uncertainty in the lower troposphere seems "crucial" but not still fully accepted (the paper is under review). It is worth to have the WO/merging part properly described elsewhere in the future, as soon as the results associated to the uncertainty propagation through WO algorithms will be available.

Page 27:

**3: here you are talking about a "more advanced form" of alphaL2 extrapolation which is described in a technical report not publicly available. Provide other references if possible or avoid to cite it.**

**13: see above. Provide other references if possible, or avoid to cite it.**

Page 28:

Eq A48, #10: why the coefficient are squared? Is it because the CP foreseen to multiply the model and its transpose?

**15-17: Clarify this sentence.**

**29: the Healy/Culverwell cited paper refers basically to a new ionospheric correction schema. Are there plans to introduce this in rOPS and to evaluate also how uncertainties are propagated through it?**

Page 29 # 19: see previous comments on this.

Technical comments

**28: typo: validation**

**28: introduce what the subscript "r" stands for, even if there is a table in support of all the definitions.**

**24: make reference to Fig 3b)**

**14: alphaG,k(za) is repeated. Moreover alphaM,k(za) is never shown. Remove it**

**16: add "making negligible the systematic uncertainty integrated from the phase/Doppler"**

**10: is this recombined MC covariance matrix the one defined through Eq. 10? If yes, put a reference to the equation.**

**28-32: here you provided a mention another possible method for estimate uncertainties along the processing chain (variance propagation). Totally fine. But it would be nice that it is properly introduced somehow in Sect 2.1.**

Page 14: "minor channel". Please reword this. In GNSS we do not have minor/major channels. We have channels associated to the lower and higher carrier frequencies or, simply channel associated to the L1 or L2 carrier frequencies. Also Page 26, #4 and #8, Page 27 #8. Page 45, Figure 10 caption (leading ? channel)

**9: typo: derivative scheme**

**7: I'd add: "In the GO approximation, the bending angle values at each grid point only depend on..."**

Figures 3-11: It would be nice to have also the uncertainty (random and systematic) plotted as relative values (relative to the state variable profile)
* * *

---

## Referee Comment (RC2) · Anonymous Referee #2 · 23 Aug 2017

General Comments

As most of remote sensing techniques, uncertainty analysis is essential to quantify the retrieval credibility in a GNSS-RO system. This article describes the uncertainty propagation of GNSS-RO with step by step approach. From excess phase to bending angle, the propagation process of both random and systematic uncertainties at each step are introduced in details. While the description of the uncertainty propagation is nearly complete and the validation results are very impressive, I recommend this article published after minor revision on several issues:

1. The "estimated system uncertainty" is not well-defined and needs more explanation.

What are the sources of the system uncertainty in excess phase? Why can we model it with eq. 8? Are these estimated system uncertainty totally uncorrelated with random uncertainties in each step so that we can treat them separately? What if the bias actually comes from the signal randomness (e.g. the bending angle bias caused by signal noise as depicted by [Sokolovskiy et al., 2010])? Should we count it as system uncertainty or random uncertainty?

2. Although the MC simulations validate the propagation process, whether the propagation results can reflect how real data behave is questionable:

(i) One thing I concern the most is the modeling of random uncertainty as normal distribution. While the "residual phase" of the RO signal suffered by thermal noise could be normally distributed (strictly speaking it is not), the excess phase calculated by unwrapping residual phase can contain cycle slips (even bias if the used model is biased) due to signal noise. Obviously the nonlinear unwrapping process is ignored in this article, and I'm wondering if it could impact the uncertainty propagation results? If this has already been considered in the eq. 6, then author should explain how this model is derived rather than simply providing a technical report reference.

(ii) The random uncertainty is highly related to the signal SNR. However, the linear extension used below 30 km removes all the corresponding SNR information. The reason of using a linear gradient model below 30 km instead of the calculated $ddL_{rm,k}$ should be given.

(iii) A key element this article lack of is the verification of the propagated uncertainty using the actual data. The direct comparison in random uncertainty might be difficult, but the system uncertainty, or bias as you defined in P.4, could be observed statistically through the comparison between RO and ECMWF (or other measurement like RAOB). We may have more confidence on the propagated system uncertainty if it matches the comparison results.

[Sokolovskiy, S., C. Rocken, W. Schreiner, and D. Hunt (2010), On the uncertainty of

radio occultation inversions in the lower troposphere, J. Geophys. Res., 115, D22111, doi:10.1029/2010JD014058.]

Specific comments

*** P7, eq. 6 & eq. 8 ***

These two equations should be better explained: why linear and where are these constants (3e6 and 3e7) come from? Why eq. 6 is better than the original ddL_{rm,k} in modeling the random uncertainty below 30 km?

*** P7, L23 – L26 ***

Why 0.1 mm and 0.2 mm for simMetOp? Why 0.2 and 0.4 mm for the other two? Why are they constants over 8 km to 80 km? What are the causes of the modeled systematic uncertainty?

*** P11, L18 ***

$F_{c2}$ is set noise dependent – How to determine the filter bandwidth? Do you have to check the spectrum first?

*** P14, L8 ***

What is the criteria used for discarding 5% of the processed profiles?

*** P20, L7 ***

Although the conclusion is the same but shouldn't the BWS filter be 41 points as you stated in P. 16?

*** P23, eq. A33 ***

Why the systematic uncertainty of bending angle is not related to the open angle?

*** P41, Fig. 6(b) ***

When comparing Fig.5(b) and Fig.6(b), it surprised me that the Doppler systematic

uncertainty increase at the bottom of the profile vanished in the one of bending angle. In figure 6, it's just a constant all the way down. Is there any specific reason for this? The Doppler uncertainty is too small compared to the orbit uncertainties? So most of the systematic uncertainty of the bending angle comes from the orbit instead of the measured Doppler?

*** P45, Fig 10(g) ***

Can you provide an explanation why several cases in simMetOp have larger L_f uncertainty between the impact altitude of 40 and 60 km?

---

## Author Comment (AC1) · 23 Nov 2017

**Response to Referee #1**

Manuscript "Integrating uncertainty propagation in GNSS radio occultation retrieval: from excess phase to atmospheric bending angle profiles"

by Jakob Schwarz, Gottfried Kirchengast, and Marc Schwaerz,
AMT Discussions paper, doi:10.5194/amt-2017-159

==================================================================

*We thank the reviewer very much for the constructive and detailed feedback to our manuscript. We thoroughly considered all comments and carefully revised the manuscript accounting for most of them. Below are our point-by-point responses.*

*Comments by the reviewer are cited* black upright*, our responses are* blue italic*. (line numbers used in our responses refer to the original AMT Discussions paper and text updates in the revised manuscript are quoted below with* yellow highlighting*)*

**General comments**

The paper details the algorithms, the underlying assumptions and the results of the uncertainties estimation (random and systematic) in the GNSS radio occultation (RO) retrieval, from the excess-phases to bending angle profiles. The paper, together with other similar papers which address the same topic once applied to the other portion of the entire RO processing chain (thus describing error propagation through the POD solution and through the retrieval steps of atmospheric profiles starting from bending angles), provided for the first time a rigorous mathematical overview of how errors are propagated through the entire RO processing chain. From the scientific point of view the paper is almost complete; involved data, algorithms, results, discussion and conclusions are presented in a well structured and clear way, even if the paper is excessively long.

In what follows some general comments are provided.

1. One of the most critical step of the RO processing chain is the retrieval of bending angles in the lower troposphere, which involves more "advanced" methods (the so called **Wave Optic retrieval**). Unfortunately error propagation through this step is not detailed (even if a paper is already available since 2015, it is written a couple of times in the manuscript that the implementation of WO uncertainty propagation in the rOPS is not yet finished... so it is not fully clear whether the results are already available or not). In order to have a complete and more coherent overview, this part is somehow necessary. My suggestion is, if results are not available yet, to remove from this paper all the discussions provided on the WO retrieval and error propagation through it (Sect 3.2.2, Sect 3.2.3, Annex A2.2, Annex A2.3), changing a bit also the title reflecting that only GO retrieval is applied.

*We carefully checked the impact of this alternative of dropping the link to WO in this paper and found it would do no good to the readers if this paper would not clearly show how the WO uncertainty work as introduced by Gorbunov and Kirchengast (2015; 2017) is embedded in the overall bending angle retrieval, including in*

*particular the merging of WO and GO and of their uncertainties. That is, even though the testing and integration-test approval of the WO uncert.prop. in the rOPS—implemented as discussed by Gorbunov and Kirchengast (2017)—is not yet completed so that we could not yet use the WO results in this study, we know with confidence that the subsequent algorithm step in the rOPS uncert,prop., the merging of GO and WO uncertainties, can be mathematically implemented and tested independently. This holds, since only if the WO uncertainty propagation would deliver large outliers with unexpected behavior there might be problems; but we know from WO tests this not to be the case. Also, since we tested a whole day of real and simulated data with a relatively large variability in u^r (and offline with some even larger test magnitudes), the stability of the merger-algorithm itself is evident for any reasonable magnitude of the uncertainty profiles delivered by the WO uncert.prop. In other words, using in this paper the uncertainties from the GO also as WO input for demonstrating the merger, allows us to have this way a sound "proxy" to reasonably cover the possible variability of the input uncertainties coming in future from WO.*

*We realize, though, that we need to update the text at several places (given that the Gorbunov and Kirchengast 2017 AMT paper is meanwhile in press and the rOPS implementation of WO is meanwhile in integration tests), aiming to make the value of linking to WO in this paper and also the referencing more clear. We note that—similar as done in Schwarz et al. Earth Space Sci. (2017) for the refractivity and dry-air retrieval—the focus of this study is on the careful introduction of the (formally correct) algorithms and on the demonstration of the robust performance, and less on the delivery and presentation of real/realistic uncertainty propagation results (see also our response to Comment 3. below). So we think that presenting the successfully implemented merger algorithm makes sense and is useful to the readers of this study. It also will allow for a more concise presentation of the WO uncertainty propagation as part of the complete chain in a future publication that will then show real-data results from the complete rOPS chain.*

*We updated the text that refers to the uncert.prop. through the WO bending angle retrieval and the merging operator particularly in the following parts:*

*On p11,L3, we updated and improved to "Because the rOPS implementation of the WO uncertainty propagation (Gorbunov and Kirchengast, 2017) was still in test phase and not yet available for integration into the simulations here, all examples in this study…"*

*and on p11, L6, to "In order to nevertheless test and validate the uncertainty propagation of the merging algorithm, WO retrieval results were artificially substituted by the GO results for the MC validation (Section 4), i.e., GO was used as proxy for WO since reasonably capturing expected WO variability as indicated by tests of Gorbunov and Kirchengast (2017)."*

*and on p24, L18, to "…prepared the merging with the WO bending angle variables (they will be actually merged in when the WO tests within the rOPS is complete), which is described next."*

2. Another general issue I want to highlight is that the entire discussion and the main results shown from Fig 3 to Fig 8 are based considering an excess-phases "time

series" and associated random/systematic uncertainties observed during **one real occultation event from the COSMIC mission, taken as representative for the entire day**. Not sure that the heuristic associated to the choice of that particular profile is correct, since the ensemble mean profiles of the uncertainties associated to the various variables involved in the retrieval (Fig 10 and Fig 11) show quite different behavior (see for example the increasing of uncertainties with height). It would be nice to have different examples (maybe the best and the wrong together with this most representative one).

*With Figures 3 to 8 we followed the same didactical approach in introducing the algorithmic steps in combination with some illustrative figures for an example event as we did in the Schwarz et al. Earth Space Sci. (2017) work. The key aim of the algorithm description in chapter 3 and the figures 3-8 is to illustrate the effects of the operators in the rOPS uncertainty and not to show the input-to-output profiles uncert.prop. performance through the chain (this is for the later figures with MC validation and full test-day results). It is for this aim that we found it most clear not to overload these illustrative figure with complex information of more than one example event (or have otherwise even more figures in what is already a quite extensive paper).*

*In this study we chose one real profile as example profile, rather than a simulated profile as in the Earth Space Sci. paper, to capture for bending angle retrieval with its higher sampling rate more characteristics of real profiles typically processed. Still the chosen (COSMIC) example is only used to illustrate the effect of the operators, its random and systematic uncertainty is representative in terms of magnitude and correlation lengths for the profiles encountered in a real processing. It is not intended to be "representative" also in the sense that the results match a quantitative "best case", or "average", or similar. This was also found suitable, since the input profile uncertainties at excess phase level are as yet not the "real" input propagated from the rOPS L1a processing, but rather a proxy and "reasonable guess" input.*

*Regarding the test-day ensemble, the mean of the test-day random uncertainty profiles indeed increases between 70 and 75 km, but this effect is stemming mostly from a relatively small share of profiles with large random uncertainty, thus also not being particularly representative for the majority of profiles.*
*Concerning the use of the "time series" principle, we refer to our response to the reviewer's specific comment below related to "Page4, L9-10".*

*So overall we preferred to continue using and illustrating just one example event, based on the reasons above; also avoiding crowed plots or too many figures in total.*

*However, in order to avoid an interpretation of our example profile as „corresponding approximately to the average uncertainty profile of the test-day RO events", we replaced „a representative" by „an example", on p3, L27, and we toned down „a representative" to „a typical" on p6, L9 (and from this point on the event is called „example case" anyway, as defined at the end of this sentence).*

3. Regarding the same input profile I found a bit inconsistent the **choice of extrapolating downward 30 km** the representative excess-phase random uncertainty profile (defined on the above mentioned COSMIC observation) with a

linear behavior (I guess linear in order to provide a "worst" boundary, could you confirm this in the paper?), whose gradient follows some estimate defined for the GRAS instrument on board METOP satellites (and I don't think that in the provided reference [ESA/EUMETSAT] such information can be easily found). The systematic uncertainty on the same input variable is also defined following error estimates characterizing the GRAS receiver. The big question is why you mixed the random uncertainty "rigorously" derived considering a COSMIC observations with some heuristic based on the GRAS instrument? Was not better to derive the random error profile from GRAS real observations? And why for the GRAS case you demonstrated the uncertainty propagation using simulated data only? Please provide a clear motivation on this or repeat the entire analysis using some representative examples taken by GRAS excess-phases.

*The main intention of this study was to develop, implement, validate and demonstrate the mathematically correct propagation algorithms for bending angle retrieval, under as few assumptions as necessary for a successful application of the uncertainty algorithm to RO data in the rOPS; so same intention as Schwarz et al. Earth Space Sci. (2017) for refractivity and dry-air retrieval. The quality and realism of the propagation results in absolute terms will of course strongly depend on the uncertainty profiles used as input at excess phase level; as also the test-day ensemble results indicate. Since for this study realistic full-profile rOPS L1a uncertainty estimates were not yet available at excess phase level, we decided for viable proxy input profiles with realistic order of magnitudes in terms of variance and correlation length.*

*Regarding the altitudes below 30 km, we chose to replace the noise-estimation scheme by a simple linear model below 30 km, because the empirical estimation approach from the noisy time series (as described in section 2.2, esp. Eq. 5 and the related text) becomes increasingly vulnerable to biases and fluctuations below 30 km due to the strong (near-exponential) increase of the excess phase magnitude. The simple linear model chosen was in fact consistent with that ESA/EUMETSAT reference as well as roughly with the overall behavior of COSMIC-profile based estimates below 30 km (which were not stable enough to be used directly).*

*To be more clear with our rationale for the construction of the input profiles used, we updated on p6, L26-28, to:* ==*"While in future the excess phase random and systematic uncertainty profiles will be more rigorously estimated by the rOPS L1a processor (Innerkofler et al. 2016) and provided as input to the L1b processor, they had to be estimated for this study from existing excess phase profiles with realistic noise (we chose UCAR/CDACC ones) and simplified modeling. To this end..."*==
*and on p7, L7, to* ==*"…roughly following estimates of ESA/EUMETSAT (1998) and the overall behavior of estimates from real excess phase profiles (the latter became too vulnerable to biases and fluctuations to continue using them below 30 km)."*==

*On the demonstration of the uncertainty propagation related to GRAS for simMetOp only, we agree that this was not really justified well (we had decided for it to save in particular some space in figures and text since we considered simMetOp is there anyway).*
*We therefore included real MetOp data now (updated Figures 10 and 11, and related text updates), in particular involving the following updates:*

*on p3, L29-31, to* "*…CHAllenging Minisatellite Payload (CHAMP) (Wickert et al., 2001), FORMOSAT-3 Constellation Observing System for Meteorology, Ionosphere, and Climate (COSMIC) (Anthes et al., 2008), and Meteorological Operational Satellite A (MetOp) (Luntama et al., 2008), and with simulated data…*"

*on p6, L4, to* "*For CHAMP, COSMIC and MetOp, orbit…*"

*on p7, L24, to* "*…for MetOp and simMetOp.*"

*on p14, L2, to* "*…complete test-day of real (CHAMP, COSMIC, MetOp) and...*"

*on p14., L29ff, expand to* "*For the real MetOp data (available here as dataset from UCAR/CDACC, as for CHAMP and COSMC), urLF1 appears similar to COSMIC (cf. Figures 10d, g) while for simMetOp (with best possible simulated GRAS-type receiver noise) it is clearly smaller than for COSMIC. From 35 to 80 km the mean random uncertainty profile for simMetOp stays below 1 mm (Figure 10j). Three individual profiles exhibit comparatively high uncertainties of larger than 2 mm within about 40 to 55 km, however, reflecting that the simMetOp error simulations are capable to partly generate higher-noise profiles of the type more frequently seen in the real MetOp data (Figure 10g).*

*On the other hand, the average correlation length/resolution profile of the ~500 real MetOp and ~700 simMetOp ensemble members is very similar, driven by the orbit being essentially the same for the real data and the simulations (Figures 10h, i, k, l). Compared to COSMIC (Figures 10e, f), the correlation length and resolution are again somewhat larger/coarser, due to an even somewhat higher scan velocity of the MetOp satellite (~820 km orbit altitude).*

*The systematic uncertainty usLF1, just co-illustrated for completeness in Figures 10a, d, g, j, is almost…*"

*on p15, L24, to* "*…for three different missions (CHAMP, COSMIC, MetOp) show…*"

*on p45, caption of Figure 10, update to* "*…for CHAMP (a); COSMIC (d), MetOp (g), and simMetOp (j).*" "*…for CHAMP (b), COSMIC (e), MetOp (h), and simMetOp (k).*" "*…for CHAMP (c), COSMIC (f), MetOp (i), and simMetOp (l).*"

*on p46, caption of Figure 11, update to* "*…shown in colors (CHAMP (yellow), COSMIC (orange), MetOp (red), simMetOp (violet)). Left column:…*"

4. And, always on the definition of the input random uncertainty example profile, does it takes into account the **merging of open loop and closed loop data** somewhere in the lower part of the profile? I guess that the uncertainties related to the open loop and closed loop tracking should be different (because of different noise levels added by the different tracking behavior, because of different sampling rates between OL and CL, which is different from COSMIC and GRAS for example). Maybe the "boundary" set with the linear extrapolation downward the 30 km includes already these different uncertainty levels. But I expect that a discussion on these aspect will be added in the paper.

*As detailed in the response to comment 3 above, the current input uncertainty model does not include sophisticated estimations of the diverse real error sources in the POD and excess phase profiles, being part of L1a processing, But as noted, this is*

*not needed for the algorithm introduction and demonstration context of this study; and these sources will be accounted for in future when the rOPS L1a processor provides the profiles and uncertainties as input to the L1b processor.*

5. All the analysis is based on the covariance matrix propagation through linear (or linearized) operators. This is fine. But what are **the residuals effects due to the linearization**? I guess that the most "critical" linearization is the one applied to impact parameter retrieval (in GO). In that case I guess you can quantify the second/higher order effects by considering more terms in the Taylor's expansion of the operator.

*Relative to the state profile magnitude, the magnitude of the uncertainties is generally very small, and the uncertainty of the uncertainties is even smaller. The relative accuracy of the uncertainty estimates we aim to achieve is within +/-3% when quantitatively determined from Type A evaluation (already comparable to a relative accuracy of the state profile of about 10^-4). The main source of uncertainty in the uncertainty estimates stems from Type B evaluation (particularly for the estimated systematic uncertainty), where it is almost impossible to push the error of the uncertainty estimates to below +/-10%. Given this context we thus consider the ~2% error due to the linearization in the propagation of the bending angle random uncertainty (we agree, the relatively largest error source) as acceptable (p21, L31).*

6. Finally, the most important references related the main topic of the paper (the description of rOPS and of the algorithms involved) are presentations to conferences or not accessible technical reports. This is a pity. **But at least, for the presentations it would be nice to have the full web-link specified**. The paper is absolutely worth to be published, but some review/clarifications should be provided. That's why I suggest for a major review. Specific and technical comments are provided in the next two sections.

*Yes, the general manuscript introducing the rOPS is still in preparation and so cannot be cited yet. We agree that providing ready access to the presentations is thus useful. We therefore added the web-links for the following presentations:*

*p32, L28: Kirchengast et al. 2015: http://irowg.org/wpcms/wp-content/uploads/2014/05/Kirchengast-IROWG-4.pdf*

*p32, L31: Kirchengast et al. 2016a: http://meetingorganizer.copernicus.org/EGU2016/EGU2016-12035-1.pdf*

*p32, L35: Kirchengast et al. 2016b: http://wegcwww.uni-graz.at/opacirowg2016/data/public/files/opacirowg2016_Gottfried_Kirchengast_presentation_261.pdf*

*p33, L1: Kirchengast et al. 2017a: http://meetingorganizer.copernicus.org/EGU2017/EGU2017-16328-2.pdf*

*p34, L27: Syndergaard 1999: http://www.cosmic.ucar.edu/groupAct/references/Sr99-6.pdf*

*And on the detailed algorithmic documentation "behind" the papers (also addressed by the reviewer in a further comment below): The main cited source for the rOPS algorithm documentation, the DAD (Kirchengast et al., 2017b) is indeed not yet publicly available in its current 1.7 version, because it needs to be completed and polished in some (other) chapters and in its introduction and overview. However, we can assure that the DAD is firmly scheduled as an open document, to be published as soon as the journal publications of the introduction and main components of the rOPS and the first reprocessing are complete (expected in 2018).*

*We therefore strongly think it is valuable to quote the DAD also in this publication, so that future readers of any rOPS article (including this one) can readily get to it as soon as it is public, since also many years from now they will find it clearly referenced in all relevant journal publications collectively introducing the rOPS. This "foresight" was also the justification for citing it as reference in the Schwarz et al. Earth Space Sci. (2017) publication, the Gorbunov and Kirchengast Atmos. Meas. Tech. (2017) in press publication, etc.*

———

Specific comments

**6: what does it mean that "the accuracy is also ensured on-orbit"? please specify**

*The way we read the decadal survey (NRC 2007, p 64) is that for climate benchmark data, the accuracy needs to be ensured by the remote sensing technique itself (as with the self-calibration property of RO), rather than through e.g. bias-correction on the ground, to ensure reproducibility in the future with independent means.*

*We improved it to "…the accuracy is also ensured on orbit, i.e., there is no need for calibration or bias correction in post-processing on ground (Leroy et al., 2006)."*

**21-22: remove references to unpublished papers or to papers in preparation. It is already mentioned that work is on-going.**

*Ok, we dropped these references to "papers in preparation" and hence simplified the text at this place to "…are part of on-going work (Innerkofler et al., 2016)." and, two lines below in L24, to "…and corresponding L2b uncertainty propagation is on-going (Li et al., 2017; Kirchengast et al., 2017a)."*

**31: first mention of the use of simulated data for GRAS.**

*See response to major comment 3 above: a test-day with real MetOp data was included now and hence the real MetOp(-A) is also mentioned at this place now.*

2.1 Methods: all the analysis is based on evaluating random/systematic uncertainties as complete independent processes. Could you please explain why the two are completely decoupled? This is probably trivial under the assumption that noise is normally distributed. But is this really the case starting from excess-phases?

*The criteria for separating the evaluation and propagation of estimated random and systematic uncertainties is that the processes giving effect to these uncertainties need to be uncorrelated (if the correlation term is zero, the other terms can be pulled out of the sum). This is independent of the shape of the probability distribution (if a process causes a biased mean of the pdf, it is counted to the (basic) systematic uncertainties, such that the random uncertainty can be assumed unbiased). The basic systematic uncertainty is caused by processes that repeated measurements under (almost) the same conditions could not detect, e.g. insufficiently defined physical constants, most model parameter errors, etc. It can thus not be correlated with effects of processes that give different results under repeated measurements.*

*The random und apparent systematic uncertainties on the other hand can be seen as uncorrelated for an analogous reasoning: Since we separate them based on correlation length, the apparent systematic uncertainty appears as a bias within the, e.g., 0.8 seconds of the Blackman window (correlation coefficient of unity over the window range), while the random uncertainties' correlation function typically decays down to zero within this range. So even if random and (apparent) systematic uncertainty were correlated at one side of the Blackman window, they will be clearly uncorrelated at the other. It is crucial, however, that the correlation lengths of the random and the systematic uncertainties remain clearly separable; would the random uncertainty be strongly correlated also over longer time windows (appearing as a bias), it could be also correlated to the systematic uncertainty (a situation that, favorably, does essentially not occur in practice for RO data processing).*

*Thus, in practice, while there might be (very) slight correlations between random and systematic uncertainties at times, we consider the additional insight from having random and systematic effects separately available more important than the problem of potentially (very) slightly overestimating the uncertainties of our RO profiles.*

*In order to further help understand this, we now added on p4, after L20, another sentence/small paragraph as follows: "Since the noise-type effects giving rise to short range-correlated random uncertainties can be considered uncorrelated to the bias-type effects inducing long range-correlated apparent systematic uncertainties, and since both are uncorrelated to basic systematic uncertainties, it is insightful and possible with due care to estimate and propagate each of these uncertainties independently."*

**9-10: not clear why the effects of temporal/spatial variations (which kind of variations have you in mind?) in repeated observations can be estimated based on an individual RO profile. This is one of the main issue. Describe/motivate this better. And, by the way, why this should be true based to oversampling in the RO raw data profile?**

*This relates to GUM concept. The idea behind this evaluation method is an assumed ergodicity of the essentially stationary observation system (in our case the gaseous atmosphere viewed as a random field), and thus equivalence between the repeated measurements within one RO event (e.g., excess phase sampled at many different atmospheric levels), and measurements from an ensemble of multiple RO events (e.g., excess phase from many events at a certain atmospheric level). Effects causing such ergodic random variations could be the thermal noise of the receiver system, to some extend random uncertainties in the positions/velocities of the receiver and transmitter satellite, the clocks, atmospheric small-scale fluctuations, residual ionospheric fluctuations.*

*We tried to improve the paragraph on p4, L8ff, to: "… Effects of unpredictable or stochastic temporal and spatial variations in repeated observations—like effects from fluctuations in the atmosphere or the thermal noise of the receiver system—could in principle be estimated by ensemble statistics from multiple RO events. However, since such effects are essentially stationary in a statistical sense, we can estimate their statistics also from individual RO event data, given their high noise-resolving sampling rate. These effects are included in the estimated random uncertainties."*

**29: what do you mean with ionospheric noise? Here you are describing the noise in excess-phase measurements, so on both L1 and L2. Are you referring to scintillations? I don't think these can be treated as "normally distributed noise". Define this better.**

*Ok, we improved this and wrote now "…the receiving system noise (i.e., thermal noise and residual clock estimation noise) and the ionospheric noise (from scintillations induced by ionospheric irregularities) are essentially normally distributed overall (Kursinski…). These noise sources are the main contribution to…"*

**29: Why random errors on positions/velocity are not considered here (later on, page 5 #30 you introduce such effects within the systematic uncertainties)?**

*Our rOPS L1a-related studies concerning uncertainties in transmitter- and receiver orbit positions and velocities show that the magnitude of temporal variations within the typical RO event duration of just 1 minute or less for scanning over the 80 km to middle troposphere altitude range are very small, i.e., orbit arcs are in practice very smooth at this RO-event timescale over the altitude range where excess phase magnitudes are less than 100 m (i.e., where the typical mm-level to cm-level noise sources will possibly be non-negligible). POD errors may thus be considered (quasi-)random on full-day timescales or so, but well classify as apparent systematic uncertainties within the 1-min time intervals of individual RO events.*

**24-25: not clear what "is aligned to a joint resolution..." mean. Please rephrase/clarify.**

*We rephrased p5, L24, to: "...the resolution of all profiles is brought to a common altitude-dependent resolution, which reflects..."*

**7: It would be nice to motivate clearly here why you are using simulated data for propagating the uncertainties for the GRAS case.**

*We used the simulated data as a lower bound 'best-possible' case, to see how low-noise data would be propagating through the algorithm. We now also processed real GRAS data additionally to the simMetOp data – see under major comment 3 above for all the related updates.*

**13-14: clarify why the "baseband" approach allows to avoid biases from numerical operations on near exponentially varying profiles (also repeated at page 17, #11-12).**

*Ok, improved this sentence to "…varying RO profiles, since the model profiles that we derive from short-range (24 h) forecasts of the European Centre for Medium-Range Weather Forecasts (ECMWF) skilfully subtract the (near-)exponential variation. The remaining increment profiles that we need to treat numerically then appear to be very linear and with low dynamical range, which leads to very low residual numerical errors of operators such as filters and derivatives."*

**4-7: here is where the downward extrapolation based on a GRAS "model" is described. See point 3 of General Comments.**

*To clarify our view of the nature of the input profiles used, we changed on p6, L26-28, to: "While in future the excess phase random and systematic uncertainty profiles will be more rigorously estimated by the rOPS L1a processor (Innerkofler et al. 2016) and provided as input to the L1b processor, they had to be estimated for this study from existing excess phase profiles with realistic noise (we chose UCAR/CDACC ones) and simplified modeling. To this end..."*

**4-7: why linearly extrapolating downward and constantly extending upward a "random" uncertainty profile does not introduce a systematic effect?**

*We do not manipulate the state profile, only the magnitude of the random uncertainty profile. Technically, of course, the random uncertainty is not completely correct at the extrapolated sections, but this is equally true for the section with the noise-based estimation. As described in the response to comment 3, we only create a proxy-input to work with. Also, we only manipulate the magnitude (variance) of the uncertainties, not the correlation structure.*

**8: Please clarify further why all the noise components responsible for random uncertainty at excess-phase level are uncorrelated (also referring to the ionospheric noise). I believe that at carrier phase level noise is uncorrelated (we have basically phase noise). But when geometry is removed as well as clock biases, and when the result is interpolated to match 50 Hz sampling rate (maybe between open and closed loop observations), I'm not sure that the various components are still uncorrelated.**

*Our assumption is that these random effects are only correlated within the range of the BWS-filter width and we can thus start the L1b processing with the assumption of vertically uncorrelated errors. The analysis of the uncertainties in the L1a processing will show whether we will be able to uphold this clear distinction (i.e., in practice, the rOPS system checks this expected very-narrow-correlation between the L1a and L2b processing, by empirically estimating the excess phase correlation matrix from a large ensemble of L1a output profiles around the "currently processed day"; in case the random error correlations would appear too strong, we would feed the L1b processor with a correlation matrix that is non-diagonal accordingly). On the other hand, if a bias is found introduced and determinable, it will be corrected for (according to the GUM); if we can't determine it, an increase of the residual systematic uncertainty bound estimated would be the way to account for these components.*

*We have added on p7, in L10, the sentence: "…for both channels. In case the future excess phase data from the rOPS L1a processor exhibit non-negligible correlations for some data from some of the RO missions, we will account for these correlations in R_Lr, since our L1b algorithm (Section 3) is prepared for full covariance propagation."*

**14: Being he entire analysis based on a COSMIC profile, why for the MC validation you started from a simulated CHAMP excess-phase profile and not from a realistic COSMIC profile? Motivate it.**

*Ok, we added on p7, L15: "... receiver system errors superimposed). Using an 'error-free' profile as basis, the particular simulated profile just serving as a representative RO profile to illustrate the MC validation, allows us to strictly ensure the consistency of the random uncertainty of the input profile with the ensemble of superimposed error profile realizations."*

**24: in defining the input random uncertainty profile, you are computing the excess phase mean value between 60-70 km. Is the constant part of the systematic uncertainty consistent with such a mean?**

*No, in the current simplified input uncertainties modeling that we use in this study for introducing the algorithm (not focusing on realism of input and output, as explained in our response to comment 3 now), we don't consider this "empirical value" as viable basis for quantifying systematic uncertainty.*

**Eq 6 and 8: Explain why the "gradient" (1/3exp(xx)) for the random and the systematic uncertainties below zGrad is different.**

*See our response to comment 3, where we explain our simplified modeling below 30 km, and note the improvements we did to the text in the revised manuscript. Specfically to the systematic uncertainty, we in addition extended on p7, L24, to:* *"…linear uncertainty gradient in the troposphere; as noted above this simplified modeling will be replaced in future by realistic uncertainty estimates received as L1b retrieval input from the L1a processor (Innerkofler et al., 2016)."*

**20-21: not clear the sentence. By the way, it would be nice to show the relative error (to the state profile) instead the absolute one.**

*We reconsidered this but preferred to stick to the rationale of our original choice here – i.e., the main two reasons why we chose to plot the absolute uncertainties, are 1. that the range of the uncertainties is relatively constant in absolute terms but strongly decreasing in relative terms, particularly for the excess phase and Doppler shift uncertainties, and 2. that when plotted in relative terms, it adds difficulty and ambiguity to the interpretation of the results, due to the influence of the denominator. Furthermore, to keep the plots consistent throughout all figures, we preferred to keep ALL subplots of all these figures in absolute terms, optimizing comparability (we also tried double-axes but it overcrowds the figs, or considered to show both separately but that leads to too many figs in what is a long paper already).*

**25: What is written there is true, but this is not clear whether these results (and the merging with bending angles uncertainties derived by GO processing) are available or not (#3-5 p 11 it is clearly stated that their integration in rOPS is not yet finished). I suggest to remove any reference to uncertainty propagation through WO/merging unless the results can be shown. See general comment 1).**

*Please refer to our response to general comment 1, we hope with that extensive answer (and the related improvements) our solution to this issue in this paper became clear.*

**6-8: this is definitely not clear. How you can "artificially" substitute WO with GO uncertainty propagation, being the vertical domain of their applicability different?**

*Please again refer to our response to general comment 1, also this should be clear now.*

**17 and 20: Not clear why the low-pass filtering of alphaM1 provides so different random uncertainty levels and broaden the correlation functions if the cut-off**

frequency is the same of the one already used to filter the corresponding excess-phases. I expected negligible or a smaller effect.

*The operations on the BWS-filtered profile, particularly the Doppler differentiation, create short-range correlation (anti-correlation) in the retrieved profiles and thus high(er) frequency variations, which are affected by the second filtering.*

*To make this more clear on p11, L14, we improved the sentence(s) to:* ==*"The chosen filter cutoff-frequency for k = 1 is fc1 = 2.5 Hz, same as the basic filtering (Section 3.1.1), just to ensure clearness of any higher-frequency effects from operators after the initial excess phase filtering (e.g., from Doppler shift derivation that induces short-range anti-correlation effects). For k = 2, the cutoff-frequency fc2 is set noise-dependent, between 2.5 and 0.5Hz (boxcar equivalent width of 0.2 to 1.0 s)."*==

**23: I found inconsistent defining the input data for the MC simulations by using a "background" error-free excess-phase profile taken by a simulated CHAMP event (mentioned not here but at page 7, #14) plus error realizations taken by a COSMIC dataset. The description of this input dataset should be better motivated and clarified. Moreover the input for the covariance matrix (for the MC simulation) seems to be some standard profile uSTD,Lr not well identified.**

*Please see our response to the comment concerning p7, L14, above, and in particular the response to the major comment 3, including the related revisions; this should clarify our approach.*
*To put it in different words again here, for the MC validation – in our eyes and experience – the most relevant aspect of the input profiles & uncertainties is that they are exactly the same as the ones used for the CP (covariance propagation). If this criterion is fulfilled, technically ANY reasonable 1-D profile could serve as basis profile, for the purpose of validating the CP through the MC ensemble (with the latter ensemble constructed by superimposing random error realization profiles on the 1-D profile).*

**28: Here you are saying that the comparisons between MC simulations and covariance matrix propagation (CP) are carried out in terms of their decomposition (error profiles and correlation functions/lengths). This is fine but it is not clear where the CPpropagated covariance matrix comes from. What is the input? The same COSMIC profile taken as example in the previous discussion? Another excess-phase profile (more likely yes, since the results shown in Fig 9 [red, blue plots] are bit different from the same results shown in the previous figures. In #19, p7, you mentioned that "the same standard profile was used as input for the CP"). Rephrase and clarify the entire introduction to Set. 4.**

*Thank you for pointing this out, we seem to have created a misunderstanding from some lack of a bit of info how the CP matrices come into play here.*
*We now updated the text on p12, L28, to:* ==*"...are calculated (Items b-g in Figure 2). Using the same input profile and uncertainty information as used to specify the MC runs (described in Section 2.2), the retrieval is then also run with covariance-based*==

*uncertainty propagation and the resulting CP-propagated covariance matrices $C_X{}^{CP}$ are compared to the MC-derived matrices $C_X{}^{MC}$. In order to be able to attribute...”*

**7: the information provided in Fig 9 are not so clear. First of all are the random error profiles taken to lye between the boundaries provided by blue/orange lines used to plot Lr (thus varying up to 20 mm)? Could you use other styles to plot such information (dotted lines or different colors). Why the area within the LF random uncertainty is colored?**

*We tried a lot to make this fig informative, and in terms of how to show we did not find a really better solution. However, we clarified its explanation in the caption in that we updated the caption of Fig 9 in several places, as follows:*
*in first sentence to "…consecutive retrieval steps are shown for LF (a-c; in a-b also for Lr) and Dr (d-f) relative to…”;*
*in second sentence to "…shows estimated random uncertainties for $f_{T1}$ (CP in blue, MC in black, in (a) $u^r{}_{Lr1}$ in light blue), the middle column for $f_{T2}$ (CP in red, MC in black, in (b) $u^r{}_{Lr2}$ in orange), the right column…”;*
*in third sentence to "…variance propagation ('VP') results (light blue) for $\alpha_r$ in (n).”*

**13-14: why it is "obvious" that the CP delivers the correct off-peak results? Provide a clarification.**

*Since we start with a diagonal covariance matrix with no correlation between distant elements and only apply a filter with a limited filter-width (corresponding to a band-matrix), the correlation of the current element with any element outside the filter window must be zero.*
*We updated on p13, L14, to: "…that the CP delivers the correct off-peak results (i.e., zero; the off-peak elements outside the BWS filter window must nominally be zero).”*

**17-21: rephrase a bit the sentence, since it is not clear.**

*Ok, we changed the text part on p13, L17ff, to: "For comparison, in Figures 9a to 9f, all quantities have been computed on the common time grid ('setting time' relative to time zero at 80 km altitude) with 50Hz sampling rate; and the corresponding impact altitude of the 'true' profile LTr is shown for additional convenience on the RHS axis. In Figures 9g to 9o, these bending angle quantities have been computed on the impact altitude grid; in these cases therefore the corresponding setting time of the 'true' profile is shown for additional convenience on the RHS axis.”*

**9: Are outliers defined in term of individual random uncertainty profiles? Or input Excess-phases profiles?**

*Ok, to better clarify this we updated the text on p14, L8-9, to:* "…because they were detected as outliers based on the magnitude of their random uncertainty profiles (these outliers are not included in the number of profiles shown)."
*We note that this was just a temporary solution for this L1b algorithms intro study; a more advanced QC is integrated as part of the rOPS L1a/L1b interfacing.*

**7: this reference is a technical report, not accessible. Since it is quite important for the main topic of the paper, please provide an open reference if available (or avoid to cite e technical report which is not public).**

*Please refer to our response to general comment no. 6; we hope our preferred approach expressed there is ok and we can implement it also in this study.*

Eq. A2, #14: The baseband approach allows the application of a certain model (in this case the excess-phase filter) to a delta variable, which is the original variable minus a zero-order model of that variable. In this particular case the excess-phase model is obtained by forward propagating ECMWF atmospheric field. Totally fine. My issue is that the filter is then applied to the delta profile only, and the filtered "delta" excessphase is then added back to the excess-phase model. High frequency components inherently contained in the model are not filtered out. Do I mislead something? Could you better clarify, since the reference points to a presentation given to a congress?

*The understanding of the reviewer is correct; and the answer is that the forward-modeled profile is smooth over the scales of the filtering window, i.e., with negligible high-frequency components (as a side note, we experimented with making this pro-actively sure with some slight filtering of the refractivity model profile already before we forward-project it to bending angle, Doppler shift, excess phase; we found though, it was not needed so far; in the full chain processing we will cross-check this again whether there's any appreciable difference).*
*We adjusted the sentence on p17, L12, to:* "…applying the filter only to the delta-profile dLrm = Lr – Lm (with the model profile being adequately smooth over the scale of the filter window width). This approach…"

Eq A16, #13: the same as before.

*As used in this context, we note that the (adequately smooth) Lm and Dm model profiles are of course strictly consistent, i.e., Dm is an accurate derivative of Lm, and, vice versa, Lm is an accurate integral over Dm.*
*Here we now updated the sentence on p19, L7-8, to:* "…the zero-order Doppler shift model profile Dm is added (the latter also available from the forward modeling, in a form strictly consistent with the excess phase model profile Lm)."

Eq A17, #15: The different coefficients identified in the first two and last two lines of $A^{L2D}$ matrix takes care of boundary effects in applying the 5 points derivative scheme. Could you please provide a further clarification on these values or a proper reference? Also, the denominator (12 Delta t) is not so straightforward.

*Ok, the reference to Syndergaard (1999) serves this.*

**5: about the "mean" tangent point location. How it is computed? Is it the lat/long of the intersection between the GPS-LEO line with WGS84 when the line is tangent to the ellipsoid (Straight Line Tangent Height = 0). Is there any effect on the uncertainty propagation in the bending angle profile due to an error in locating the center of symmetry? Could you provide a clarification on this?**

*Ok, we added in a text part on p21, L5ff, as follows: "…at the mean tangent point (MTP) location of the RO event. The MTP location is defined as the geodetic (geographic) location on the WGS84 ellipsoid, where the straight-line path between transmitter and receiver touches this ellipsoid, i.e., where the straight-line tangent height is zero. This can be computed with very high accuracy at the sub-meter level (see Scherllin-Pirscher et al. (2017) for more details on the geolocation accuracy of RO). Using the MTP location's center of local curvature…"*

**9: the impact parameter retrieval is "mildly non-linear". Fine. But what are the effects of the non linear part in the uncertainty propagation? (see also point 5) in general comments).**

*Please refer to our response to general comment 5, where we have answered to this.*

**31 - 3 p.22: This sentence is crucial but not clear. Is 2% the residual error on the bending angle or impact parameter left to the non-linearity (see above comment)? You are saying that the assumption is reasonable given the high quality of the forward modeled profiles? Why the high quality related to this error?**

*As we also allude to in our response to comment 5, we use the 2 % error due to the linearization (as estimated by Melbourne 1994) as reasonable bound for another error, namely the error we make by replacing the "true" impact parameter by the "forecast" impact parameter. We then argue that, since the "replacement error" due to this is smaller than the "linearization error" of up to 2 %, we find it acceptable. Both contribute to the inaccuracy of the random uncertainty estimate ur_alphaG,i and therefore we increase this uncertainty estimate by 2 % (Eq.A29). (We admit, we also in some view do not fully play the GUM recommendations at this spot as we increase the random uncertainty estimate to be conservative instead of explicitly co-estimating also these small further contributors which we theoretically could with effort). However, and important as a cross-validation, the MC validation shows that the magnitude of the combined effect of these two errors appears to be small, since the*

*MC covariance matrix is calculated without the linearization (but using A22-A25) and the results are very close to the results of the CP (which does use the linearization).*

*To better clarify this now we updated the corresponding text on p21, L31 - p22, L2, to: "We use the forward-modeled impact parameter $a_{tm}$ instead (i.e., adopt $a_{tm} = a_{Tt}$ ) and accept the additional error thus incurred, assuming it is smaller than the 2 % relative error due to the linearization estimated by Melbourne et al. (1994). This is a reasonable assumption given the high quality of our forward-modeled profiles derived from ECMWF short-range forecast refractivity fields.*
  *As a consequence we have to accept that the overall inaccuracy of our random uncertainty estimate cannot be brought below 2 %. Therefore, to ensure that our simplified estimate does not underestimate the real uncertainty, we account for the linearization error by multiplying a factor f_uolin = 1.02 to the uncertainty...";*
*and in Eq.A29 we made the RHS term more clear, changing it to*
*"= f_uolin · ur_alphaG(za),i .";*
*and in the line below (p22, L5) we updated to: "In this way we acknowledge that although the calculation..."*

A29, #4: G(za) also in the first member? I'd say that is the linearization error of 2% applied to all the levels?

*Yes, it is a conservative correction of our random uncertainty estimate over all grid-points, by a factor of 1.02 (see response to previous comment).*

**7: in Eq A29, the random uncertainty profile ualphaG is already interpolated to common monotonic impact height grid. Why it is repeated here?**

*For e.g. two transmitter frequencies: The two random uncertainty profiles calculated in A28 are the uncertainties of the two bending angles relative to coordinate variable z_a, but the two 1D vectors (of the bending angle profiles) still lie on the common time grid points (and these correspond to different z_a values for each individual profile). So in order to now state each bending angle vector on the same common impact altitude grid, we would need to interpolate bending angle 2 to the grid of bending angle 1 (in practice, because we want a monotonic grid, we modify it first).*

**18: "zero-order contributions and no terms higher than...". Is this the Taylor expansion used to linearize A22-A24 (zero-order is already used to identify the model applied to the baseband approach for the state vector retrieval)? Referring to point 5) in the general comment, you can estimate the effects of non-linearity on uncertainty propagation evaluating ualphaG also including higher order terms.**

*Ok, to avoid potential "terminology confusion" we improved on p22, in L18 to "…state quantities (serving as zero-order state) and no terms higher than…"*

*And thanks for the thought (again) on the potential utility of explicitly estimating also higher-order terms to gain some estimates also on the uncertainty of the uncertainty estimates. We understand this and have loosely some (cluttered) notes in this direction but see it clearly beyond the scope of this introductory (version 1) algorithms introduced in this study (no surprise it implies a significant share of extra work to get this done in a neat form).*

A33, #20: why the derivative of alpha wrt theta is not considered?

*Thank you for spotting this, this was left as a bug in latex-setting here; of course this added term is needed under the square root (p23, L20); it is now added in there.*

Page 24 and Page 25

I'd remove both A2.2 and A2.3. See issue 1) in the general comment section And, by the way, the effect of the apparent systematic uncertainty in the lower troposphere seems "crucial" but not still fully accepted (the paper is under review). It is worth to have the WO/merging part properly described elsewhere in the future, as soon as the results associated to the uncertainty propagation through WO algorithms will be available.

*I hope we could address this concern sufficiently with our detailed response to general comment 1, and our related updates quoted there. As expressed there, we strongly consider our approach important for using this paper as properly in future together with the other rOPS introduction papers.*

Page 27:

**3: here you are talking about a "more advanced form" of alphaL2 extrapolation which is described in a technical report not publicly available. Provide other references if possible or avoid to cite it.**

*We understand this concern, and in other contexts than the rOPS-related set of papers and underlying documentation would simply agree; but given it is this special context (see again our response to general comment 6) we strongly prefer to keep this way back-referencing to an even more detailed source that (we can guarantee) will become open "background publication" (and should be linked to all "foreground papers").*

**13: see above. Provide other references if possible, or avoid to cite it.**

*Please see the previous comment.*

Page 28:

Eq A48, #10: why the coefficient are squared? Is it because the CP foreseen to multiply the model and its transpose?

*It is due to the application of the general uncertainty propagation rule (e.g., GUM part for treatment of uncertainties for multivariate data) – can be viewed as a generalization to covariance matrices of the well-known uncertainty propagation rule of the form $u^2_z = (dz/dx)^2 u^2_x + (dz/dy)^2 u^2_y$ that applies to variances.*

**15-17: Clarify this sentence.**

*Ok, we updated on p28, L15-17, to: "where it is assumed that the systematic errors in $\alpha_{F1}$ and $\alpha_{F2}$ are positively correlated, i.e., have the same sign, and the associated uncertainty estimates are hence subtracted from one another (as the bending angles are in Equation A46). This assumption is reasonable, since the same sources of non-ionospheric systematic effects apply to both frequency channels (Doppler shift, orbit velocity, and orbit position uncertainties)."*

**29: the Healy/Culverwell cited paper refers basically to a new ionospheric correction schema. Are there plans to introduce this in rOPS and to evaluate also how uncertainties are propagated through it?**

*Yes, it is planned to implement the higher-order correction of Healy-Culverwell in the rOPS, and to evaluate how it possibly helps to reduce the us_RIB due to actively modeling and subtracting this "higher-order ionospheric bias".*

**19: see previous comments on this.**

*Please see the previous answers to this.*

**Technical comments**

**28: typo: validation**

*Thanks, corrected to "validation" (p3, L28).*

**28: introduce what the subscript "r" stands for, even if there is a table in support of all the definitions.**

*We added "retrieved" before "excess phase profile" on p5, in L27.*

**24: make reference to Fig 3b)**

*We inserted this on p7 in L29 in the form "…smooth these simple uncertainty models around these transition altitudes (for the example profile, $u^s_{Lr}$ is visible in Figure 3b)."*

**14: alphaG,k(za) is repeated. Moreover alphaM,k(za) is never shown. Remove it**

*These are two different variables, since the coordinate variable (the argument) changes. alphaM and the two different alphaG's are shown in figure 2, which we refer to in this paragraph. Also the algorithm for calculating alphaM and its uncertainties is presented (please see also our arguments concerning general comment 1).*

**16: add "making negligible the systematic uncertainty integrated from the phase/Doppler"**

*Please see in the previous relevant comment how we have changed this; it is accounted for now.*

**10: is this recombined MC covariance matrix the one defined through Eq. 10? If yes, put a reference to the equation.**

*Ok, added reference to Eq. 10 in L10, p13: "…how far down the MC covariance matrix (Equation 10) reaches."*

**28-32: here you provided a mention another possible method for estimate uncertainties along the processing chain (variance propagation). Totally fine. But it would be nice that it is properly introduced somehow in Sect 2.1.**

*We inserted a paragraph on p5, L11ff, as follows: "Since the covariance propagation of random uncertainties requires extensive matrix multiplications for each measurement model along the entire retrieval chain, we also tested simpler variance propagation, for which correlations are ignored; Appendix B summarizes the relevant algorithms. However, as shown in Section 4, variance propagation unduly overestimates random uncertainties so that covariance propagation is required. …When the operator is linear…"*

Page 14: "minor channel". Please reword this. In GNSS we do not have minor/major channels. We have channels associated to the lower and higher carrier frequencies or, simply channel associated to the L1 or L2 carrier frequencies. Also Page 26, #4 and #8, Page 27 #8. Page 45, Figure 10 caption (leading ? channel)

*We did not immediately find a better term expressing what we want to say with „leading channel" and „minor channel" from the point of view oft he algorithmic approach (not the point of view of GNSS, or RO as a method). We would, however, if we have a good idea also correct this in the editing of a final production manuscript.*

**9: typo: derivative scheme**

*Thanks, we corrected to "derivative" (L9, p19).*

**7: I'd add: "In the GO approximation, the bending angle values at each grid point only depend on..."**

*Thanks, extended the beginning of the sentence on p22, L8, to: "In the GO approximation, the bending angle values…"*

Figures 3-11: It would be nice to have also the uncertainty (random and systematic) plotted as relative values (relative to the state variable profile)

*Please refer to our response to comment p9, L20 (and one other related comment).*

---

## Author Comment (AC2) · 23 Nov 2017

**Response to Referee #2**

Manuscript "Integrating uncertainty propagation in GNSS radio occultation retrieval: from excess phase to atmospheric bending angle profiles"

by Jakob Schwarz, Gottfried Kirchengast, and Marc Schwaerz,
AMT Discussions paper, doi:10.5194/amt-2017-159

================================================================

*We thank the reviewer very much for the constructive feedback to our manuscript. We carefully considered all comments and revised the manuscript accounting for most of them. Our point-by-point responses to the comments are given below.*

*Comments by the reviewer are cited* black upright*, our responses are* blue italic*. (line numbers used in our responses refer to the original AMT Discussions paper and text updates in the revised manuscript are quoted below with* yellow highlighting*)*

**General Comments**

As most of remote sensing techniques, uncertainty analysis is essential to quantify the retrieval credibility in a GNSS-RO system. This article describes the uncertainty propagation of GNSS-RO with step by step approach. From excess phase to bending angle, the propagation process of both random and systematic uncertainties at each step are introduced in details. While the description of the uncertainty propagation is nearly complete and the validation results are very impressive, I recommend this article published after minor revision on several issues:

1. The "**estimated system uncertainty" is not well-defined** and needs more explanation.

What are the sources of the system uncertainty in excess phase? Why can we model it with eq. 8? Are these estimated system uncertainty totally uncorrelated with random uncertainties in each step so that we **can treat them separately**? What if the bias actually comes from the signal randomness (e.g. the bending angle bias caused by signal noise as depicted by [Sokolovskiy et al., 2010])? Should we count it as system uncertainty or random uncertainty?

*We see two aspects in this comment, which we would like to respond to separately.*

*First, on the sources of the systematic uncertainty and our (simplified) modeling of it, i.e., on the rationale why we modeled the (systematic and random) uncertainties in the excess phase the way we did it in the particular context of this study:*

*The main intention of this study was to develop, implement, validate and demonstrate the mathematically correct propagation algorithms for bending angle retrieval, under as few assumptions as necessary for a successful application of the uncertainty algorithm to RO data in the rOPS; so same intention as Schwarz et al. Earth Space Sci. (2017) for refractivity and dry-air retrieval. The quality and realism of the propagation results in absolute terms will of course strongly depend on the uncertainty profiles used as input at excess phase level; as also the test-day*

*ensemble results indicate. Since for this study realistic full-profile rOPS L1a uncertainty estimates were not yet available at excess phase level, we decided for viable proxy input profiles with realistic order of magnitudes.*

*Regarding the altitudes below 30 km, we chose to replace the noise-estimation scheme by a simple linear model below 30 km, because the empirical estimation approach from the noisy time series (as described in section 2.2, esp. Eq. 5 and the related text) becomes increasingly vulnerable to biases and fluctuations below 30 km due to the strong (near-exponential) increase of the excess phase magnitude. The simple linear model chosen was in fact consistent with that ESA/EUMETSAT reference as well as roughly with the overall behavior of COSMIC-profile based estimates below 30 km (which were not stable enough to be used directly). Regarding specifically the simplified Eq.8-modeling of the systematic excess phase uncertainty we also point to the algorithm demonstration focus of this study, rather than very realistic input and output; and also in this specific respect we will of course use the more realistic rOPS L1a-provided estimates once available.*

*To be more clear with our rationale for the construction of the input profiles used, we updated on p6, L26-28, to: "While in future the excess phase random and systematic uncertainty profiles will be more rigorously estimated by the rOPS L1a processor (Innerkofler et al. 2016) and provided as input to the L1b processor, they had to be estimated for this study from existing excess phase profiles with realistic noise (we chose UCAR/CDACC ones) and simplified modeling. To this end..."
and on p7, L7, to "…roughly following estimates of ESA/EUMETSAT (1998) and the overall behavior of estimates from real excess phase profiles (the latter became too vulnerable to biases and fluctuations to continue using them below 30 km)."
and on p7, L24, to "…linear uncertainty gradient in the troposphere; as noted above this simplified modeling will be replaced in future by realistic uncertainty estimates received as L1b retrieval input from the L1a processor (Innerkofler et al., 2016)."*

*In general, we will be able to use real uncertainty profiles from the L1a uncertainty propagation when the L1a/L1b interfacing is complete, and we then clearly intend to inter-validate the RO data and uncertainty estimates from the full rOPS chain also against other sources of quality data like from RAOBs and ECMWF.*

*Second, as response regarding the question of separate estimation of systematic and random uncertainties:*

*The criteria for separating the evaluation and propagation of estimated random and systematic uncertainties is that the processes giving effect to these uncertainties need to be uncorrelated (if the correlation term is zero, the other terms can be pulled out of the sum). This is independent of the shape of the probability distribution (if a process causes a biased mean of the pdf, it is counted to the (basic) systematic uncertainties, such that the random uncertainty can be assumed unbiased). The basic systematic uncertainty is caused by processes that repeated measurements under (almost) the same conditions could not detect, e.g. insufficiently defined physical constants, most model parameter errors, etc. It can thus not be correlated with effects of processes that give different results under repeated measurements.*

*The random und apparent systematic uncertainties on the other hand can be seen as uncorrelated for an analogous reasoning: Since we separate them based on*

*correlation length, the apparent systematic uncertainty appears as a bias within the, e.g., 0.8 seconds of the Blackman window (correlation coefficient of unity over the window range), while the random uncertainties' correlation function typically decays down to zero within this range. So even if random and (apparent) systematic uncertainty were correlated at one side of the Blackman window, they will be clearly uncorrelated at the other. It is crucial, however, that the correlation lengths of the random and the systematic uncertainties remain clearly separable; would the random uncertainty be strongly correlated also over longer time windows (appearing as a bias), it could be also correlated to the systematic uncertainty (a situation that, favorably, does essentially not occur in practice for RO data processing).*

*Thus, in practice, while there might be (very) slight correlations between random and systematic uncertainties at times, we consider the additional insight from having random and systematic effects separately available more important than the problem of potentially (very) slightly overestimating the uncertainties of our RO profiles.*

*In order to further help understand this, we now added on p4, after L20, another sentence/small paragraph as follows: "Since the noise-type effects giving rise to short range-correlated random uncertainties can be considered uncorrelated to the bias-type effects inducing long range-correlated apparent systematic uncertainties, and since both are uncorrelated to basic systematic uncertainties, it is insightful and possible with due care to estimate and propagate each of these uncertainties independently."*

2. Although the MC simulations validate the propagation process, whether the propagation results can reflect how real data behave is questionable:

(i) One thing I concern the most is the modeling of **random uncertainty as normal distribution**. While the "residual phase" of the RO signal suffered by thermal noise could be normally distributed (strictly speaking it is not), the excess phase calculated by unwrapping residual phase can contain cycle slips (even bias if the used model is biased) due to signal noise. Obviously the nonlinear unwrapping process is ignored in this article, and I'm wondering if it could impact the uncertainty propagation results? If this has already been considered in the eq. 6, then author should explain how this model is derived rather than simply providing a technical report reference.

*In line with our response to the previous comment, we like to refer back first to the argument that our input quantities, including the input uncertainty estimates, will improve in their realism when the L1a part of the propagation chain is available. Second, we of course share the theoretical view that the assumption of a normal distribution is important for the formal validity (and mathematical proof) of the (linear) covariance propagation, but practically – since we remove biases when known, as recommended by the GUM, and therefore have essentially unbiased pdf-means – only skewness could be a (small) issue. Since our uncertainty estimates need not be highly accurate quantitatively, however (several % relative accuracy of an uncertainty estimate is very good already), minor pdf-skewness effects will not be critical.*
*We also note that the overall combined noise – given that the central limit theorem implies that the superposition of individual non-Gaussian noise sources approaches a Gaussian pdf for the overall noise – can be expected to be fairly close to Gaussian*

*(which also RO data statistics indicate). We certainly will closely expect the adequacy of the L1a excess phase uncertainty estimates when available, including the effectiveness of cycle slip correction and other potential systematic effects, and their related residual uncertainty.*

*In order to better reflect in particular the normal-distribution issue also in the text, where we mention the various main noise sources, we have improved on p4, L29ff, to "…the receiving system noise (i.e., thermal noise and residual clock estimation noise) and the ionospheric noise (from scintillations induced by ionospheric irregularities) are essentially normally distributed overall (Kursinski…). These noise sources are the main contribution to…"*

(ii) The random uncertainty is highly related to the signal SNR. However, the linear extension used below 30 km removes all the corresponding SNR information. The reason of using **a linear gradient model below 30 km** instead of the calculated ddL_{rm,k} should be given.

*We have included the response to this comment in our answer to comment 1 above; please see there.*

(iii) A key element this article lack of is the **verification of the propagated uncertainty using the actual data**. The direct comparison in random uncertainty might be difficult, but the system uncertainty, or bias as you defined in P.4, could be observed statistically through the comparison between RO and ECMWF (or other measurement like RAOB). We may have more confidence on the propagated system uncertainty if it matches the comparison results.

[Sokolovskiy, S., C. Rocken, W. Schreiner, and D. Hunt (2010), On the uncertainty of radio occultation inversions in the lower troposphere, J. Geophys. Res., 115, D22111, doi:10.1029/2010JD014058.]

*We hope the first part of our response to comment 1 above, and the related text improvements in the manuscript noted there, addressed also this concern. As noted, we will validate the results of the uncertainty propagation also against real data (such as high quality data from RAOBs and ECMWF) when the L1a uncertainty estimates are available and interfaced to L1b. For context, we did compare the output of the rOPS L2a uncertainty propagation (shown in the Schwarz et al. Earth Space Sci. 2017 paper) to empirical/statistical estimates of uncertainties of dry temperature etc., and we also see that the output uncertainties of the L1b chain reasonably corresponds with the assumed input for the L2a chain, so we can be quite confident that our uncertainty propagation will enable a sequential chain of estimates of appropriate quality.*

**Specific comments**

***P7,eq. 6&eq. 8***

These two equations should be better explained: why linear and where are these constants (3e6 and 3e7) come from? Why eq. 6 is better than the original ddL_{rm,k} in modeling the random uncertainty below 30 km?

*Please see the response to comment 1 above and the related text improvements before and near Eqs. 6 and 8 that we have quoted there and included in the revised manuscript.*

*** P7, L23 – L26 ***

Why 0.1 mm and 0.2 mm for simMetOp? Why 0.2 and 0.4 mm for the other two? Why are they constants over 8 km to 80 km? What are the causes of the modeled systematic uncertainty?

*As explained as part of the response to comment 1 above, this is a simplified model; see also the text improvement now on p7, L24ff, that we quote there, which now explicitly states the fact that this is a simplified modeling that will be replaced in future by the realistic input from the L1a processor.*
*Regarding the different settings in the current simple model for CHAMP and COSMIC, compared to (sim)MetOp, we added now on p7, L26, as follows "For CHAMP and COSMIC we set us_Lr1 = 0.2 mm and us_Lr2 = 0.4 mm, to roughly reflect the fact that these RO receivers are lower-cost instruments with lower gain, and thus somewhat lower tracking performance, than the GPS receiver GRAS on MetOp (e.g., Luntama et al., 2008; Angerer et al., 2017)."*

*[Note: Angerer et al., 2017, which inter-compares performances from different RO missions, including CHAMP, COSMIC, and MetOp, is included as a new reference: Angerer, B., Ladstädter, F., Scherllin-Pirscher, B., Schwärz, M., Steiner, A. K., Foelsche, U., and Kirchengast, G.: Quality aspects of the Wegener Center multi-satellite GPS radio occultation record OPSv5.6, Atmos. Meas. Tech., in press, doi:10.5194/amt-2017-225, 2017.]*

*** P11, L18 ***

$F_{c2}$ is set noise dependent – How to determine the filter bandwidth? Do you have to check the spectrum first?

*As described in Sokolovsky et al. (2009), and in some detail also in the current study in the Appendix section "A.3.1 Adpative Lowpass Filtering…", the filter bandwidth is derived by choosing from a set of cutoff frequencies, based on a minimization algorithm (i.e. there is a loop through the following steps for each frequency: 1. the frequency is chosen, 2. the filter is applied, 3. the ionospheric correction is applied, 4. the noise of the resulting bending angle profile calculated, 5. if smaller than for the preceeding frequency, the profile is kept, 6. the next frequency is chosen).*

*In the rOPS case, the set of cutoff-frequencies are pre-determined, constant (using the cutoff frequency of the pre-doppler BWS filter as upper bound for the highest frequency), and are independent of the spectrum of the individual event. For the more detailed related description see the Appendix section A.3.1 (p26).*

*** P14, L8 ***

What is the criteria used for discarding 5% of the processed profiles?

*The QC applied to the uncertainty propagation results was based on the magnitude of the maximum of the $u^r\_Lr$ estimate, which appeared to let about 5% of the profiles to be discarded.*
*To better express this we updated the sentence on p14, L8-9, to:* ==*"…because they were detected as outliers based on the magnitude of their random uncertainty profiles (these outliers are not included in the number of profiles shown)."*==
*We note that this was just a temporary solution for this L1b algorithm intro study; a more advanced QC is integrated as part of the rOPS L1a/L1b interfacing.*

*** P20, L7 ***

Although the conclusion is the same but shouldn't the BWS filter be 41 points as you stated in P. 16?

*Note that we talk in this sentence about the "effective filter width" of eleven points, i.e., the width similar in its smoothing to an eleven-point boxcar filtering. For a more detailed description see Appendix section A1.1 (including explanation also of Figure A1): the window width at half-maximum of the filter function of the 2.5 Hz BWS filter (with a Blackman window of 41 points) corresponds to the filter function of a boxcar filter of about 11 points window width.*

*** P23, eq. A33 ***

Why the systematic uncertainty of bending angle is not related to the open angle?

*Thank you for spotting this and pointing this out; of course the uncertainty is also related to the opening angle, the corresponding equation (A33) was not properly updated, it is now corrected (p23, L20).*

*** P41, Fig. 6(b) ***

When comparing Fig.5(b) and Fig.6(b), it surprised me that the Doppler systematic uncertainty increase at the bottom of the profile vanished in the one of bending angle. In figure 6, it's just a constant all the way down. Is there any specific reason for this? The Doppler uncertainty is too small compared to the orbit uncertainties? So most of the systematic uncertainty of the bending angle comes from the orbit instead of the measured Doppler?

*Exactly, given our input uncertainties for the excess phase and the orbit position and velocities, the uncertainty component coming from the systematic uncertainty of the receiver velocity is typically about 2 magnitudes larger than the systematic uncertainty from the Doppler shift. The receiver position uncertainty has about the*

*same magnitude as the receiver velocity uncertainty, and the transmitter uncertainties are about 1-2 magnitudes smaller.*

*To point to this already in the relevant text in section 3.2, we added a brief sentence on p10, L16, as follows: "...(Figure 6b). Compared to this magnitude, the systematic uncertainty contributed by the Doppler shift uncertainty is very small."*

\*\*\* P45, Fig 10(g) \*\*\*

Can you provide an explanation why several cases in simMetOp have larger L_f uncertainty between the impact altitude of 40 and 60 km?

*These three simMetOp cases seem to have a somewhat different, but not completely uncommon noise characteristic, stemming from the noise superposition done in the simulation of the excess phase profiles. Some of the discarded simMetOp outliers show similar characteristics, and quite a range of the discarded COSMIC and CHAMP profiles do (and also several remaining ones, as visible in Fig. 10 as well).*

*We have now included on p15, L29ff, a sentence of explanation on this point, as follows: "Three individual profiles exhibit comparatively high uncertainties of larger than 2 mm within about 40 to 55 km, however, reflecting that the simMetOp error simulations are capable to partly generate higher-noise profiles of the type more frequently seen in the real MetOp data (Figure 10g)."*
*[Note: the revised manuscript now includes also test-day results for real MetOp data; so this sentence is part of further text updates in this part of the manuscript, based upon suggestion by Reviewer #1]*

---

## Author Response (AR1)

[revised manuscript text omitted]
_{\alpha r}^r$ is therefore considerably larger than $u_{\alpha F1}^r$ and $u_{\alpha F2}^r$ (cf. Figure 8c and 7c).

Figure 8d shows how the correlation functions—as obtained through covariance propagation—are combining the characteristics of the correlation functions from the two matrices $R_{\alpha F1}$ and $R_{\alpha F2}$, with essentially inheriting the $\alpha_{F1}$ behavior, since the $\alpha_{F2}$ influence into the ionospheric correction is comparatively minor (see Section A3).

The residual higher order ionospheric effects are accounted for by a 'conservative best-guess' value ($0.05\,\mu$rad, reflecting results of Liu et al. (2015) and Danzer et al. (2013, 2015)) and added (in root-mean-square form) to the systematic uncertainty profile $u_{\alpha r}^s$, leading to a total estimated systematic uncertainty in this example case of $\sim 0.07\,\mu$rad (Figure 8b). Within this uncertainty, the one dominating component from orbit uncertainties ($\sim 0.05\,\mu$rad, cf. Figure 6b) can be considered an apparent systematic uncertainty that will essentially average out in ensemble-averaging (e.g., climatologies) while the other dominating component from residual higher-order ionospheric biases (also estimated $\sim 0.05\,\mu$rad as noted above) can be considered a basic systematic uncertainty. For the latter it is therefore useful and prepared for in the rOPS—in line with GUM recommendations and as discussed in the introductory Section 1—to correct for the quantifiable part of it in the future so that the total basic systematic uncertainty may be mitigated down to the $\sim 0.01\,\mu$rad level.

The resolution profile $w_{\alpha r}$ of the retrieved bending angle (Figure 8f) is dominated by the contribution of $\alpha_{F1}$ that strongly dominates (intentionally by construction) the ionospheric correction results in terms of the small-scale bending angle variability. Similar as for the correlation length profile $l_{\alpha r}$ it is therefore very close to $w_{\alpha F1}$ and only slightly larger.

**4 Algorithm Validation**

The GUM advises to use a Monte-Carlo (MC) method for uncertainty propagation if the retrieval operators do not fulfill the criteria for a GUM-type CP. In our case the MC method is put to another beneficial use, to validate the results of the CP, as recommended by JCGM (2011).

For the validation of the covariance propagation by the MC method, we sampled the input excess phase profile random error distribution, statistically described by

$$\mathbf{C}_{Lr}^{MC} = u_{Lr,i}^{r,STD} \cdot u_{Lr,j}^{r,STD} \cdot \delta_{ij}, \tag{9}$$

by a large number $M$ of draws $L_r^T + \epsilon_{Lr,j}^r$ (with $j \in \{1,...,M\}$ and $M = 1000$). For each of these $M$ profile realizations, the state retrieval is run through the L1b retrieval chain, to give $M$ realizations of the output variable $X_j$ (with $X_j \in \{L_{F,kj}(t),$ $D_{r,kj}(t), \alpha_{G,kj}(z_a), \alpha_{F,kj}(z_a), \alpha_{r,j}(z_a)\}$ and $k \in \{1,2\}$). From these individual realizations the mean profiles $X^{MC}$, and the

covariance matrices $\mathbf{C}_X^{\mathrm{MC}}$,

$$\mathbf{C}_X^{\mathrm{MC}} = \frac{1}{M-1}[(X_1 - X^{\mathrm{MC}})(X_1 - X^{\mathrm{MC}})^T + ... + (X_M - X^{\mathrm{MC}})(X_M - X^{\mathrm{MC}})^T)], \tag{10}$$

are calculated (Items b-g in Figure 2). Using the same input profile and uncertainty information as used to specify the MC runs (described in Section 2.2), the retrieval is then also run with covariance-based uncertainty propagation and

5  the resulting CP-propagated covariance matrices $\mathbf{C}_X^{\mathrm{CP}}$ are compared to the MC-derived matrices $\mathbf{C}_X^{\mathrm{MC}}$. In order to be able to attribute potential changes between CP and MC covariance matrices better, we decompose $\mathbf{C}_X$ into $u_X^r$ and $\mathbf{R}_X$ (Equations A7 and A8), and compare them separately.

Figure 9 shows the different steps along the retrieval chain from $L_{\mathrm{F},k}(t)$ to $D_{\mathrm{r},k}(t)$, $\alpha_{\mathrm{G},k}(z_a)$, $\alpha_{\mathrm{F},k}(z_a)$, and $\alpha_{\mathrm{r}}(z_a)$ in the rows, for $k = 1$ (GPS $f_{\mathrm{T}1}$ frequency) in the left column and for $k = 2$ (GPS $f_{\mathrm{T}2}$ frequency) in the middle column. The right

10  column shows multiple representative correlation functions, from near $10\,\mathrm{km}$ to near $70\,\mathrm{km}$. Due to the limited number of MC draws, the MC results (black lines) show some jitter both in the estimated random uncertainty and in the correlation functions. Since the purpose of the MC results is only to demonstrate the correctness of the CP result, we can disregard this behavior.

Figures 9a (light blue) and 9b (orange) show the random uncertainties $u_{L\mathrm{r},1}^r$ and $u_{L\mathrm{r},2}^r$ respectively, which characterize the input distribution and from which the random error profiles $\epsilon_{L\mathrm{r},j}^r$ are drawn. They also show the CP results for the random

15  uncertainty $u_{L\mathrm{F}1}^r$ (dark blue in Figure 9a) and $u_{L\mathrm{F}2}^r$ (red in Figure 9b), compared to the MC propagated random uncertainties (black).

The CP and MC lines match very well and show that the implemented CP algorithm delivers correct results for the basic filtering step. For $f_{\mathrm{T}2}$, the MC uncertainties do not reach down as far as the CP uncertainties, because the shortest of all draws of the large ensemble of size $M$ determines how far down the recombined MC covariance matrix (Equation 10) reaches.

20  Figure 9c compares CP correlation functions $R_{L\mathrm{F},i1}$ (blue) and $R_{L\mathrm{F},i2}$ (red) to the corresponding MC correlation functions (black dashed).

Also the CP and MC correlation functions agree well. Both capture the narrow peak, broadened by the BWS filter. Again the MC correlation functions fluctuate around zero left and right of the peak, from the finite ensemble size, but it is obvious that the CP delivers the correct off-peak results (i.e., zero; the off-peak elements outside the BWS filter window must nominally be

25  zero). The MC validation (black) of $u_{D\mathrm{r}1}^r$ (Figure 9d), $u_{D\mathrm{r}2}^r$ (Figure 9e) and $R_{D\mathrm{r},
[revised manuscript text omitted]

and

15    $$R_{Dr,ij} = \frac{C_{Dr,ij}}{u_{Dr,i}^r u_{Dr,j}^r}. \tag{A20}$$

For the *estimated systematic uncertainty*, further on interpreted as basic systematic uncertainty (cf. Equation A10), we apply the derivative operator (Item 1.4 in Figure 2) to the systematic uncertainties, with no zero-order profile subtracted, i.e.,

$$u_{Dr,i}^s = \sum_{j=1}^{N} A_{ij}^{\mathrm{L2D}} \cdot u_{LF,j}^s. \tag{A21}$$

The *resolution* remains unaffected by the Doppler shift derivation, since the five-point sample width of the derivative operator
20    is fully within the eleven-point effective filter width (stopband) of the BWS filter applied before, so that $\tau_{Dr} = \tau_{LF}$ and $w_{Dr} = w_{LF}$.

**A2 Bending Angle Retrieval**

**A2.1 GO Bending Angle Retrieval**

From the Doppler shift *state* profile $D_r$ (again for both frequencies of the given GNSS system) we can derive the impact parameter profile $a_t$ and geometric-optics (GO) bending angle profile $\alpha_G$ (Item 2.1 in Figure 2) using first the geometric relation

$$D_{r,i} = [v_{R,i}\cos(\phi_{R,i}) - v_{T,i}\cos(\phi_{T,i})] - \dot{r}_{RT,i}, \tag{A22}$$

where

$$\phi_{R,i} = \eta_{R,i} - \arcsin\left(\frac{a_{t,i}}{r_{R,i}}\right), \tag{A23}$$

and

$$\phi_{T,i} = (\pi - \eta_{T,i}) - \arcsin\left(\frac{a_{t,i}}{r_{T,i}}\right) \tag{A24}$$

for each individual level of the time grid $t_i$, in order to determine $a_t$ from sequential application to all levels (Kursinski et al., 1997; Syndergaard, 1999). Here $v_{R,i} := |\boldsymbol{v}_{R,i}|$ is the receiver velocity, $r_{R,i} := |\boldsymbol{r}_{R,i}|$ the receiver radial position, $\eta_{
[revised manuscript text omitted]

$$k_{at,i} := \frac{\partial D_\mathrm{r}}{\partial \phi_\mathrm{R}}\bigg|_i \cdot \frac{\partial \phi_\mathrm{R}}{\partial a_t}\bigg|_i + \frac{\partial D_\mathrm{r}}{\partial \phi_\mathrm{T}}\bigg|_i \cdot \frac{\partial \phi_\mathrm{T}}{\partial a_t}\bigg|_i = -v_{\mathrm{R},i} \cdot \sin\phi_{\mathrm{R}i} \cdot \frac{1}{\sqrt{r_{\mathrm{R},i}^2 - a_{t,i}^2}} - v_{\mathrm{T},i} \cdot \sin\phi_{\mathrm{T}i} \frac{1}{\sqrt{r_{\mathrm{T},i}^2 - a_{t,i}^2}}, \tag{A32}$$

$$k_{v\mathrm{R},i} := -\frac{\partial D_\mathrm{r}}{\partial v_\mathrm{R}}\bigg|_i = -\cos\phi_{\mathrm{R}i},$$

$$k_{v\mathrm{T},i} := -\frac{\partial D_\mathrm{r}}{\partial v_\mathrm{T}}\bigg|_i = -\cos\phi_{\mathrm{T}i},$$

$$k_{r\mathrm{R},i} := \frac{\partial D_\mathrm{r}}{\partial \phi_\mathrm{R}}\bigg|_i \cdot \frac{\partial \phi_\mathrm{R}}{\partial r_\mathrm{R}}\bigg|_i = \frac{v_{\mathrm{R},i} \cdot \sin\phi_{\mathrm{R}i} \cdot a_{t,i}}{r_{\mathrm{R},i}\sqrt{r_{\mathrm{R},i}^2 - a_{t,i}^2}},$$

$$\quad k_{r\mathrm{T},i} := \frac{\partial D_\mathrm{r}}{\partial \phi_\mathrm{T}}\bigg|_i \cdot \frac{\partial \phi_\mathrm{T}}{\partial r_\mathrm{T}}\bigg|_i = \frac{v_{\mathrm{T},i} \cdot \sin\phi_{\mathrm{T}i} \cdot a_{t,i}}{r_{\mathrm{T},i}\sqrt{r_{\mathrm{T},i}^2 - a_{t,i}^2}}.$$

A number of simplifications have been made to arrive at this result. First, the last term in Equation A22 is disregarded since errors in the positions are assumed to be constant with respect to the short time duration of an RO event; remaining errors $\Delta \dot{r}_\mathrm{RT}$ after taking the derivative are therefore of higher order. Next, orbit position and velocity uncertainties are both assumed to be constant within the short duration of an event and the velocity uncertainties obtained are interpreted as uncertainties along

10     the direction of the velocity vector. Consequentially, the uncertainty is also projected along with the vector into the raypath direction. A more conservative estimation (that we consider overly conservative in context) would interpret the uncertainties as ellipsoids at the velocity vectors' heads, and would hence take the full magnitude of the uncertainties along the raypath direction (without projection).

Furthermore, since all error sources (the processing of the occultation tracking data and the POD for transmitter and receiver)

15     are essentially independent from each other, the different input uncertainties are assumed to be uncorrelated. Finally, we reasonably assumed the errors of the angle between the position and velocity vectors ($\eta$) to be negligible ($u_\eta^s \approx 0$) for the purpose here, for both the transmitter and receiver.

In order to finally derive the systematic uncertainty of the bending angle from the impact parameter's uncertainty, we continue with a linearization of Equation A25 and arrive at

$$u_{\alpha\mathrm{G},i}^s = \sqrt{(k_{at,i} \cdot u_{at,i}^s)^2 + (k_{r\mathrm{R},i} \cdot u_{r\mathrm{R},i}^s)^2 + (k_{r\mathrm{T},i} \cdot u_{r\mathrm{T},i}^s)^2}\sqrt{(u_{\theta\mathrm{RT},i}^s)^2 + (k_{at,i} \cdot u_{at,i}^s)^2 + (k_{r\mathrm{R},i} \cdot u_{r\mathrm{R},i}^s)^2 + (k_{r\mathrm{T},i} \cdot u_{r\mathrm{T},
[revised manuscript text omitted]
_{\mathrm{F1}}$ and $z_{a\mathrm{Bot2}} \leq z_{a\mathrm{Bot2Max}}$ (with $z_{a\mathrm{Bot2Max}}$ currently set to 15 km), a *tropospheric bending angle extrapolation* (TBAE) is applied in order to artificially extend $\alpha_{\mathrm{M2}}$ to also reach down to $z_{a\mathrm{Bot1}}$ (Item 3.3 in Figure 2).

Briefly summarized, this TBAE is currrently implemented as follows. A linear gradient profile for the difference profile be-
5 tween the two bending angles, $\alpha_{\mathrm{F12}} = (\alpha_{\mathrm{F1}} - \alpha_{\mathrm{F2}})$, is estimated by a least squares fit over a sufficiently wide impact altitude range from $z_{a\mathrm{Bot2}}$ upward (as wide as the extrapolation range, at least 10 km). This gradient profile is then linearly extended down to $z_{a\mathrm{Bot1}}$ and subtracted from $\alpha_{\mathrm{F1}}$, to obtain the extrapolated part of $\alpha_{\mathrm{F2}}$ from $z_{a\mathrm{Bot2}}$ to $z_{a\mathrm{Bot1}}$. If $z_{a\mathrm{Bot2}} > z_{a\mathrm{Bot2Max}}$ then no TBAE is performed since the extrapolation range is considered too large. Details are provided by Kirchengast et al. (2017b), where the most recent version of the atmospheric bending angle derivation is described that includes this $\alpha_{\mathrm{F12}}$ ex-
10 trapolation in a further advanced form.

For the propagation of the *estimated random uncertainty* we get (Item 3.2 in Figure 2),

$$\mathbf{C}_{\alpha\mathrm{F},k} = \mathbf{A}_k^{\mathrm{BWS}} \cdot \mathbf{C}_{\alpha\mathrm{M},k} \cdot (\mathbf{A}_k^{\mathrm{BWS}})^{\mathrm{T}}, \tag{A42}$$

for the bending angle error covariance matrices of the leading ($k = 1$) and minor ($k = 2$) channel.

In case a TBAE is applied to $\alpha_{\mathrm{F2}}$, the random uncertainty of $\alpha_{\mathrm{F2}}$ below $z_{a\mathrm{Bot2}}$ is equal to the one of $\alpha_{\mathrm{F1}}$, because the noise
15 is "copied" from $\alpha_{\mathrm{F1}}$ since the linear gradient profile from fitting $\alpha_{\mathrm{F12}}$ is noise-free. As a consequence, in these cases, we set the matrix elements of $\mathbf{C}_{\alpha\mathrm{F2}}$ to (Item 3.4 in Figure 2)

$$\mathrm{C}_{\alpha\mathrm{F2},ij} = \begin{cases} \mathrm{C}_{\alpha\mathrm{F2},ij} & \text{for } z_{a\mathrm{Top}} > z_{a,i} > z_{a\mathrm{Bot2}} \text{ and } z_{a\mathrm{Top}} > z_{a,j} > z_{a\mathrm{Bot2}} \\ \mathrm{C}_{\alpha\mathrm{F1},ij} & \text{for } z_{a\mathrm{Bot2}} > z_{a,i} > z_{a\mathrm{Bot1}} \text{ and } z_{a\mathrm{Bot2}} > z_{a,j} > z_{a\mathrm{Bot1}} \\ 0 & \text{for } z_{a\mathrm{Bot2}} > z_{a,i} > z_{a\mathrm{Bot1}} \text{ and } z_{a\mathrm{Top}} > z_{a,j} > z_{a\mathrm{Bot2}} \\ 0 & \text{and } z_{a\mathrm{Top}} > z_{a,i} > z_{a\mathrm{Bot2}} \text{ for } z_{a\mathrm{Bot2}} > z_{a,j} > z_{a\mathrm{Bot1}} \end{cases}. \tag{A43}$$

$\mathbf{C}_{\alpha\mathrm{F1}}$ and $\mathbf{C}_{\alpha\mathrm{F2}}$ can then be decomposed as needed into $u_{\alpha\mathrm{F1}}^r$, $\mathbf{R}_{\alpha\mathrm{F1}}$, and $u_{\alpha\mathrm{F2}}^r$, $\mathbf{R}_{\alpha\mathrm{
[revised manuscript text omitted]
 (redviolet)). Left column: Mean random uncertainty $u^r_{X\text{r}}$ (heavy) and mean systematic uncertainty $u^s_{X\text{r}}$ (light) profiles (panels a, d, g, j); the latter shown as $10 \times u^s_{L\text{F1}}$ (in a) and $100 \times u^s_{X\text{r}}$ (in d, g, j) for enabling visibility of these small quantities. Middle column: Correlation length profiles $l_{X\text{r}}$ (panels b, e, h, k). Right column: Vertical resolution profiles $w_{X\text{r}}$ (panels c, f, i, l).

[Figure]

**Figure A1.** Comparison of the Blackman windowed-sinc (BWS) lowpass filter and boxcar (BC) filters based on a representative segment (between 30.3 and 32.7 s) of the excess phase profile $L_{r1}$ of the COSMIC example event. Panel (a): Filter functions for the BWS filter with $f_c = 2.5\,\mathrm{Hz}$ and $M = 41\,\mathrm{pts}$ ('BWS', red) and boxcar filters with $M = 21\,\mathrm{pts}$ ('BC21', green) and with $M = 11\,\mathrm{pts}$ ('BC11', blue), around the central value of the segment (31.5 s). Panel (b): Filter effects on the excess phase profile $L_{r1}$ from running the filters over the segment. Shown are the unfiltered excess phase delta profile ('$\delta L_{rm}$', light gray), the BWS filtered profile with $f_c = 2.5\,\mathrm{Hz}$ and $M = 41\,\mathrm{pts}$ ('$\delta L_{Fm}^{BWS}$', red), and the Boxcar filtered profiles with $M = 21\,\mathrm{pts}$ ('$\delta L_{Fm}^{BC21}$', green) and $M = 11\,\mathrm{pts}$ ('$\delta L_{Fm}^{BC11}$', blue), respectively.

---

## Referee Report (RR1)

**Response to Referee #1**

Manuscript "Integrating uncertainty propagation in GNSS radio occultation retrieval: from excess phase to atmospheric bending angle profiles"

by Jakob Schwarz, Gottfried Kirchengast, and Marc Schwaerz,
AMT Discussions paper, doi:10.5194/amt-2017-159

==================================================================

*We thank the reviewer very much for the constructive and detailed feedback to our manuscript. We thoroughly considered all comments and carefully revised the manuscript accounting for most of them. Below are our point-by-point responses.*

*Comments by the reviewer are cited* black upright*, our responses are* blue italic. *(line numbers used in our responses refer to the original AMT Discussions paper and text updates in the revised manuscript are quoted below with* yellow highlighting*)*

==================================================================

*NEW COMMENTS ARE IN* RED

==================================================================

**General comments**

The paper details the algorithms, the underlying assumptions and the results of the uncertainties estimation (random and systematic) in the GNSS radio occultation (RO) retrieval, from the excess-phases to bending angle profiles. The paper, together with other similar papers which address the same topic once applied to the other portion of the entire RO processing chain (thus describing error propagation through the POD solution and through the retrieval steps of atmospheric profiles starting from bending angles), provided for the first time a rigorous mathematical overview of how errors are propagated through the entire RO processing chain. From the scientific point of view the paper is almost complete; involved data, algorithms, results, discussion and conclusions are presented in a well structured and clear way, even if the paper is excessively long.

In what follows some general comments are provided.

1. One of the most critical step of the RO processing chain is the retrieval of bending angles in the lower troposphere, which involves more "advanced" methods (the so called **Wave Optic retrieval**). Unfortunately error propagation through this step is not detailed (even if a paper is already available since 2015, it is written a couple of times in the manuscript that the implementation of WO uncertainty propagation in the rOPS is not yet finished... so it is not fully clear whether the results are already available or not). In order to have a complete and more coherent overview, this part is somehow necessary. My suggestion is, if results are not available yet, to remove from this paper all the discussions provided on the WO retrieval and error propagation through

it (Sect 3.2.2, Sect 3.2.3, Annex A2.2, Annex A2.3), changing a bit also the title reflecting that only GO retrieval is applied.

*We carefully checked the impact of this alternative of dropping the link to WO in this paper and found it would do no good to the readers if this paper would not clearly show how the WO uncertainty work as introduced by Gorbunov and Kirchengast (2015; 2017) is embedded in the overall bending angle retrieval, including in particular the merging of WO and GO and of their uncertainties. That is, even though the testing and integration-test approval of the WO uncert.prop. in the rOPS— implemented as discussed by Gorbunov and Kirchengast (2017)—is not yet completed so that we could not yet use the WO results in this study, we know with confidence that the subsequent algorithm step in the rOPS uncert,prop., the merging of GO and WO uncertainties, can be mathematically implemented and tested independently. This holds, since only if the WO uncertainty propagation would deliver large outliers with unexpected behavior there might be problems; but we know from WO tests this not to be the case. Also, since we tested a whole day of real and simulated data with a relatively large variability in u^r (and offline with some even larger test magnitudes), the stability of the merger-algorithm itself is evident for any reasonable magnitude of the uncertainty profiles delivered by the WO uncert.prop. In other words, using in this paper the uncertainties from the GO also as WO input for demonstrating the merger, allows us to have this way a sound "proxy" to reasonably cover the possible variability of the input uncertainties coming in future from WO.*

*We realize, though, that we need to update the text at several places (given that the Gorbunov and Kirchengast 2017 AMT paper is meanwhile in press and the rOPS implementation of WO is meanwhile in integration tests), aiming to make the value of linking to WO in this paper and also the referencing more clear. We note that—similar as done in Schwarz et al. Earth Space Sci. (2017) for the refractivity and dry-air retrieval—the focus of this study is on the careful introduction of the (formally correct) algorithms and on the demonstration of the robust performance, and less on the delivery and presentation of real/realistic uncertainty propagation results (see also our response to Comment 3. below). So we think that presenting the successfully implemented merger algorithm makes sense and is useful to the readers of this study. It also will allow for a more concise presentation of the WO uncertainty propagation as part of the complete chain in a future publication that will then show real-data results from the complete rOPS chain.*

*We updated the text that refers to the uncert.prop. through the WO bending angle retrieval and the merging operator particularly in the following parts:*

*On p11,L3, we updated and improved to "Because the rOPS implementation of the WO uncertainty propagation (Gorbunov and Kirchengast, 2017) was still in test phase and not yet available for integration into the simulations here, all examples in this study…"*

*and on p11, L6, to "In order to nevertheless test and validate the uncertainty propagation of the merging algorithm, WO retrieval results were artificially substituted by the GO results for the MC validation (Section 4), i.e., GO was used as proxy for*

*WO since reasonably capturing expected WO variability as indicated by tests of Gorbunov and Kirchengast (2017)."*

*and on p24, L18, to "…prepared the merging with the WO bending angle variables (they will be actually merged in when the WO tests within the rOPS is complete), which is described next."*

*Ok. I agree with this idea and with the applied changes*

2. Another general issue I want to highlight is that the entire discussion and the main results shown from Fig 3 to Fig 8 are based considering an excess-phases "time series" and associated random/systematic uncertainties observed during **one real occultation event from the COSMIC mission, taken as representative for the entire day**. Not sure that the heuristic associated to the choice of that particular profile is correct, since the ensemble mean profiles of the uncertainties associated to the various variables involved in the retrieval (Fig 10 and Fig 11) show quite different behavior (see for example the increasing of uncertainties with height). It would be nice to have different examples (maybe the best and the wrong together with this most representative one).

*With Figures 3 to 8 we followed the same didactical approach in introducing the algorithmic steps in combination with some illustrative figures for an example event as we did in the Schwarz et al. Earth Space Sci. (2017) work. The key aim of the algorithm description in chapter 3 and the figures 3-8 is to illustrate the effects of the operators in the rOPS uncertainty and not to show the input-to-output profiles uncert.prop. performance through the chain (this is for the later figures with MC validation and full test-day results). It is for this aim that we found it most clear not to overload these illustrative figure with complex information of more than one example event (or have otherwise even more figures in what is already a quite extensive paper).*

*In this study we chose one real profile as example profile, rather than a simulated profile as in the Earth Space Sci. paper, to capture for bending angle retrieval with its higher sampling rate more characteristics of real profiles typically processed. Still the chosen (COSMIC) example is only used to illustrate the effect of the operators, its random and systematic uncertainty is representative in terms of magnitude and correlation lengths for the profiles encountered in a real processing. It is not intended to be "representative" also in the sense that the results match a quantitative "best case", or "average", or similar. This was also found suitable, since the input profile uncertainties at excess phase level are as yet not the "real" input propagated from the rOPS L1a processing, but rather a proxy and "reasonable guess" input.*

*Regarding the test-day ensemble, the mean of the test-day random uncertainty profiles indeed increases between 70 and 75 km, but this effect is stemming mostly from a relatively small share of profiles with large random uncertainty, thus also not being particularly representative for the majority of profiles.*
*Concerning the use of the "time series" principle, we refer to our response to the reviewer's specific comment below related to "Page4, L9-10".*

*So overall we preferred to continue using and illustrating just one example event, based on the reasons above; also avoiding crowed plots or too many figures in total.*

*However, in order to avoid an interpretation of our example* profile we as *replaced „a representative" by „an example", on p3, L27* toned down „a representative" to „a typical" on p6, L9 (and from this „corresponding point approximately to the average uncertainty profile of the test-day RO events",* , and we *on the event is called „example case" anyway, as defined at the end of this sentence).*

*Ok. But it would be nice to have this clearly stated in the introduction. Could you please add a sentence which summarizes the* green highlighted part *of your text here above?*

3. Regarding the same input profile I found a bit inconsistent the **choice of extrapolating downward 30 km** the representative excess-phase random uncertainty profile (defined on the above mentioned COSMIC observation) with a linear behavior (I guess linear in order to provide a "worst" boundary, could you confirm this in the paper?), whose gradient follows some estimate defined for the GRAS instrument on board METOP satellites (and I don't think that in the provided reference [ESA/EUMETSAT] such information can be easily found). The systematic uncertainty on the same input variable is also defined following error estimates characterizing the GRAS receiver. The big question is why you mixed the random uncertainty "rigorously" derived considering a COSMIC observations with some heuristic based on the GRAS instrument? Was not better to derive the random error profile from GRAS real observations? And why for the GRAS case you demonstrated the uncertainty propagation using simulated data only? Please provide a clear motivation on this or repeat the entire analysis using some representative examples taken by GRAS excess-phases.

*The main intention of this study was to develop, implement, validate and demonstrate the mathematically correct propagation algorithms for bending angle retrieval, under as few assumptions as necessary for a successful application of the uncertainty algorithm to RO data in the rOPS; so same intention as Schwarz et al. Earth Space Sci. (2017) for refractivity and dry-air retrieval. The quality and realism of the propagation results in absolute terms will of course strongly depend on the uncertainty profiles used as input at excess phase level; as also the test-day ensemble results indicate. Since for this study realistic full-profile rOPS L1a uncertainty estimates were not yet available at excess phase level, we decided for viable proxy input profiles with realistic order of magnitudes in terms of variance and correlation length.*

*Regarding the altitudes below 30 km, we chose to replace the noise-estimation scheme by a simple linear model below 30 km, because the empirical estimation*

approach from the noisy time series (as described in section 2.2, esp. Eq. 5 and the related text) becomes increasingly vulnerable to biases and fluctuations below 30 km due to the strong (near-exponential) increase of the excess phase magnitude. The simple linear model chosen was in fact consistent with that ESA/EUMETSAT reference as well as roughly with the overall behavior of COSMIC-profile based estimates below 30 km (which were not stable enough to be used directly).

To be more clear with our rationale for the construction of the input profiles used, we updated on p6, L26-28, to: "While in future the excess phase random and systematic uncertainty profiles will be more rigorously estimated by the rOPS L1a processor (Innerkofler et al. 2016) and provided as input to the L1b processor, they had to be estimated for this study from existing excess phase profiles with realistic noise (we chose UCAR/CDACC ones) and simplified modeling. To this end..." and on p7, L7, to "…roughly following estimates of ESA/EUMETSAT (1998) and the overall behavior of estimates from real excess phase profiles (the latter became too vulnerable to biases and fluctuations to continue using them below 30 km)."

On the demonstration of the uncertainty propagation related to GRAS for simMetOp only, we agree that this was not really justified well (we had decided for it to save in particular some space in figures and text since we considered simMetOp is there anyway).
We therefore included real MetOp data now (updated Figures 10 and 11, and related text updates), in particular involving the following updates:

on p3, L29-31, to "…CHAllenging Minisatellite Payload (CHAMP) (Wickert et al., 2001), FORMOSAT-3 Constellation Observing System for Meteorology, Ionosphere, and Climate (COSMIC) (Anthes et al., 2008), and Meteorological Operational Satellite A (MetOp) (Luntama et al., 2008), and with simulated data…"
on p6, L4, to "For CHAMP, COSMIC and MetOp, orbit…" on p7, L24, to "…for MetOp and simMetOp." on p14, L2, to "…complete test-day of real (CHAMP, COSMIC, MetOp) and..."

on p14., L29ff, expand to "For the real MetOp data (available here as dataset from UCAR/CDACC, as for CHAMP and COSMC), urLF1 appears similar to COSMIC (cf. Figures 10d, g) while for simMetOp (with best possible simulated GRAS-type receiver noise) it is clearly smaller than for COSMIC. From 35 to 80 km the mean random uncertainty profile for simMetOp stays below 1 mm (Figure 10j). Three individual profiles exhibit comparatively high uncertainties of larger than 2 mm within about 40 to 55 km, however, reflecting that the simMetOp error simulations are capable to partly generate higher-noise profiles of the type more frequently seen in the real MetOp data (Figure 10g).
    On the other hand, the average correlation length/resolution profile of the ~500 real MetOp and ~700 simMetOp ensemble members is very similar, driven by the orbit being essentially the same for the real data and the simulations (Figures 10h, i, k, l). Compared to COSMIC (Figures 10e, f), the correlation length and resolution are again somewhat larger/coarser, due to an even somewhat higher scan velocity of the MetOp satellite (~820 km orbit altitude).

*The systematic uncertainty usLF1, just co-illustrated for completeness in Figures 10a, d, g, j, is almost…" on p15, L24, to "…for three different missions (CHAMP, COSMIC, MetOp) show…"*

*on p45, caption of Figure 10, update to "…for CHAMP (a); COSMIC (d), MetOp (g), and simMetOp (j)." "…for CHAMP (b), COSMIC (e), MetOp (h), and simMetOp (k)." "…for CHAMP (c), COSMIC (f), MetOp (i), and simMetOp (l)."*

*on p46, caption of Figure 11, update to "…shown in colors (CHAMP (yellow), COSMIC (orange), MetOp (red), simMetOp (violet)). Left column:…"*

*Ok. I agree with the changes you applied to the manuscript. But, in any case, it would be nice if you can add around p6 L26-28 something summarizing the green highlighted part of your text. Regarding this, in any case, no evidence is provided related to the fact that the linear downward extrapolation is consistent also for COSMIC. Could you add a figure or an example which demonstrates this?*

4.      And, always on the definition of the input random uncertainty example profile, does it takes into account the **merging of open loop and closed loop data** somewhere in the lower part of the profile? I guess that the uncertainties related to the open loop and closed loop tracking should be different (because of different noise levels added by the different tracking behavior, because of different sampling rates between OL and CL, which is different from COSMIC and GRAS for example). Maybe the "boundary" set with the linear extrapolation downward the 30 km includes already these different uncertainty levels. But I expect that a discussion on these aspect will be added in the paper.

*As detailed in the response to comment 3 above, the current input uncertainty model does not include sophisticated estimations of the diverse real error sources in the POD and excess phase profiles, being part of L1a processing, But as noted, this is not needed for the algorithm introduction and demonstration context of this study; and these sources will be accounted for in future when the rOPS L1a processor provides the profiles and uncertainties as input to the L1b processor.*

*It is not a matter to include "sophisticated" estimations of uncertainties characterizing the open/closed loop tracking. I expect that somehow these uncertainties have different magnitude. Could you confirm/specify that the linear extrapolation downward 30 km is supposed to include also the effect of such different uncertainty magnitudes? And, in any case, it would be nice to check and better addressed this in a future study, when the error propagation chain will start from the very beginning.*

5.      All the analysis is based on the covariance matrix propagation through linear (or linearized) operators. This is fine. But what are **the residuals effects due to the linearization**? I guess that the most "critical" linearization is the one applied to impact parameter retrieval (in GO). In that case I guess you can quantify the

second/higher order effects by considering more terms in the Taylor's expansion of the operator.

*Relative to the state profile magnitude, the magnitude of the uncertainties is generally very small, and the uncertainty of the uncertainties is even smaller. The relative accuracy of the uncertainty estimates we aim to achieve is within +/-3% when quantitatively determined from Type A evaluation (already comparable to a relative accuracy of the state profile of about 10^-4). The main source of uncertainty in the uncertainty estimates stems from Type B evaluation (particularly for the estimated systematic uncertainty), where it is almost impossible to push the error of the uncertainty estimates to below +/-10%. Given this context we thus consider the ~2% error due to the linearization in the propagation of the bending angle random uncertainty (we agree, the relatively largest error source) as acceptable (p21, L31).*

*Not fully clear. Are Type A and Type B different approaches to evaluate the uncertainty magnitude, being the Type A the one including statistical methods like mean, std? So, what is the final boundary to the relative uncertainty of the uncertainty? +/- 3% or +/- 10%? And how you can demonstrate that 2% is a reliable measure for the uncertainty associated to the "linearization" of the RO operators?*

*Anyway, it would be nice that your answer is summarized somewhere in the manuscript.*

6.      Finally, the most important references related the main topic of the paper (the description of rOPS and of the algorithms involved) are presentations to conferences or not accessible technical reports. This is a pity. **But at least, for the presentations it would be nice to have the full web-link specified**. The paper is absolutely worth to be published, but some review/clarifications should be provided. That's why I suggest for a major review. Specific and technical comments are provided in the next two sections.

*Yes, the general manuscript introducing the rOPS is still in preparation and so cannot be cited yet. We agree that providing ready access to the presentations is thus useful. We therefore added the web-links for the following presentations:*
*p32,     L28:     Kirchengast     et     al.     2015:*
*http://irowg.org/wpcms/wp-content/uploads/2014/05/Kirchengast-IROWG-4.pdf*
*p32, L31: Kirchengast et al. 2016a:*
*http://meetingorganizer.copernicus.org/EGU2016/EGU2016-12035-1.pdf*
*p32, L35: Kirchengast et al. 2016b: http://wegcwww.uni-graz.at/opacirowg2016/data/public/files/opacirowg2016_Gottfried_Kirchengast_presentation_261.pdf*
*p33, L1: Kirchengast et al. 2017a:*

*p34,         L27:         Syndergaard         1999:* *http://www.cosmic.ucar.edu/groupAct/references/Sr99-6.pdf*

*And on the detailed algorithmic documentation "behind" the papers (also addressed by the reviewer in a further comment below): The main cited source for the rOPS algorithm documentation, the DAD (Kirchengast et al., 2017b) is indeed not yet publicly available in its current 1.7 version, because it needs to be completed and polished in some (other) chapters and in its introduction and overview. However, we can assure that the DAD is firmly scheduled as an open document, to be published as soon as the journal publications of the introduction and main components of the rOPS and the first reprocessing are complete (expected in 2018).*

*We therefore strongly think it is valuable to quote the DAD also in this publication, so that future readers of any rOPS article (including this one) can readily get to it as soon as it is public, since also many years from now they will find it clearly referenced in all relevant journal publications collectively introducing the rOPS. This "foresight" was also the justification for citing it as reference in the Schwarz et al. Earth Space Sci. (2017) publication, the Gorbunov and Kirchengast Atmos. Meas. Tech. (2017) in press publication, etc.*

*Ok. I agree with the idea but it would be nice to have this second part also clearly stated in the manuscript. It is really important to have at a certain point some ATBD describing the science behind rOPS. Could you please add a sentence which summarizes the green highlighted part of your text here above?*

———

Specific comments

**6: what does it mean that "the accuracy is also ensured on-orbit"? please specify**

*The way we read the decadal survey (NRC 2007, p 64) is that for climate benchmark data, the accuracy needs to be ensured by the remote sensing technique itself (as with the self-calibration property of RO), rather than through e.g. bias-correction on the ground, to ensure reproducibility in the future with independent means.*

*We improved it to "…the accuracy is also ensured on orbit, i.e., there is no need for calibration or bias correction in post-processing on ground (Leroy et al., 2006)."*

*Ok.*

**21-22: remove references to unpublished papers or to papers in preparation. It is already mentioned that work is on-going.**

*Ok, we dropped these references to "papers in preparation" and hence simplified the text at this place to "…are part of on-going work (Innerkofler et al., 2016)." and, two lines below in L24, to "…and corresponding L2b uncertainty propagation is on-going (Li et al., 2017; Kirchengast et al., 2017a)."*

*What does it mean that Li et al., 2017 is "in submission"? Pls provide a doi if it is already submitted but under review process or remove it*

**31: first mention of the use of simulated data for GRAS.**

*See response to major comment 3 above: a test-day with real MetOp data was included now and hence the real MetOp(-A) is also mentioned at this place now.*

*OK*

2.1 Methods: all the analysis is based on evaluating random/systematic uncertainties as complete independent processes. Could you please explain why the two are completely decoupled? This is probably trivial under the assumption that noise is normally distributed. But is this really the case starting from excess-phases?

*The criteria for separating the evaluation and propagation of estimated random and systematic uncertainties is that the processes giving effect to these uncertainties need to be uncorrelated (if the correlation term is zero, the other terms can be pulled out of the sum). This is independent of the shape of the probability distribution (if a process causes a biased mean of the pdf, it is counted to the (basic) systematic uncertainties, such that the random uncertainty can be assumed unbiased). The basic systematic uncertainty is caused by processes that repeated measurements under (almost) the same conditions could not detect, e.g. insufficiently defined physical constants, most model parameter errors, etc. It can thus not be correlated with effects of processes that give different results under repeated measurements.*

*The random und apparent systematic uncertainties on the other hand can be seen as uncorrelated for an analogous reasoning: Since we separate them based on correlation length, the apparent systematic uncertainty appears as a bias within the, e.g., 0.8 seconds of the Blackman window (correlation coefficient of unity over the window range), while the random uncertainties' correlation function typically decays down to zero within this range. So even if random and (apparent) systematic uncertainty were correlated at one side of the Blackman window, they will be clearly uncorrelated at the other. It is crucial, however, that the correlation lengths of the random and the systematic uncertainties remain clearly separable; would the random uncertainty be strongly correlated also over longer time windows (appearing as a bias), it could be also correlated to the systematic uncertainty (a situation that, favorably, does essentially not occur in practice for RO data processing).*

*Thus, in practice, while there might be (very) slight correlations between random and systematic uncertainties at times, we consider the additional insight from having random and systematic effects separately available more important than the problem of potentially (very) slightly overestimating the uncertainties of our RO profiles.*

*In order to further help understand this, we now added on p4, after L20, another sentence/small paragraph as follows:* *"Since the noise-type effects giving rise to short range-correlated random uncertainties can be considered uncorrelated to the biastype effects inducing long range-correlated apparent systematic uncertainties, and since both are uncorrelated to basic systematic uncertainties, it is insightful and possible with due care to estimate and propagate each of these uncertainties independently."*

*OK*

**9-10: not clear why the effects of temporal/spatial variations (which kind of variations have you in mind?) in repeated observations can be estimated based on an individual RO profile. This is one of the main issue. Describe/motivate this better. And, by the way, why this should be true based to oversampling in the RO raw data profile?**

*This relates to GUM concept. The idea behind this evaluation method is an assumed ergodicity of the essentially stationary observation system (in our case the gaseous atmosphere viewed as a random field), and thus equivalence between the repeated measurements within one RO event (e.g., excess phase sampled at many different atmospheric levels), and measurements from an ensemble of multiple RO events (e.g., excess phase from many events at a certain atmospheric level). Effects causing such ergodic random variations could be the thermal noise of the receiver system, to some extend random uncertainties in the positions/velocities of the receiver and transmitter satellite, the clocks, atmospheric small-scale fluctuations, residual ionospheric fluctuations.*

*We tried to improve the paragraph on p4, L8ff, to:* *"… Effects of unpredictable or stochastic temporal and spatial variations in repeated observations—like effects from fluctuations in the atmosphere or the thermal noise of the receiver system—could in principle be estimated by ensemble statistics from multiple RO events. However, since such effects are essentially stationary in a statistical sense, we can estimate their statistics also from individual RO event data, given their high noise-resolving sampling rate. These effects are included in the estimated random uncertainties."*

*I do not fully agree with this. I expect that the distribution [A], characterizing the effects related to fluctuations in atmosphere is completely different from the (normal) distribution [B], characterizing instrument noise. [A] depends on location, [B] not (well, maybe during day/night the instrument noise is a bit different). So the excess phase from an ensamble of RO observations at a certain atmospheric level or excess phase sampled at many different levels do not define the same ergodic process. Maybe I've misled something. Could you please better specify this in the manuscript?*

**29: what do you mean with ionospheric noise? Here you are describing the noise in excess-phase measurements, so on both L1 and L2. Are you referring to**

scintillations? I don't think these can be treated as "normally distributed noise". Define this better.

*Ok, we improved this and wrote now "…the receiving system noise (i.e., thermal noise and residual clock estimation noise) and the ionospheric noise (from scintillations induced by ionospheric irregularities) are essentially normally distributed overall (Kursinski…). These noise sources are the main contribution to…"*

*OK, even if I'm not 100% sure that the ionospheric noise is really normally distributed.*

**29: Why random errors on positions/velocity are not considered here (later on, page 5 #30 you introduce such effects within the systematic uncertainties)?**

*Our rOPS L1a-related studies concerning uncertainties in transmitter- and receiver orbit positions and velocities show that the magnitude of temporal variations within the typical RO event duration of just 1 minute or less for scanning over the 80 km to middle troposphere altitude range are very small, i.e., orbit arcs are in practice very smooth at this RO-event timescale over the altitude range where excess phase magnitudes are less than 100 m (i.e., where the typical mm-level to cm-level noise sources will possibly be non-negligible). POD errors may thus be considered (quasi-)random on full-day timescales or so, but well classify as apparent systematic uncertainties within the 1-min time intervals of individual RO events.*

*Ok. But please, it would be nice to have this clearly stated in the manuscript.*

**24-25: not clear what "is aligned to a joint resolution..." mean. Please rephrase/clarify.**

*We rephrased p5, L24, to: "...the resolution of all profiles is brought to a common altitude-dependent resolution, which reflects..."*

*OK*

**7: It would be nice to motivate clearly here why you are using simulated data for propagating the uncertainties for the GRAS case.**

*We used the simulated data as a lower bound 'best-possible' case, to see how lownoise data would be propagating through the algorithm. We now also processed real GRAS data additionally to the simMetOp data – see under major comment 3 above for all the related updates.*

*OK*

**13-14: clarify why the "baseband" approach allows to avoid biases from numerical operations on near exponentially varying profiles (also repeated at page 17, #11-12).**

*Ok, improved this sentence to "…varying RO profiles, since the model profiles that we derive from short-range (24 h) forecasts of the European Centre for MediumRange Weather Forecasts (ECMWF) skilfully subtract the (near-)exponential variation. The remaining increment profiles that we need to treat numerically then appear to be very linear and with low dynamical range, which leads to very low residual numerical errors of operators such as filters and derivatives."*

OK

**4-7: here is where the downward extrapolation based on a GRAS "model" is described. See point 3 of General Comments.**

*To clarify our view of the nature of the input profiles used, we changed on p6, L26-28, to: "While in future the excess phase random and systematic uncertainty profiles will be more rigorously estimated by the rOPS L1a processor (Innerkofler et al. 2016) and provided as input to the L1b processor, they had to be estimated for this study from existing excess phase profiles with realistic noise (we chose UCAR/CDACC ones) and simplified modeling. To this end..."*

OK

**4-7: why linearly extrapolating downward and constantly extending upward a "random" uncertainty profile does not introduce a systematic effect?**

*We do not manipulate the state profile, only the magnitude of the random uncertainty profile. Technically, of course, the random uncertainty is not completely correct at the extrapolated sections, but this is equally true for the section with the noise-based estimation. As described in the response to comment 3, we only create a proxy-input to work with. Also, we only manipulate the magnitude (variance) of the uncertainties, not the correlation structure.*

OK

**8: Please clarify further why all the noise components responsible for random uncertainty at excess-phase level are uncorrelated (also referring to the ionospheric noise). I believe that at carrier phase level noise is uncorrelated (we have basically phase noise). But when geometry is removed as well as clock biases, and when the result is interpolated to match 50 Hz sampling rate (maybe between open and closed loop observations), I'm not sure that the various components are still uncorrelated.**

*Our assumption is that these random effects are only correlated within the range of the BWS-filter width and we can thus start the L1b processing with the assumption of*

*vertically uncorrelated errors. The analysis of the uncertainties in the L1a processing will show whether we will be able to uphold this clear distinction (i.e., in practice, the rOPS system checks this expected very-narrow-correlation between the L1a and L2b processing, by empirically estimating the excess phase correlation matrix from a large ensemble of L1a output profiles around the "currently processed day"; in case the random error correlations would appear too strong, we would feed the L1b processor with a correlation matrix that is non-diagonal accordingly). On the other hand, if a bias is found introduced and determinable, it will be corrected for (according to the GUM); if we can't determine it, an increase of the residual systematic uncertainty bound estimated would be the way to account for these components.*

*We have added on p7, in L10, the sentence: "…for both channels. In case the future excess phase data from the rOPS L1a processor exhibit non-negligible correlations for some data from some of the RO missions, we will account for these correlations in R_Lr, since our L1b algorithm (Section 3) is prepared for full covariance propagation."*

 *OK, perfect.*

**14: Being he entire analysis based on a COSMIC profile, why for the MC validation you started from a simulated CHAMP excess-phase profile and not from a realistic COSMIC profile? Motivate it.**

*Ok, we added on p7, L15: "... receiver system errors superimposed). Using an 'errorfree' profile as basis, the particular simulated profile just serving as a representative RO profile to illustrate the MC validation, allows us to strictly ensure the consistency of the random uncertainty of the input profile with the ensemble of superimposed error profile realizations."*

 *OK*

**24: in defining the input random uncertainty profile, you are computing the excess phase mean value between 60-70 km. Is the constant part of the systematic uncertainty consistent with such a mean?**

*No, in the current simplified input uncertainties modeling that we use in this study for introducing the algorithm (not focusing on realism of input and output, as explained in our response to comment 3 now), we don't consider this "empirical value" as viable basis for quantifying systematic uncertainty.*

*OK. And I guess that my comment was not appropriate, since the mean of the excess-phase at 60-70 km is not 0 because of systematic uncertainty but because an atmospheric effect (very small) is still present there.*

**Eq 6 and 8: Explain why the "gradient" (1/3exp(xx)) for the random and the systematic uncertainties below zGrad is different.**

*See our response to comment 3, where we explain our simplified modeling below 30 km, and note the improvements we did to the text in the revised manuscript. Specfically to the systematic uncertainty, we in addition extended on p7, L24, to: "…linear uncertainty gradient in the troposphere; as noted above this simplified modeling will be replaced in future by realistic uncertainty estimates received as L1b retrieval input from the L1a processor (Innerkofler et al., 2016)."*

*OK*

**20-21: not clear the sentence. By the way, it would be nice to show the relative error (to the state profile) instead the absolute one.**

*We reconsidered this but preferred to stick to the rationale of our original choice here – i.e., the main two reasons why we chose to plot the absolute uncertainties, are 1. that the range of the uncertainties is relatively constant in absolute terms but strongly decreasing in relative terms, particularly for the excess phase and Doppler shift uncertainties, and 2. that when plotted in relative terms, it adds difficulty and ambiguity to the interpretation of the results, due to the influence of the denominator. Furthermore, to keep the plots consistent throughout all figures, we preferred to keep ALL subplots of all these figures in absolute terms, optimizing comparability (we also tried double-axes but it overcrowds the figs, or considered to show both separately but that leads to too many figs in what is a long paper already).*

*OK*

**25: What is written there is true, but this is not clear whether these results (and the merging with bending angles uncertainties derived by GO processing) are available or not (#3-5 p 11 it is clearly stated that their integration in rOPS is not yet finished). I suggest to remove any reference to uncertainty propagation through WO/merging unless the results can be shown. See general comment 1).**

*Please refer to our response to general comment 1, we hope with that extensive answer (and the related improvements) our solution to this issue in this paper became clear.*

*OK*

**6-8: this is definitely not clear. How you can "artificially" substitute WO with GO uncertainty propagation, being the vertical domain of their applicability different?**

*Please again refer to our response to general comment 1, also this should be clear now.*

*OK*

**17 and 20: Not clear why the low-pass filtering of alphaM1 provides so different random uncertainty levels and broaden the correlation functions if the cut-off frequency is the same of the one already used to filter the corresponding excessphases. I expected negligible or a smaller effect.**

*The operations on the BWS-filtered profile, particularly the Doppler differentiation, create short-range correlation (anti-correlation) in the retrieved profiles and thus high(er) frequency variations, which are affected by the second filtering.*

*To make this more clear on p11, L14, we improved the sentence(s) to: "The chosen filter cutoff-frequency for k = 1 is fc1 = 2.5 Hz, same as the basic filtering (Section 3.1.1), just to ensure clearness of any higher-frequency effects from operators after the initial excess phase filtering (e.g., from Doppler shift derivation that induces shortrange anti-correlation effects). For k = 2, the cutoff-frequency fc2 is set noisedependent, between 2.5 and 0.5Hz (boxcar equivalent width of 0.2 to 1.0 s)."*

*OK*

**23: I found inconsistent defining the input data for the MC simulations by using a "background" error-free excess-phase profile taken by a simulated CHAMP event (mentioned not here but at page 7, #14) plus error realizations taken by a COSMIC dataset. The description of this input dataset should be better motivated and clarified. Moreover the input for the covariance matrix (for the MC simulation) seems to be some standard profile uSTD,Lr not well identified.**

*Please see our response to the comment concerning p7, L14, above, and in particular the response to the major comment 3, including the related revisions; this should clarify our approach.*
*To put it in different words again here, for the MC validation – in our eyes and experience – the most relevant aspect of the input profiles & uncertainties is that they are exactly the same as the ones used for the CP (covariance propagation). If this criterion is fulfilled, technically ANY reasonable 1-D profile could serve as basis profile, for the purpose of validating the CP through the MC ensemble (with the latter ensemble constructed by superimposing random error realization profiles on the 1-D profile).*

*Ok. But, it would be nice if you can add something summarizing the green highlighted part of your text to strength the idea behind your approach.*

**28: Here you are saying that the comparisons between MC simulations and covariance matrix propagation (CP) are carried out in terms of their decomposition**

(error profiles and correlation functions/lengths). This is fine but it is not clear where the CPpropagated covariance matrix comes from. What is the input? The same COSMIC profile taken as example in the previous discussion? Another excess-phase profile (more likely yes, since the results shown in Fig 9 [red, blue plots] are bit different from the same results shown in the previous figures. In #19, p7, you mentioned that "the same standard profile was used as input for the CP"). Rephrase and clarify the entire introduction to Set. 4.

*Thank you for pointing this out, we seem to have created a misunderstanding from some lack of a bit of info how the CP matrices come into play here.*
*We now updated the text on p12, L28, to: "...are calculated (Items b-g in Figure 2). Using the same input profile and uncertainty information as used to specify the MC runs (described in Section 2.2), the retrieval is then also run with covariance-based uncertainty propagation and the resulting CP-propagated covariance matrices $C_X^{CP}$ are compared to the MC-derived matrices $C_X^{MC}$. In order to be able to attribute..."*

*OK*

**7: the information provided in Fig 9 are not so clear. First of all are the random error profiles taken to lye between the boundaries provided by blue/orange lines used to plot Lr (thus varying up to 20 mm)? Could you use other styles to plot such information (dotted lines or different colors). Why the area within the LF random uncertainty is colored?**

*We tried a lot to make this fig informative, and in terms of how to show we did not find a really better solution. However, we clarified its explanation in the caption in that we updated the caption of Fig 9 in several places, as follows:*
*in first sentence to "…consecutive retrieval steps are shown for LF (a-c; in a-b also for Lr) and Dr (d-f) relative to…"; in second sentence to "…shows estimated random uncertainties for $f_{T1}$ (CP in blue, MC in black, in (a) $u^r_{Lr1}$ in light blue), the middle column for $f_{T2}$ (CP in red, MC in black, in (b) $u^r_{Lr2}$ in orange), the right column…";*
*in third sentence to "…variance propagation ('VP') results (light blue) for $\alpha_r$ in (n)."*

*OK*

**13-14: why it is "obvious" that the CP delivers the correct off-peak results? Provide a clarification.**

*Since we start with a diagonal covariance matrix with no correlation between distant elements and only apply a filter with a limited filter-width (corresponding to a bandmatrix), the correlation of the current element with any element outside the filter window must be zero.*

*We updated on p13, L14, to: "...that the CP delivers the correct off-peak results (i.e., zero; the off-peak elements outside the BWS filter window must nominally be zero)."*

*OK*

**17-21: rephrase a bit the sentence, since it is not clear.**

*Ok, we changed the text part on p13, L17ff, to: "For comparison, in Figures 9a to 9f, all quantities have been computed on the common time grid ('setting time' relative to time zero at 80 km altitude) with 50Hz sampling rate; and the corresponding impact altitude of the 'true' profile LTr is shown for additional convenience on the RHS axis. In Figures 9g to 9o, these bending angle quantities have been computed on the impact altitude grid; in these cases therefore the corresponding setting time of the 'true' profile is shown for additional convenience on the RHS axis."*

*OK*

**9: Are outliers defined in term of individual random uncertainty profiles? Or input Excess-phases profiles?**

*Ok, to better clarify this we updated the text on p14, L8-9, to: "...because they were detected as outliers based on the magnitude of their random uncertainty profiles (these outliers are not included in the number of profiles shown)."*
*We note that this was just a temporary solution for this L1b algorithms intro study; a more advanced QC is integrated as part of the rOPS L1a/L1b interfacing.*

*OK*

**7: this reference is a technical report, not accessible. Since it is quite important for the main topic of the paper, please provide an open reference if available (or avoid to cite e technical report which is not public).**

*Please refer to our response to general comment no. 6; we hope our preferred approach expressed there is ok and we can implement it also in this study.*

*OK*

Eq. A2, #14: The baseband approach allows the application of a certain model (in this case the excess-phase filter) to a delta variable, which is the original variable minus a zero-order model of that variable. In this particular case the excess-phase model is obtained by forward propagating ECMWF atmospheric field. Totally fine. My issue is that the filter is then applied to the delta profile only, and the filtered "delta" excessphase is then added back to the excess-phase model. High frequency components inherently contained in the model are not filtered out. Do I mislead something? Could you better clarify, since the reference points to a presentation given to a congress?

*The understanding of the reviewer is correct; and the answer is that the forwardmodeled profile is smooth over the scales of the filtering window, i.e., with negligible high-frequency components (as a side note, we experimented with making this proactively sure with some slight filtering of the refractivity model profile already before we forward-project it to bending angle, Doppler shift, excess phase; we found though, it was not needed so far; in the full chain processing we will cross-check this again whether there's any appreciable difference).*
*We adjusted the sentence on p17, L12, to: "…applying the filter only to the deltaprofile dLrm = Lr – Lm (with the model profile being adequately smooth over the scale of the filter window width). This approach…"*

*OK*

Eq A16, #13: the same as before.

*As used in this context, we note that the (adequately smooth) Lm and Dm model profiles are of course strictly consistent, i.e., Dm is an accurate derivative of Lm, and, vice versa, Lm is an accurate integral over Dm.*
*Here we now updated the sentence on p19, L7-8, to: "…the zero-order Doppler shift model profile Dm is added (the latter also available from the forward modeling, in a form strictly consistent with the excess phase model profile Lm)."*

*OK*

Eq A17, #15: The different coefficients identified in the first two and last two lines of $A^{L2D}$ matrix takes care of boundary effects in applying the 5 points derivative scheme. Could you please provide a further clarification on these values or a proper reference? Also, the denominator (12 Delta t) is not so straightforward.

*Ok, the reference to Syndergaard (1999) serves this.*

*OK*

**5: about the "mean" tangent point location. How it is computed? Is it the lat/long of the intersection between the GPS-LEO line with WGS84 when the line is tangent to the ellipsoid (Straight Line Tangent Height = 0). Is there any effect on the uncertainty propagation in the bending angle profile due to an error in locating the center of symmetry? Could you provide a clarification on this?**

*Ok, we added in a text part on p21, L5ff, as follows: "…at the mean tangent point (MTP) location of the RO event. The MTP location is defined as the geodetic (geographic) location on the WGS84 ellipsoid, where the straight-line path between transmitter and receiver touches this ellipsoid, i.e., where the straight-line tangent height is zero. This can be computed with very high accuracy at the sub-meter level (see Scherllin-Pirscher et al. (2017) for more details on the geolocation accuracy of RO). Using the MTP location's center of local curvature…"*

*OK*

**9: the impact parameter retrieval is "mildly non-linear". Fine. But what are the effects of the non linear part in the uncertainty propagation? (see also point 5) in general comments).**

*Please refer to our response to general comment 5, where we have answered to this.*

*Since I've still some issue with the answer you provided in 5, this comment is still valid.*

**31 - 3 p.22: This sentence is crucial but not clear. Is 2% the residual error on the bending angle or impact parameter left to the non-linearity (see above comment)? You are saying that the assumption is reasonable given the high quality of the forward modeled profiles? Why the high quality related to this error?**

*As we also allude to in our response to comment 5, we use the 2 % error due to the linearization (as estimated by Melbourne 1994) as reasonable bound for another error, namely the error we make by replacing the "true" impact parameter by the "forecast" impact parameter. We then argue that, since the "replacement error" due to this is smaller than the "linearization error" of up to 2 %, we find it acceptable. Both contribute to the inaccuracy of the random uncertainty estimate ur_alphaG,i and therefore we increase this uncertainty estimate by 2 % (Eq.A29). (We admit, we also in some view do not fully play the GUM recommendations at this spot as we increase the random uncertainty estimate to be conservative instead of explicitly co-estimating also these small further contributors which we theoretically could with effort). However, and important as a cross-validation, the MC validation shows that the magnitude of the combined effect of these two errors appears to be small, since the MC covariance matrix is calculated without the linearization (but using A22-A25) and the results are very close to the results of the CP (which does use the linearization).*

*To better clarify this now we updated the corresponding text on p21, L31 - p22, L2, to: "We use the forward-modeled impact parameter $a_{tm}$ instead (i.e., adopt $a_{tm} = a_{Tt}$ )*

*and accept the additional error thus incurred, assuming it is smaller than the 2 % relative error due to the linearization estimated by Melbourne et al. (1994). This is a reasonable assumption given the high quality of our forward-modeled profiles derived from ECMWF short-range forecast refractivity fields.*

   *As a consequence we have to accept that the overall inaccuracy of our random uncertainty estimate cannot be brought below 2 %. Therefore, to ensure that our simplified estimate does not underestimate the real uncertainty, we account for the linearization error by multiplying a factor f_uolin = 1.02 to the uncertainty…";*

*and in Eq.A29 we made the RHS term more clear, changing it to*

*"= f_uolin ⬜ ur_alphaG(za),i .";*

*and in the line below (p22, L5) we updated to: "In this way we acknowledge that although the calculation..."*

  *OK*

A29, #4: G(za) also in the first member? I'd say that is the linearization error of 2% applied to all the levels?

*Yes, it is a conservative correction of our random uncertainty estimate over all gridpoints, by a factor of 1.02 (see response to previous comment).*

  *OK*

**7: in Eq A29, the random uncertainty profile ualphaG is already interpolated to common monotonic impact height grid. Why it is repeated here?**

*For e.g. two transmitter frequencies: The two random uncertainty profiles calculated in A28 are the uncertainties of the two bending angles relative to coordinate variable z_a, but the two 1D vectors (of the bending angle profiles) still lie on the common time grid points (and these correspond to different z_a values for each individual profile). So in order to now state each bending angle vector on the same common impact altitude grid, we would need to interpolate bending angle 2 to the grid of bending angle 1 (in practice, because we want a monotonic grid, we modify it first).*

  *OK. Sorry my comment was not appropriate.*

**18: "zero-order contributions and no terms higher than...". Is this the Taylor expansion used to linearize A22-A24 (zero-order is already used to identify the model applied to the baseband approach for the state vector retrieval)? Referring to point 5) in the general comment, you can estimate the effects of non-linearity on uncertainty propagation evaluating ualphaG also including higher order terms.**

*Ok, to avoid potential "terminology confusion" we improved on p22, in L18 to "…state quantities (serving as zero-order state) and no terms higher than…"*

*And thanks for the thought (again) on the potential utility of explicitly estimating also higher-order terms to gain some estimates also on the uncertainty of the uncertainty estimates. We understand this and have loosely some (cluttered) notes in this*

*direction but see it clearly beyond the scope of this introductory (version 1) algorithms introduced in this study (no surprise it implies a significant share of extra work to get this done in a neat form).*

 *OK*

A33, #20: why the derivative of alpha wrt theta is not considered?

*Thank you for spotting this, this was left as a bug in latex-setting here; of course this added term is needed under the square root (p23, L20); it is now added in there.*

 *OK*

Page 24 and Page 25

I'd remove both A2.2 and A2.3. See issue 1) in the general comment section And, by the way, the effect of the apparent systematic uncertainty in the lower troposphere seems "crucial" but not still fully accepted (the paper is under review). It is worth to have the WO/merging part properly described elsewhere in the future, as soon as the results associated to the uncertainty propagation through WO algorithms will be available.

*I hope we could address this concern sufficiently with our detailed response to general comment 1, and our related updates quoted there. As expressed there, we strongly consider our approach important for using this paper as properly in future together with the other rOPS introduction papers.*

 *OK*

Page 27:

**3: here you are talking about a "more advanced form" of alphaL2 extrapolation which is described in a technical report not publicly available. Provide other references if possible or avoid to cite it.**

*We understand this concern, and in other contexts than the rOPS-related set of papers and underlying documentation would simply agree; but given it is this special context (see again our response to general comment 6) we strongly prefer to keep this way back-referencing to an even more detailed source that (we can guarantee) will become open "background publication" (and should be linked to all "foreground papers").*

 *I fully agree with your idea, but could you please better address what this "more advanced form" mean? Not the details, but something more specific description….*

**13: see above. Provide other references if possible, or avoid to cite it.**

*Please see the previous comment.*

 *OK*

Page 28:

Eq A48, #10: why the coefficient are squared? Is it because the CP foreseen to multiply the model and its transpose?

*It is due to the application of the general uncertainty propagation rule (e.g., GUM part for treatment of uncertainties for multivariate data) – can be viewed as a generalization to covariance matrices of the well-known uncertainty propagation rule of the form $u^2_z = (dz/dx)^2 u^2_x + (dz/dy)^2 u^2_y$ that applies to variances.*

*OK. Just for the sake of completeness, could you mention this on the text (or point to the GUM)?*

**15-17: Clarify this sentence.**

*Ok, we updated on p28, L15-17, to: "where it is assumed that the systematic errors in $\alpha_{F1}$ and $\alpha_{F2}$ are positively correlated, i.e., have the same sign, and the associated uncertainty estimates are hence subtracted from one another (as the bending angles are in Equation A46). This assumption is reasonable, since the same sources of nonionospheric systematic effects apply to both frequency channels (Doppler shift, orbit velocity, and orbit position uncertainties)."*

*OK*

**29: the Healy/Culverwell cited paper refers basically to a new ionospheric correction schema. Are there plans to introduce this in rOPS and to evaluate also how uncertainties are propagated through it?**

*Yes, it is planned to implement the higher-order correction of Healy-Culverwell in the rOPS, and to evaluate how it possibly helps to reduce the us_RIB due to actively modeling and subtracting this "higher-order ionospheric bias".*

*OK.*

**19: see previous comments on this.**

*Please see the previous answers to this.*

*OK*

**Technical comments**

**28: typo: validation**

*Thanks, corrected to "validation" (p3, L28).*

*OK*

**28: introduce what the subscript "r" stands for, even if there is a table in support of all the definitions.**

*We added "retrieved" before "excess phase profile" on p5, in L27.*

*OK*

**24: make reference to Fig 3b)**

*We inserted this on p7 in L29 in the form "…smooth these simple uncertainty models around these transition altitudes (for the example profile, $u^s_{Lr}$ is visible in Figure 3b)."*

*OK*

**14: alphaG,k(za) is repeated. Moreover alphaM,k(za) is never shown. Remove it**

*These are two different variables, since the coordinate variable (the argument) changes. alphaM and the two different alphaG's are shown in figure 2, which we refer to in this paragraph. Also the algorithm for calculating alphaM and its uncertainties is presented (please see also our arguments concerning general comment 1).*

*OK*

**16: add "making negligible the systematic uncertainty integrated from the phase/Doppler"**

*Please see in the previous relevant comment how we have changed this; it is accounted for now.*

*OK*

**10: is this recombined MC covariance matrix the one defined through Eq. 10? If yes, put a reference to the equation.**

*Ok, added reference to Eq. 10 in L10, p13: "…how far down the MC covariance matrix (Equation 10) reaches."*

*OK*

**28-32: here you provided a mention another possible method for estimate uncertainties along the processing chain (variance propagation). Totally fine. But it would be nice that it is properly introduced somehow in Sect 2.1.**

*We inserted a paragraph on p5, L11ff, as follows: "Since the covariance propagation of random uncertainties requires extensive matrix multiplications for each measurement model along the entire retrieval chain, we also tested simpler variance propagation, for which correlations are ignored; Appendix B summarizes the relevant algorithms. However, as shown in Section 4, variance propagation unduly overestimates random uncertainties so that covariance propagation is required. …When the operator is linear…"*

*OK*

Page 14: "minor channel". Please reword this. In GNSS we do not have minor/major channels. We have channels associated to the lower and higher carrier frequencies or, simply channel associated to the L1 or L2 carrier frequencies. Also Page 26, #4 and #8, Page 27 #8. Page 45, Figure 10 caption (leading ? channel)

*We did not immediately find a better term expressing what we want to say with „leading channel" and „minor channel" from the point of view oft he algorithmic approach (not the point of view of GNSS, or RO as a method). We would, however, if we have a good idea also correct this in the editing of a final production manuscript.*

*OK, whatever terminology you want to use is fine, but it should be defined in the text. What is not clear to me is whether the leading channel can be (algorithmically speaking) the one associated to L2 or if it is always the one associated to L1.*

**9: typo: derivative scheme**

*Thanks, we corrected to "derivative" (L9, p19).*

*OK*

**7: I'd add: "In the GO approximation, the bending angle values at each grid point only depend on..."**

*Thanks, extended the beginning of the sentence on p22, L8, to: "In the GO approximation, the bending angle values…"*

*OK*

Figures 3-11: It would be nice to have also the uncertainty (random and systematic) plotted as relative values (relative to the state variable profile)

*Please refer to our response to comment p9, L20 (and one other related comment).*

*OK*